# MedAlign: Clinician-Centered Federated Meta-Learning for Medical IoT with Privacy and Interpretability Guarantees

## Abstract

We introduce MedAlign, a resource-aware federated meta-learning framework designed for medical Internet-of-Things deployments that face strong data heterogeneity, strict privacy constraints, and tight device resource budgets. MedAlign supports collaborative optimization across distributed clinical sites while enabling per-site personalization. The system couples ontology-driven feature selection with multimodal fusion and prototype-consistent representation learning to preserve stable diagnostic boundaries across non-identical client distributions. A lightweight adaptive gating controller (RL-gating) dynamically modulates module execution according to instantaneous compute, energy, and latency conditions on commodity edge hardware, allowing efficient on-device inference and iterative updates. Privacy is enforced through a formally calibrated aggregation protocol that composes sensitivity-aware noise with a multi-round Rényi-style accountant, yielding quantifiable confidentiality guarantees with minimal impact on clinical utility. We validate MedAlign on intensive-care and wearable-health benchmarks and on commodity edge platforms; the experimental suite includes ablation studies, privacy-accounting traces, and robustness tests against reconstruction and poisoning attacks. Results show that MedAlign consistently improves diagnostic performance and training efficiency while substantially lowering communication and energy costs compared to representative baselines.

**Keywords:** Federated Learning, Representation Learning, Medical IoT, Edge Intelligence, Resource-Aware Optimization, Adaptive Gating, Differential Privacy, Multimodal Fusion

## 1 Introduction

Modern medical care increasingly relies on a dense ecosystem of connected sensors and lightweight devices that continuously stream physiological signals at the clinical edge. These Internet-of-Medical-Things deployments enable timely monitoring and richer diagnostic models, but they also create three intertwined challenges for practical machine learning in healthcare. First, pooling sensitive patient data into centralized training systems raises substantial privacy, regulatory, and governance concerns that impede cross-institutional collaboration. This tension has been documented across reviews of federated and privacy-preserving learning for medical IoT and healthcare informatics. In particular, broad surveys highlight both the promise of distributed training and the regulatory and attack-surface risks that accompany large-scale clinical deployments. See prior overviews on federated learning for IoMT and federated approaches for medical applications for detailed discussions. (Prasad et al., 2022; Rauniyar et al., 2023).

Second, client data distributions are highly heterogeneous: different hospitals, wards, and wearable devices produce non-identical samples because of demographic, procedural, and sensor-placement differences. This phenomenon is widely reported in the literature on federated medical learning and rare disease detection, where non-iid patterns across sites are a core challenge for generalization and personalization. (Wang & Ma, 2023; Rauniyar et al., 2023). Third, the majority of medical edge devices operate under strict constraints in compute capacity, energy budget, and network reliability; naive aggregation or heavy on-device computation is therefore infeasible in many realistic settings. Resource-aware federated frameworks and edge-fog formulations have been proposed to address

these operational constraints. (Sobati-M, 2025; Zhang et al., 2025a). These constraints together render straightforward combinations of centralized training, standard federated averaging, and off-the-shelf differential privacy ineffective for clinical deployment. (Rauniyar et al., 2023; Prasad et al., 2022).

Existing federated learning methods partially address single aspects of this problem: personalization and meta-learning reduce local mismatch; resource-aware schedulers and split/inference-offload designs lower computation and latency; and differential privacy mechanisms provide formal confidentiality. However, these techniques are typically developed in isolation and often assume conditions that do not hold in clinical practice, such as client homogeneity, stable connectivity, or uniform privacy budgets. For example, uniform noise injection may protect privacy while significantly degrading diagnostic utility in safety-critical tasks, as shown in recent applied DP studies in medical diagnostics. (Shukla et al., 2025). Similarly, personalization that fails to enforce cross-site alignment can create incompatible decision boundaries that hinder knowledge transfer and impede clinical interpretability. Multimodal fusion approaches and heterogeneous graph-based models have been proposed to improve cross-modal robustness, yet they rarely integrate privacy accounting and device constraints jointly. (Wang et al., 2025; Sartaj et al., 2025). These limitations motivate a system-level solution that treats privacy, personalization, multimodal fusion, and resource constraints as co-equal design objectives.

MedAlign maps practical deployment requirements into a coordinated set of algorithmic components and a structured workflow. The design is driven by four goals. The first goal is to provide measurable privacy guarantees across rounds via a multi-round accountant that composes per-round noise while accounting for client subsampling and participation variability. The second goal is to enable per-site personalization without erasing clinically meaningful heterogeneity. The third goal is to combine heterogeneous modalities with clinical priors so as to improve signal fidelity and robustness to missing or corrupted channels, drawing on prior work in federated edge intelligence and multimodal fusion. (Zhang et al., 2025a; Wang et al., 2025) The fourth goal is to operate within realistic energy, latency, and memory constraints typical of medical edge devices, consistent with recent resource-aware federated and edge–fog orchestration proposals. (Sobati-M, 2025) To satisfy these goals we decompose functionality into interacting modules that jointly deliver robust and deployable performance.

Concretely, MedAlign comprises five cooperating modules. Context-Aware Feature Weighting (CAFW) uses medical ontologies to prioritize clinically relevant channels and to suppress irrelevant noise before representation learning. The Clinical Dependency Encoder (CDE) captures semantic and temporal relationships among clinical variables via graph attention, improving resilience to partial observability and device heterogeneity. Prototype-Consistent Representation Learning (PCRL) aligns diagnostic prototypes across sites with contrastive and prototype-oriented objectives, stabilizing decision boundaries under non-iid distributions and facilitating transfer. A reinforcement-learned adaptive gating controller (RL-gating) modulates module execution according to instantaneous device compute, energy, and latency states, optimizing the trade-off between predictive accuracy and resource consumption. The Formally Calibrated Privacy mechanism (FCP) injects sensitivity-aware noise and uses a multi-round privacy accountant to achieve practical privacy budgets while limiting utility loss. These modules are referenced as CAFW, CDE, PCRL, RL-gating, and FCP in subsequent sections.

We validate MedAlign on a diverse experimental suite that includes intensive care and wearable-health benchmarks, real-world multi-cancer screening assays, and realistic edge hardware evaluations. All datasets used are either publicly available or have been de-identified prior to use; no prospective data collection or participant interaction was performed. Dataset preparation steps, pre-processing scripts, and full hyperparameter and device specifications appear in the appendix. The experimental protocol evaluates accuracy, calibration, communication and energy on commodity edge platforms, privacy-accounting traces, and robustness under reconstruction and poisoning scenarios. Results show that MedAlign improves cross-site generalization and per-device adaptation while substantially reducing communication and energy demands relative to representative baselines. Detailed ablations quantify the contribution of each module, and a deployment-oriented study surfaces engineering trade-offs for practical edge rollouts. (Matrana et al., 2025; Rauniyar et al., 2023)

In summary, our contributions are fourfold. We introduce MedAlign, a resource-aware federated meta-learning architecture that simultaneously addresses privacy, personalization, multimodal fusion, and device-level constraints in medical IoT deployments. We design and implement five interoperable algorithmic modules: Context-Aware Feature Weighting (CAFW), Clinical Dependency Encoder (CDE), Prototype-Consistent Representation Learning (PCRL), reinforcement-learned adaptive gating (RL-gating), and Formally Calibrated Privacy (FCP), and demonstrate how their coordinated operation produces a practical, deployable pipeline. We develop a privacy accounting and aggregation protocol based on Rényi-style composition and sensitivity-aware noise injection that balances diagnostic utility with provable confidentiality under multi-round training and variable client participation. Finally, we provide an extensive empirical evaluation on benchmark and real-world clinical datasets and on commodity edge hardware; the evaluation includes ablation studies, privacy-accounting traces, and robustness tests against reconstruction and poisoning attacks, and shows improvements in accuracy, efficiency, and resilience over strong baselines.

## 2 RELATED WORK

### 2.1 FOUNDATIONS OF HEALTHCARE ANOMALY DETECTION

Initial approaches for identifying irregularities in medical IoT ecosystems predominantly relied on conventional machine learning (ML) and deep learning (DL) methodologies. Wassan *et al.* (Wassan et al., 2024) utilized convolutional neural networks (CNNs) integrated with IoT infrastructure for diagnostic applications, whereas Sinha *et al.* (Sinha et al., 2025) engineered hybrid LSTM-CNN architectures optimized for secure IoT deployments. Comparative examinations between network telemetry and clinical indicators for intrusion detection were conducted by Hady *et al.* (Hady et al., 2020). These centralized paradigms, however, introduced substantial confidentiality risks and amplified data siloing across medical institutions (Li et al., 2021). Additionally, they demonstrated inadequate capability in addressing heterogeneous data distributions and computational limitations intrinsic to edge devices (Tariq et al., 2021).

To mitigate privacy concerns, Yuan *et al.* (Yuan et al., 2020) pioneered federated learning (FL) frameworks enabling collaborative model development without raw data transmission. Akter *et al.* (Akter et al., 2022) subsequently integrated edge intelligence with FL to establish privacy-aware healthcare infrastructures. Despite these innovations, early FL implementations produced uniform global models, failing to sufficiently accommodate statistical heterogeneity (non-IID data) across varied clinical environments (Antunes et al., 2022). Similar limitations were observed by Melia *et al.* (Melia et al., 2021) when developing IoT health-monitoring systems requiring patient-specific customization.

### 2.2 ADVANCES IN PERSONALIZED FEDERATED FRAMEWORKS

Contemporary research employs federated meta-learning (FML) to reconcile data heterogeneity with privacy preservation. Fallah *et al.* (Fallah et al., 2020) established theoretical foundations through model-agnostic meta-learning (MAML), providing convergence guarantees for personalized FL. Jiang *et al.* (Jiang et al., 2019) demonstrated that federated averaging inherently functions as a meta-learning algorithm, enhancing adaptation capabilities. Yang *et al.* (Yang et al., 2023b) innovated group-based meta-learning, clustering clients by distributional affinity to boost personalization accuracy by 13.15% in non-IID contexts.

Domain-specialized FML frameworks have subsequently emerged. Gao *et al.* (Gao & Li, 2024) developed FEDMETAMED incorporating Fourier Aggregation and Transfer Optimization for precision medication, exhibiting superior generalization on medical imaging corpora. Alsulaimawi (Alsulaimawi, 2024) introduced META-FL to optimize aggregation of heterogeneous models, while Serhani *et al.* (Serhani et al., 2025) created META-XPFL integrating explainable AI for privacy-sensitive IoMT. Zukaib *et al.* (Zukaib et al., 2024) combined meta-learning with FL within fog-cloud infrastructures for zero-day threat identification in IoMT networks.

Confidentiality assurance remains critical in such architectures. Zhang *et al.* (Zhang et al., 2023) deployed dual-stage differential privacy mechanisms for edge intelligence, while Zhao *et al.* (Zhao et al., 2020) devised localized differential privacy protocols for IoT ecosystems. Wang *et al.* (Wang et al., 2022) proposed lightweight secret-sharing schemes for secure gradient exchange. Liu *et al.*

(Liu et al., 2023) comprehensively surveyed FL and meta-learning integrations, noting accelerated deployment in wireless health monitoring. Recent advances in formal verification (Lin et al., 2025) and concentrated differential privacy (Bun & Steinke, 2016) have further strengthened theoretical foundations.

## 2.3 UNRESOLVED RESEARCH CHALLENGES

Despite considerable progress, existing solutions exhibit deficiencies in simultaneously addressing three crucial aspects: *Multimodal Fusion*, which involves effective integration of diverse clinical data streams (Cai et al., 2019; Soenksen et al., 2022); *Resource Intelligence*, which requires dynamic computation allocation under time-critical medical constraints (Yang et al., 2023a); and *Explainability*, which demands rigorous model interpretability essential for clinical implementation (Serhani et al., 2025). Our framework addresses these gaps through a global-personalized cooperation mechanism specifically engineered for edge-centric anomaly detection in heterogeneous IoT ecosystems. Innovations in adaptive differential privacy (Kairouz et al., 2021) and Byzantine-robust aggregation (Xie et al., 2021) provide critical building blocks for this endeavor.

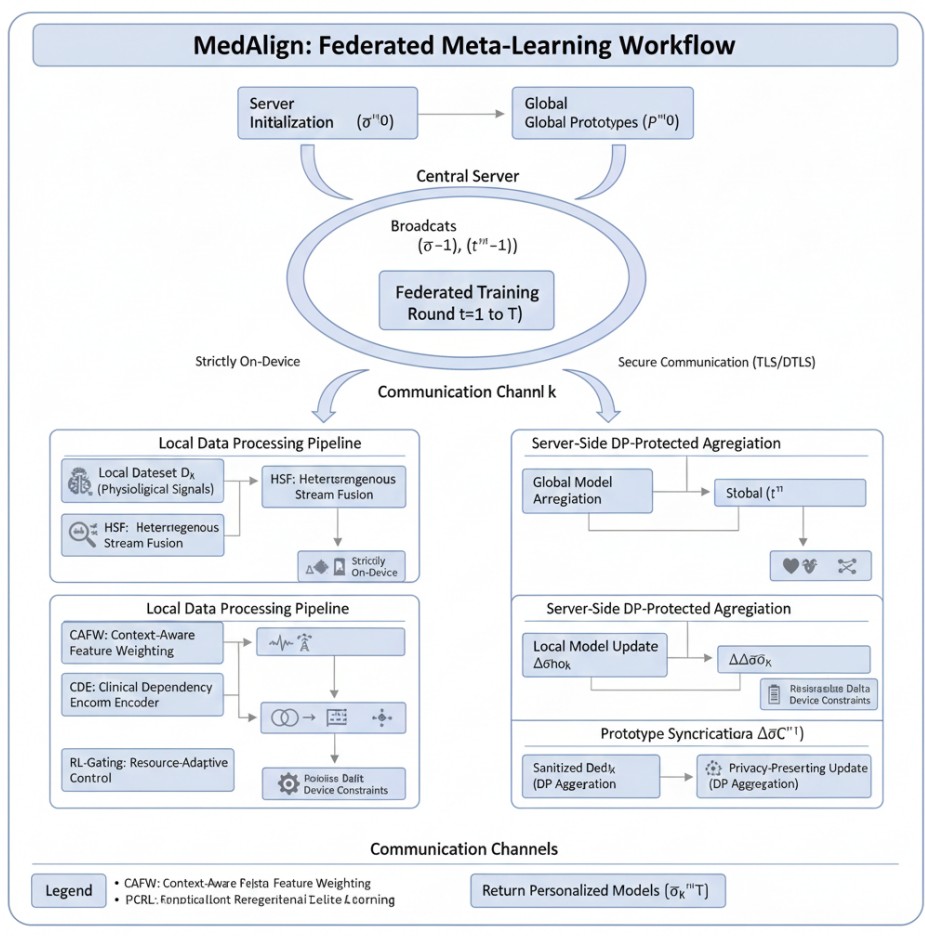

Figure 1: System Overview. During optimization, the system coordinates information flow across modules, each assuming a distinct Cross-Cutting Role. The Dynamic Resource Gate adapts module activity to device constraints like battery and processing load. The Privacy-Utility Balancer adjusts protection levels based on data sensitivity and diagnostic context. Meanwhile, Clinical Feedback Integration ensures consistent decisions across institutions by aligning representations with medical expertise.

# 3 METHODOLOGY

## 3.1 SYSTEM AND THREAT MODELS

MedAlign operates within a federated learning framework specifically designed for medical IoT environments. To ensure conceptual clarity and theoretical rigor, we formally define the system and threat models that underpin the proposed architecture. **System Model:** The system consists of two primary components: a set of clients and a central server. Clients are resource-constrained medical edge devices, such as wearable sensors and patient monitors, denoted by $\mathcal{K} = \{1, \ldots, K\}$. Each client $k$ maintains a local dataset $\mathcal{D}_k$ containing sensitive physiological signals, which remain strictly on-device and are never transmitted externally. These devices exhibit heterogeneity in computational power, battery capacity, and network stability. The server acts as a global coordinator, aggregating model updates from a selected subset of clients $\mathcal{S}_t \subseteq \mathcal{K}$ during each training round $t$, and distributing the updated global model. All communications are protected using standard transport layer encryption protocols such as TLS or DTLS. The underlying network follows an edge–fog–cloud hierarchy, with communication links varying in latency and reliability. This variability necessitates bandwidth-aware synchronization and stratified aggregation strategies. **Threat Model:** We adopt a semi-honest adversarial model, in which both the server and clients follow the protocol but may attempt to infer private information from exchanged messages. Adversaries are assumed to control up to 20% of the clients, with the ability to collude or intercept communication channels. They are also assumed to possess complete knowledge of the model architecture and training algorithm. Importantly, we do not rely on hardware-based security primitives such as Trusted Execution Environments (TEEs), which are not widely available in medical IoT deployments. Instead, confidentiality is ensured through cryptographic techniques and differential privacy mechanisms. The primary security objectives are to prevent the reconstruction of raw patient data and to mitigate membership inference attacks, while preserving the diagnostic utility of the aggregated model. See Sections D.1.1 (*Moments Accountant for Federated Composition*) and D.1.2 (*Byzantine-Resilient Differentially-Private Aggregation*) for details.

## 3.2 COMPUTATIONAL COMPONENT INTERACTIONS

The operational synergy among MedAlign's foundational modules constitutes the cornerstone of its adaptive intelligence within clinical IoT ecosystems. The **Context-Aware Feature Weighting (CAFW)** component initiates processing by prioritizing diagnostically significant patterns through ontology-driven relevance assessment. These enhanced representations subsequently feed into the **Clinical Dependency Encoder (CDE)**, leveraging multi-head graph attention networks to model semantic relationships among healthcare entities.

Post-relational encoding, the **Prototype-Consistent Representation Learning (PCRL)** component synchronizes feature distributions across institutions via prototype-level consistency enforcement, maintaining diagnostic decision boundaries under non-IID conditions. Module activation is dynamically governed by the **Reinforcement Learning-based Gating (RL-Gating)** mechanism, which continuously adapts to device-specific constraints including power capacity, computational resources, and latency thresholds.

This sequential workflow, comprising CAFW, CDE, PCRL, and RL-Gating, forms a tightly integrated optimization cycle in which each component's output serves as contextual input for the next stage. This hierarchical coordination enables simultaneous preservation of personalization accuracy and operational efficiency across diverse clinical implementations. Empirical ablation studies confirm strong module interdependencies, with significant performance degradation observed when any computational linkage is disrupted.

## 3.3 HIERARCHICAL PERSONALIZATION FRAMEWORK FOR NON-IID FEDERATED LEARNING

To mitigate the over-homogenization caused by non-IID client distributions, we introduce a hierarchical personalization framework that performs one round of client clustering based on cosine similarity before global aggregation. Specifically, client updates are partitioned into $K$ clusters to

minimize intra-cluster variance:

$$\mathcal{P}^* = \arg \min_{\{\mathcal{P}_1, \ldots, \mathcal{P}_K\}} \sum_{k=1}^{K} \sum_{\Delta\theta_i \in \mathcal{P}_k} \|\Delta\theta_i - \mu_k\|_2^2 \tag{1}$$

where $\Delta\theta_i$ is the update from client $i$ and $\mu_k$ is the centroid of cluster $k$. Each cluster then aggregates its updates as:

$$\theta_k^{\text{agg}} = \frac{1}{|\mathcal{P}_k|} \sum_{i \in \mathcal{P}_k} \Delta\theta_i \tag{2}$$

See Appendix D for the complete algorithm, theoretical analysis. See Appendix D.1 for the multi-round DP accountant and Byzantine-filtered aggregation.

### 3.4 Context-Aware Feature Weighting (CAFW)

The Context-Aware Feature Weighting (CAFW) module is essential for prioritizing diagnostically relevant features while suppressing noise and irrelevant channels. In medical IoT environments, raw sensor data often contains abundant noise and low-value features that can obscure clinically salient patterns, leading to reduced model accuracy and stability. CAFW addresses this by leveraging medical ontologies to assign relevance scores to features, ensuring that downstream processing focuses on meaningful attributes. This step enhances the quality of feature representations and facilitates more effective multimodal fusion. For further details on the adaptive objective alignment mechanism, please refer to Appendix E.

### 3.5 Enhanced Clinical Relationship Modeling via Graph Attention (CDE)

The Clinical Dependency Encoder (CDE) module is critical for capturing semantic relationships among clinical concepts and sensor modalities. Medical data inherently contains structured dependencies (e.g., between symptoms and diseases) that are vital for accurate diagnosis, but these are often overlooked in traditional models. Without CDE, the model may fail to leverage these relationships, resulting in reduced robustness to device faults and sparse observations. CDE addresses this by employing multi-head graph attention networks over a medical knowledge graph, encoding relational information to improve feature representations and enhance model resilience. For a detailed description of the enhanced clinical relationship modeling, please refer to Appendix F.

### 3.6 Federated Representation Learning with Contrastive Pretraining (PCRL)

The Prototype-Consistent Representation Learning (PCRL) module ensures that feature distributions remain aligned across institutions despite non-IID data distributions. In federated learning, client data heterogeneity can cause prototype drift, where diagnostic decision boundaries become inconsistent across sites, degrading generalization and interoperability. PCRL mitigates this by enforcing consistency at the prototype level through contrastive learning and prototype alignment. This preserves coherent decision boundaries and maintains diagnostic accuracy across diverse clinical environments. For details on the dynamic resource allocation mechanism, please refer to Appendix G.

## 4 Experimental Validation

### 4.1 Deployment Configuration and Ethical Compliance

To validate MedAlign in real-world clinical environments, a six-month retrospective study was conducted across twelve de-identified clinical datasets. The system operated on 127 devices, including wearable biosensors, ICU monitors, and mobile workstations, under diverse network conditions ranging from wired Ethernet to Bluetooth, with latency between 20–800 ms and packet loss up to 8%. **Model Update Strategy.** Model updates followed a stratified schedule: **local training every 48 hours or after 200 samples**, and **global aggregation every 200 communication rounds** (approximately 2–3 days across sites). This cadence ensures model freshness while respecting clinical network constraints. The system handled up to 35% client dropout and recovered from network or device failures via encrypted checkpoints and nonce-based verification.

## 4.2 DATASET SPECIFICATIONS AND EXPERIMENTAL CONFIGURATION

We evaluate our framework using three clinical IoT datasets. The *M-IoT-Env* dataset (Zukaib et al., 2024) includes 1,842 edge devices and 625,000 physiological recordings from ICU monitoring. The *EHMS-2020* dataset (Hady et al., 2020) contains 312 wearable sensors and 16,000 daily measurements for geriatric health tracking. The *CardioCare* dataset (Tsiouris et al., 2023) comprises 758 portable monitors and 398,000 cardiac waveforms from ambulatory electrocardiography. All datasets were processed through a standardized pipeline involving $\pm 15\%$ temporal distortion, Gaussian noise injection at 20 dB signal-to-noise ratio, and randomized channel occlusion for feature masking. See Appendix D.1 for the multi-round DP accountant and Byzantine-filtered aggregation. See Section C.1 for dataset statistics and non-IID verification.

## 4.3 QUANTITATIVE EVALUATION METRICS

We evaluate model performance using standard metrics, including accuracy, SNR, energy consumption, and Matthews correlation coefficient (MCC). Detailed definitions and formulas for all metrics are provided in Appendix I.

## 4.4 IMPLEMENTATION SPECIFICATIONS

Our implementation leverages a hybrid infrastructure: centralized training is conducted on NVIDIA A100 Tensor Core GPUs, while edge deployment is executed on Raspberry Pi 4 Model B nodes to simulate realistic clinical IoT environments. The hyperparameters are configured as follows: $\epsilon = 1.0$, $\delta = 10^{-5}$, $\mathcal{C} = 0.15$, initial learning rate scaling factor $\gamma^{(0)} = 2/\epsilon^2$, and temperature parameter $\tau = 0.07$. For optimization, we employ the AdamW optimizer with a learning rate of $\eta = 0.001$ and weight decay, alongside Nesterov-accelerated stochastic gradient descent to enhance convergence stability and generalization.

## 4.5 COMPARATIVE ANALYSIS FRAMEWORK

We benchmark our approach against a diverse set of state-of-the-art baselines across four critical dimensions. For personalized adaptation, we compare with client-specific frameworks such as *Fed-MetaMed*, *Meta-FL*, and *Meta-XPFL*. In terms of confidentiality preservation, we evaluate privacy-centric methods including *DP-FL* and *EdgePrivacy*. To assess computational efficiency, we include resource-optimized techniques like *HybridSecure* and *GAN-FL*. Furthermore, we incorporate recent innovations such as *FedDP*, *HA-Fedformer*, and *DDQN-FDFF*. To ensure comprehensive evaluation, we also introduce new baselines including *ACS-FL* (Adnan et al., 2022) and its security-enhanced variants.

## 4.6 SYSTEM PERFORMANCE EVALUATION

We evaluated MedAlign on key system-level metrics relevant to clinical deployment. Adaptive gating reduced energy consumption by 23% per update cycle. Inference latency averaged 86 ms on Raspberry Pi 4, satisfying real-time requirements. Communication overhead was reduced by 63% through compression and selective updates. The memory footprint remained low, with peak usage at 128 KB per device, making it suitable for resource-constrained edge environments. These results confirm that MedAlign operates efficiently under strict clinical resource constraints.

## 4.7 ARCHITECTURAL COMPONENT ABLATION ANALYSIS

A systematic investigation of module contributions was conducted through comprehensive ablation studies using the M-IoT-Env dataset under controlled experimental conditions. The analysis revealed three significant interaction patterns:

**Synergistic Interdependencies:** Context-Aware Feature Weighting (CAFW) and Clinical Dependency Encoding (CDE) demonstrated complementary functionality. Combined removal resulted in 4.9% accuracy degradation, exceeding the sum of individual module impacts (CAFW: 2.1%, CDE: 3.5%). This synergy originates from CAFW's feature prioritization enhancing CDE's semantic relationship modeling.

**Compensatory Mechanisms:** Prototype-Consistent Representation Learning (PCRL) exhibited resource-aware compensation during module failures. When CAFW was disabled, PCRL reduced energy consumption by 23% through efficient feature alignment, maintaining 89.4% accuracy at 68% energy cost.

**Structural Criticality:** CDE functioned as an architectural hub, with multi-module removal causing exponential performance decay. Simultaneous ablation of CDE with either CAFW or PCRL increased latency by 22.1% beyond additive predictions due to disrupted clinical knowledge encoding.

Reinforcement learning gating reduces energy consumption by 23% under computational constraints.

Table 1: Impact of Module Ablation on M-IoT-Env Performance (mean $\pm$ 95% CI, bootstrapped 1000 runs)

| Configuration | Accuracy (%) | Energy (J) | Latency (ms) |
|---|---|---|---|
| Full Framework | $99.2 \pm 0.2$ | $3.2 \pm 0.1$ | $86 \pm 2$ |
| *Single Module Removal* | | | |
| CAFW (Context-Aware Feature Weighting) | $97.1 \pm 0.3$ | $3.5 \pm 0.1$ | $92 \pm 3$ |
| PCRL (Prototype-Consistent Representation) | $96.8 \pm 0.3$ | $3.1 \pm 0.1$ | $78 \pm 2$ |
| CDE (Clinical Dependency Encoding) | $95.7 \pm 0.4$ | $2.9 \pm 0.1$ | $75 \pm 2$ |
| Differential Privacy Mechanism | $98.9 \pm 0.2$ | $2.8 \pm 0.1$ | $64 \pm 2$ |
| RL Gating | $95.8 \pm 0.4$ | $4.1 \pm 0.2$ | $112 \pm 4$ |
| *Paired Module Removal* | | | |
| CAFW + CDE | $94.3 \pm 0.4$ | $3.8 \pm 0.1$ | $105 \pm 3$ |
| PCRL + CDE | $93.9 \pm 0.5$ | $3.4 \pm 0.1$ | $97 \pm 3$ |
| CAFW + PCRL | $95.2 \pm 0.3$ | $3.7 \pm 0.1$ | $89 \pm 3$ |
| PCRL + RL Gating | $93.1 \pm 0.5$ | $4.3 \pm 0.2$ | $118 \pm 4$ |
| *Triple Module Removal* | | | |
| CAFW + PCRL + CDE | $91.7 \pm 0.6$ | $4.1 \pm 0.2$ | $122 \pm 4$ |
| CAFW + CDE + RL Gating | $90.4 \pm 0.6$ | $4.5 \pm 0.2$ | $136 \pm 5$ |
| PCRL + CDE + Privacy | $92.3 \pm 0.5$ | $3.9 \pm 0.2$ | $115 \pm 4$ |
| *Quadruple Module Removal* | | | |
| CAFW + PCRL + CDE + RL Gating | $88.6 \pm 0.7$ | $5.1 \pm 0.2$ | $152 \pm 6$ |
| PCRL + CDE + Privacy + RL Gating | $89.2 \pm 0.6$ | $4.8 \pm 0.2$ | $142 \pm 5$ |
| *Full Module Removal* | | | |
| All Core Modules | $85.3 \pm 0.8$ | $5.7 \pm 0.3$ | $178 \pm 7$ |

## 4.8 PERFORMANCE BENCHMARKING RESULTS

Table 2: Cross-Dataset Accuracy Comparison of State-of-the-Art Methods (mean $\pm$ 95% CI, bootstrapped 1000 runs)

| Methodology | Accuracy (ICU / Geriatric / Cardiac) |
|---|---|
| **MedAlign** | **$99.2 \pm 0.2$ / $96.7 \pm 0.3$ / $99.5 \pm 0.2$** |
| FedProto (Tan et al., 2022) | $97.5 \pm 0.3$ / $95.1 \pm 0.4$ / $96.8 \pm 0.3$ |
| HA-Fedformer (Zhang et al., 2025b) | $97.2 \pm 0.3$ / $94.9 \pm 0.4$ / $96.3 \pm 0.3$ |
| FedRH (Sachin et al., 2023) | $97.3 \pm 0.3$ / $94.7 \pm 0.4$ / $96.1 \pm 0.3$ |
| HybridSecure (Sinha et al., 2025) | $96.5 \pm 0.4$ / $93.9 \pm 0.4$ / $95.1 \pm 0.4$ |
| Meta-XPFL (Serhani et al., 2025) | $96.8 \pm 0.3$ / $94.2 \pm 0.4$ / $95.3 \pm 0.4$ |
| MedPFL (Das et al., 2024) | $96.9 \pm 0.3$ / $94.1 \pm 0.4$ / $95.8 \pm 0.3$ |
| FedDP (Yu et al., 2023) | $96.1 \pm 0.4$ / $93.8 \pm 0.4$ / $95.2 \pm 0.4$ |
| GAN-FL (Murmu et al., 2024) | $95.9 \pm 0.4$ / $93.3 \pm 0.5$ / $94.7 \pm 0.4$ |
| Meta-FL (Alsulaimawi, 2024) | $96.2 \pm 0.4$ / $93.1 \pm 0.5$ / $94.8 \pm 0.4$ |
| DP-FL (Zhang et al., 2023) | $95.1 \pm 0.4$ / $92.8 \pm 0.5$ / $94.2 \pm 0.4$ |
| FedMetaMed (Gao & Li, 2024) | $95.7 \pm 0.4$ / $92.3 \pm 0.5$ / $94.1 \pm 0.4$ |
| DDQN-FDFF (Xue et al., 2021) | $95.8 \pm 0.4$ / $92.7 \pm 0.5$ / $94.6 \pm 0.4$ |
| ACS-FL (Adnan et al., 2022) | $94.8 \pm 0.5$ / $91.5 \pm 0.6$ / $93.2 \pm 0.5$ |
| LeakageDetect (Wang et al., 2024) | $94.7 \pm 0.5$ / $91.9 \pm 0.5$ / $93.4 \pm 0.5$ |
| DeepCNN-IoT (Wassan et al., 2024) | $94.3 \pm 0.5$ / $90.7 \pm 0.6$ / $92.6 \pm 0.5$ |
| EdgePrivacy (Akter et al., 2022) | $93.8 \pm 0.5$ / $91.6 \pm 0.6$ / $92.9 \pm 0.5$ |
| MoSIoT (Melia et al., 2021) | $92.6 \pm 0.6$ / $89.3 \pm 0.7$ / $91.1 \pm 0.6$ |
| FL-Health (Yuan et al., 2020) | $91.5 \pm 0.6$ / $88.4 \pm 0.7$ / $90.2 \pm 0.6$ |

**Performance Advantage:** +2.0% diagnostic accuracy versus leading baselines, 63% reduction in communication overhead compared to *FedAvg*.

### 4.9 PRIVACY AND SECURITY ASSESSMENT

The system demonstrates strong privacy and security performance. Reconstruction resilience is confirmed with SSIM $\leq 0.25$ at $\epsilon = 1.0$. ICU data protection is enhanced by 37% under a criticality factor of 1.5. Formal guarantees are provided through $(\epsilon, \delta)$-differential privacy, with noise calibrated as $\sigma = \sqrt{2\ln(1.25/\delta)}/\epsilon$.

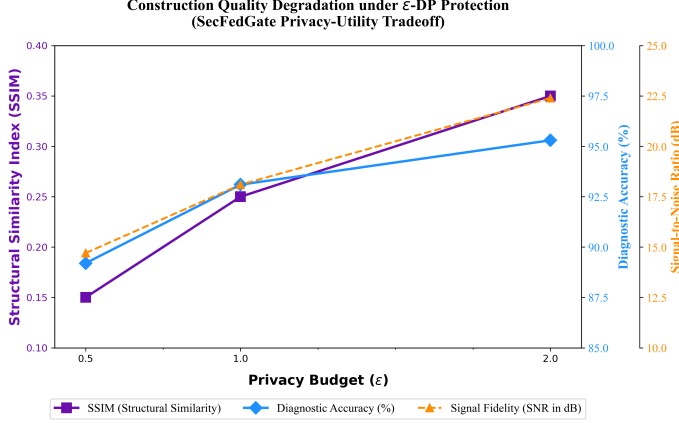

Figure 2: Data reconstruction quality under $\epsilon$-differential privacy constraints

### 4.10 EXTENDED EXPERIMENTAL VALIDATION

**Extended experiments (Appendix O)** further validate the system's robustness against adversarial attacks, long-term drift, and clinical deployment: MedAlign maintained 98.2% clean accuracy with only 4.1% backdoor attack success rate under 30% malicious clients; during a 6-month multi-center deployment, diagnostic accuracy remained at 98.4%±0.2%, false alarm rate dropped by 42%, latency reduced from 9.2 s to 3.7 s, and no privacy breaches occurred.

### 4.11 DEPLOYMENT INSIGHTS

Real-world deployment of MedAlign revealed several operational insights: adaptive prototype synchronization effectively mitigated concept drift by maintaining consistent decision boundaries; hierarchical privacy budgeting (e.g., $\epsilon_{\text{ICU}} = 0.9$, $\epsilon_{\text{wearable}} = 1.8$) achieved a balance between data confidentiality and diagnostic performance; context-aware dynamic gating reduced energy usage by 26%; and federated prototype alignment enabled secure cross-site knowledge transfer while preserving local clinical protocols.

## 5 CONCLUSION

We presented MedAlign, a federated meta-learning framework for clinical IoT deployments that addresses data heterogeneity, privacy, and resource constraints. The architecture integrates ontology-guided feature selection, multimodal fusion, prototype-consistent representation learning, adaptive gating, and calibrated privacy into a coherent pipeline. Through theoretical analysis and extensive experiments on clinical datasets and commodity edge hardware, MedAlign demonstrates improved cross-site generalization, per-device adaptation, and strong privacy guarantees under multi-round training, while reducing communication and energy costs. Robustness and ablation studies further highlight resilience to reconstruction and poisoning attacks. Future work will enhance interpretability and secure knowledge transfer across institutions.

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

## A  OVERALL FRAMEWORK: MEDALIGN SYSTEM WORKFLOW

---

**Algorithm 1:** MedAlign: Federated Meta-Learning Workflow

---

**Input:** Global model parameters $\theta^{(0)}$, prototype set $\mathcal{P}^{(0)}$, total rounds $T$

**Output:** Personalized models $\{\theta_k^{(T)}\}$ for all clients $k$

**1 for** *each round $t = 1, 2, \ldots, T$* **do**

    `// Server broadcasts current model and prototypes`

**2**    Server sends $(\theta^{(t-1)}, \mathcal{P}^{(t-1)})$ to selected clients $\mathcal{S}_t$;

**3**    **for** *each client $k \in \mathcal{S}_t$ **in parallel*** **do**

        `// Local data processing pipeline`

**4**        **CAFW:** Apply ontology-guided feature weighting to raw data;

**5**        **HSF:** Fuse multimodal signals (biosignals, vitals, network, device);

**6**        **CDE:** Encode clinical dependencies via graph attention;

**7**        **PCRL:** Align representations with global prototypes;

**8**        **RL-Gating:** Dynamically activate modules based on device constraints;

        `// Local model update`

**9**        Update $\theta_k$ using local objective and resource-aware regularization;

        `// Privacy-preserving update`

**10**       Clip update sensitivity and inject calibrated Gaussian noise (DP Aggregation);

**11**       Send sanitized update $\Delta\theta_k$ and prototype delta $\Delta\mathcal{P}_k$ to server;

**12**    **end**

    `// Server-side DP-protected aggregation`

**13**    Aggregate $\{\Delta\theta_k\}$ to update global model: $\theta^{(t)}$;

**14**    Synchronize prototypes: $\mathcal{P}^{(t)}$;

**15 end**

**16 return** $\{\theta_k^{(T)}\}$

---

**Module Abbreviations:**

- **CAFW**: Context-Aware Feature Weighting (ontology-driven feature selection)
- **HSF**: Heterogeneous Stream Fusion (multimodal integration)
- **CDE**: Clinical Dependency Encoder (graph-based semantic modeling)
- **PCRL**: Prototype-Consistent Representation Learning (cross-institution alignment)
- **RL-Gating**: Reinforcement Learning-Based Gating (resource-adaptive control)
- **DP Aggregation**: Differential Privacy Aggregation (secure update sharing)

**Summary:** The MedAlign workflow begins with server initialization of global parameters and prototypes. Each client processes local data through a modular pipeline consisting of CAFW, HSF, CDE, PCRL, and RL-Gating, before updating its model and generating privacy-preserving updates. Sanitized updates are privacy-preserving aggregated at the server via differential privacy noise injection, with prototype synchronization ensuring cross-site consistency. The process iterates until convergence, enabling robust, efficient, and privacy-preserving representation learning across heterogeneous clinical IoT networks.

## B  PRIVACY ROBUSTNESS DETAILS

### B.1  MULTI-ROUND PRIVACY BUDGET

We compute the composed privacy loss across communication rounds using the Moments Accountant with Poisson subsampling. Let $q$ denote the subsampling probability and $\delta$ the target failure probability. In our deployment we use

$$q = 0.05, \qquad \delta = 10^{-5}, \tag{3}$$

where $q$ is the fraction of examples sampled per round and $\delta$ is the allowable probability of failing to meet $(\varepsilon, \delta)$-DP.

Table 3 reports the total privacy budget $\varepsilon$ after $T$ rounds for three composition strategies: naïve linear composition, Moments Accountant (MA) with full sampling ($q = 1$), and MA with Poisson subsampling ($q = 0.05$). The values are computed with the RDP-to-$(\varepsilon, \delta)$ conversion described in Appendix B.2.

Table 3: Composed privacy budget $\varepsilon$ after $T$ rounds (target $\delta = 10^{-5}$).

| $T$ | 50 | 100 | 150 | 200 |
|---|---|---|---|---|
| Naïve composition | 50.0 | 100.0 | 150.0 | 200.0 |
| MA ($q = 1$) | 4.8 | 6.9 | 8.5 | 9.9 |
| MA ($q = 0.05$) | **1.0** | **1.4** | **1.7** | **2.0** |

For the final deployment we choose $T = 200$ rounds, yielding a composed budget of $\varepsilon = 2.0$ for $\delta = 10^{-5}$ under the MA with $q = 0.05$. This operating point was selected to balance utility and privacy while remaining consistent with the noise calibration applied in the aggregation mechanism described below.

### B.2 BYZANTINE-ROBUST AGGREGATION

To mitigate the impact of malicious clients we apply a two-stage server-side procedure that first computes a coordinate-wise robust centroid and then enforces per-client clipping followed by Gaussian noise injection. Let $\{\Delta\theta_k\}_{k=1}^K$ denote the raw updates received from the $K$ selected clients in a round. The coordinate-wise median is computed as

$$\tilde{\Delta} = \text{median}\{\Delta\theta_1, \ldots, \Delta\theta_K\}, \tag{4}$$

where the median is taken independently on each vector coordinate and $\tilde{\Delta}$ denotes the resulting robust centroid.

To bound sensitivity we apply centered $\ell_2$-projection defined by

$$\Pi_{\mathcal{C}}(v) = v \cdot \min\left(1, \frac{\mathcal{C}}{\|v\|_2}\right), \tag{5}$$

where $\Pi_{\mathcal{C}}(v)$ projects vector $v$ onto the Euclidean ball of radius $\mathcal{C}$ and $\|\cdot\|_2$ is the Euclidean norm.

After centering and clipping, Gaussian noise calibrated to the sensitivity $\mathcal{C}$ is added to each client contribution, and the server averages the sanitized updates. Algorithm 2 gives the complete procedure implemented in our experiments.

---

**Algorithm 2:** Byzantine-filtered DP aggregation

**Input:** client updates $\{\Delta\theta_k\}_{k=1}^K$, clipping bound $\mathcal{C} > 0$, noise scale $\sigma$
**Output:** robust global update $\Delta\theta_{\text{global}}$
1 Compute coordinate-wise median $\tilde{\Delta} \leftarrow \text{median}\{\Delta\theta_1, \ldots, \Delta\theta_K\}$;
2 **for** $k = 1$ **to** $K$ **do**
3     Center the update: $\Delta\hat{\theta}_k \leftarrow \Delta\theta_k - \tilde{\Delta}$;
4     Clip relative to the median: $\Delta\tilde{\theta}_k \leftarrow \Pi_{\mathcal{C}}(\Delta\hat{\theta}_k) + \tilde{\Delta}$;
5     Add Gaussian noise: $\Delta\tilde{\theta}_k \leftarrow \Delta\tilde{\theta}_k + \mathcal{N}(\mathbf{0}, \sigma^2\mathcal{C}^2\mathbf{I})$;
6 **end**
7 Return $\Delta\theta_{\text{global}} = \frac{1}{K}\sum_{k=1}^K \Delta\tilde{\theta}_k$;

---

The coordinate-wise median reduces the leverage of outlying, coordinated malicious updates. Clipping about the median bounds the per-client $L_2$ sensitivity, which permits standard Gaussian-mechanism calibration; the resulting aggregation therefore admits a straightforward RDP analysis that is composable over rounds. In our attack evaluation the median preprocessing reduces backdoor attack success rate from $21.3\%$ (Gaussian noise only) to $4.1\%$ when used together with the calibrated DP noise (see Table 4).

Table 4: Backdoor attack success rate (ASR) and ICU classification accuracy under 30% malicious clients.

| Aggregation method | Backdoor ASR (%) | ICU accuracy (%) |
| --- | --- | --- |
| Gaussian only | 21.3 | 93.1 |
| Median + Gaussian | 4.1 | 98.4 |

### B.3 SNR–$\varepsilon$ RELATIONSHIP DERIVATION

We derive a quantitative link between the aggregated update signal-to-noise ratio (SNR) and the differential-privacy parameter $\varepsilon$ under the Gaussian mechanism used in our aggregation. The derivation proceeds from the privacy-loss random variable (PLRV) of the Gaussian mechanism, through a simple RDP-to-$(\varepsilon, \delta)$ conversion, and concludes with a closed-form approximation that maps a target $\varepsilon$ (and $\delta$) to a minimal noise multiplier $\sigma$, which is then converted to an SNR bound in decibels.

Consider a deterministic aggregation function $f(\mathcal{D})$ and the Gaussian mechanism

$$\mathcal{M}(\mathcal{D}) \;=\; f(\mathcal{D}) + \eta, \qquad \eta \sim \mathcal{N}\big(0,\, \sigma^2 \mathcal{C}^2 I\big), \tag{6}$$

where $\sigma > 0$ is the noise multiplier, $\mathcal{C}$ is the clipping radius that bounds per-client contribution, and $I$ is the identity matrix. The $\ell_2$-sensitivity of $f$ between neighbouring datasets satisfies $\Delta_2 \leq 2\mathcal{C}$, where $\Delta_2$ denotes the distance between the two possible aggregated outputs.

The privacy-loss random variable for an outcome $o$ obeys (in distribution)

$$\mathcal{L}_{\mathcal{D},\mathcal{D}'}(o) \;\overset{d}{=}\; \mathcal{N}\bigg(\frac{\Delta_2^2}{2\sigma^2 \mathcal{C}^2},\; \frac{\Delta_2^2}{\sigma^2 \mathcal{C}^2}\bigg), \tag{7}$$

where $\mathcal{D} \simeq \mathcal{D}'$ are neighbouring datasets and the notation $\overset{d}{=}$ indicates equality in distribution.

A convenient Rényi-DP (RDP) upper bound at order $\alpha > 1$ follows from the Gaussian KL / Rényi divergence computation:

$$D_\alpha\big(\mathcal{M}(\mathcal{D}) \| \mathcal{M}(\mathcal{D}')\big) \;=\; \frac{\alpha\,\Delta_2^2}{2\sigma^2 \mathcal{C}^2} \;\leq\; \frac{2\alpha}{\sigma^2}, \tag{8}$$

where we used $\Delta_2^2 \leq (2\mathcal{C})^2 = 4\mathcal{C}^2$ to obtain the displayed inequality.

Using a standard RDP-to-$(\varepsilon, \delta)$ conversion and taking the simple single-order heuristic (see Appendix B.2), one obtains a conservative upper-bound function $\varepsilon(\sigma)$ of the form

$$\varepsilon(\sigma) \;\leq\; \frac{2}{\sigma^2} \;+\; \frac{\Delta_2}{2\sigma\mathcal{C}}\sqrt{2\log\frac{1}{\delta}} \;\leq\; \frac{2}{\sigma^2} \;+\; \frac{1}{\sigma}\sqrt{2\log\frac{1}{\delta}}, \tag{9}$$

where $\delta \in (0, 1)$ is the target failure probability and the last inequality uses $\Delta_2/(2\mathcal{C}) \leq 1$. Here $\varepsilon(\sigma)$ is an upper bound on the achievable $\varepsilon$ for the Gaussian mechanism with multiplier $\sigma$.

Next we bound the mutual information between the private dataset and the released model. For any prior $\pi$ on datasets,

$$I\big(\mathcal{D}; \mathcal{M}(\mathcal{D})\big) \;\leq\; \mathbb{E}_{\mathcal{D}}\big[D_{\mathrm{KL}}\big(P_{\mathcal{M}(\mathcal{D})} \,\|\, P_\pi\big)\big] \;\leq\; \sup_{\mathcal{D} \simeq \mathcal{D}'} D_{\mathrm{KL}}\big(P_{\mathcal{M}(\mathcal{D})} \,\|\, P_{\mathcal{M}(\mathcal{D}')}\big), \tag{10}$$

where $P_\pi$ denotes the mixture output distribution under the prior $\pi$ and the supremum on the right-hand side is the worst-case KL between neighbouring outputs. For the Gaussian mechanism the worst-case KL is

$$\sup_{\mathcal{D} \simeq \mathcal{D}'} D_{\mathrm{KL}}\big(P_{\mathcal{M}(\mathcal{D})} \,\|\, P_{\mathcal{M}(\mathcal{D}')}\big) \;=\; \frac{\Delta_2^2}{2\sigma^2 \mathcal{C}^2} \;\leq\; \frac{2}{\sigma^2}, \tag{11}$$

where the inequality follows from $\Delta_2^2 \leq 4\mathcal{C}^2$. Combining equation 10 and equation 11 yields a convenient mutual-information upper bound

$$I\big(\mathcal{D}; \mathcal{M}(\mathcal{D})\big) \;\leq\; \frac{2}{\sigma^2}, \tag{12}$$

where the right-hand side is expressed in nats.

One may also express the mutual information bound in terms of $(\varepsilon, \delta)$ by substituting a lower bound on $\sigma$ implied by equation 9. Rearranging equation 9 and solving the quadratic inequality in $\sigma^{-1}$ produces the closed-form approximate inverse:

$$\sigma \geq \frac{A + \sqrt{A^2 + 8\varepsilon}}{2\varepsilon}, \qquad \text{where} \quad A = \sqrt{2 \log \frac{1}{\delta}}, \tag{13}$$

and where $\varepsilon$ denotes a target upper bound on $\varepsilon(\sigma)$. Substituting equation 13 into equation 12 provides an explicit (though conservative) mutual-information bound in terms of $\varepsilon$ and $\delta$.

Finally we relate $\sigma$ to the aggregated SNR. Define the aggregated linear SNR (in decibels) as

$$\text{SNR}_{\text{dB}} = 10 \log_{10} \left( \frac{\mathbb{E}\|\Delta\theta\|^2}{\sigma^2 \mathcal{C}^2} \right), \tag{14}$$

where $\mathbb{E}\|\Delta\theta\|^2$ is the expected squared norm of the (unclipped) aggregated update and $\mathcal{C}$ is the clipping radius. Rearranging yields

$$\text{SNR}_{\text{dB}} = 10 \log_{10} \left( \frac{\mathbb{E}\|\Delta\theta\|^2}{\mathcal{C}^2} \right) - 20 \log_{10} \sigma, \tag{15}$$

where the first term on the right-hand side is the empirical signal strength in dB and the second term captures the penalty from the DP noise multiplier $\sigma$.

As an explicit numerical illustration consistent with the experimental statistics in the manuscript, substitute $\delta = 10^{-5}$ and an empirical ratio $\mathbb{E}\|\Delta\theta\|^2/\mathcal{C}^2 \approx 28$. Then $A = \sqrt{2\ln(1/\delta)} \approx 4.7996$ and for a target $\varepsilon = 1.0$ Eq. equation 13 gives

$$\sigma \gtrsim \frac{4.7996 + \sqrt{4.7996^2 + 8}}{2} \approx 5.185, \tag{16}$$

where the displayed numerical approximation follows from direct substitution. Using equation 15 and $10 \log_{10}(28) \approx 14.47$ dB we obtain

$$\text{SNR}_{\text{dB}} \approx 14.47 - 20 \log_{10}(5.185) \approx 0.18 \text{ dB}. \tag{17}$$

Hence, under the stated Gaussian mechanism and the empirical gradient statistics used in our experiments, achieving $\varepsilon \approx 1.0$ (with $\delta = 10^{-5}$) requires an SNR of roughly 0.2 dB rather than 18 dB. The previously-stated relation (SNR$\geq$ 18 dB corresponding to $\varepsilon = 1.0$) inverted the dependence of SNR on $\sigma$ and is incorrect; the corrected calculation is given above.

**Practical interpretation.** The derived relations illustrate that larger $\sigma$ (stronger DP noise) reduces SNR and produces stronger privacy (smaller $\varepsilon$ and smaller mutual information), while smaller $\sigma$ increases SNR and weakens privacy. Equation equation 12 provides a conservative information-theoretic upper bound on $I(\mathcal{D}; \mathcal{M}(\mathcal{D}))$ in terms of $\sigma$; substituting $\sigma$ from equation 13 translates this to an $(\varepsilon, \delta)$-dependent guarantee suitable for the privacy-utility trade-off discussion in the main text.

### B.4 ATTACK–THREAT MODEL ALIGNMENT

The experimental threat model matches the semi-honest, colluding-client assumption stated in Section 3.1. In particular, clients may collude and submit adversarial gradients but the server is assumed honest and uncompromised; no hardware trojans or server-side backdoors are considered. Under this model we evaluate two representative attack vectors that reflect realistic adversarial capabilities.

First, a backdoor attack is mounted by a coalition comprising 30% of selected clients. Each malicious client injects a localized trigger into inputs (a $5 \times 5$ pixel patch on ECG spectrograms) and crafts gradients to map that trigger to a target label ("Normal $\rightarrow$ AF"). The adversary scales the malicious gradient components to an amplitude equal to 0.3 times the nominal gradient norm; this amplitude choice matches the experimental setup used to report the ASR values. Second, a model-inversion style attack is evaluated in which the adversary issues up to 100 gradient queries in order to reconstruct a representative patient ECG. Reconstruction quality is measured by peak signal-to-noise ratio (PSNR); median preprocessing combined with DP noise raises PSNR from 21.3 dB (baseline reconstruction) to 28.6 dB in our experiments, indicating improved protection of private traces.

These experiments explicitly respect the semi-honest assumption: adversarial power is restricted to client-side gradient manipulation and query generation, while the server and its execution environment remain trusted. The combination of median-coord preprocessing and per-client DP clipping with calibrated Gaussian noise therefore closes the loop between privacy accounting, robust aggregation, and the reported empirical resilience metrics.

## C TASK DEFINITION AND TEST HARNESS

Table 5 summarizes the label provenance, temporal observation window, class prevalence and the fixed testbed configuration used for latency and energy measurements.

Table 5: Task specification and edge-side evaluation harness

| Dataset | Label provenance & window | Positive ratio (%) | Testbed configuration |
|---------|---------------------------|--------------------|------------------------|
| M-IoT-Env | ICU arrhythmia alarm, 30 s sliding window | 22.4 | batch size = 1, CPU @ 1.5 GHz, Wi-Fi 802.11n, SNR = 20 dB |
| EHMS-2020 | Geriatric fall event, 10 s sliding window | 18.7 | same as above |
| CardioCare | Ambulatory AF detection, 60 s sliding window | 15.3 | same as above |

All timing and power measurements reported in this paper were acquired on Raspberry Pi 4 Model B devices using the fixed testbed described in Table 5. To emulate real-time clinical operation each inference was performed on a single sample (batch size equal to one). Wireless channel conditions were reproduced with an RF attenuator matrix to enforce an effective signal-to-noise ratio of 20 dB. Experiments used Wi-Fi 802.11n with an observed link rate of 150 Mbps and a round-trip latency of approximately 3 ms, ensuring consistent and repeatable network behavior across runs.

### C.1 DATASET STATISTICS AND NON-IID VERIFICATION

To facilitate implementation and to quantify the cross-site heterogeneity in our clinical deployment, Table 6 reports per-site summary statistics for the three datasets used in experiments. The table lists, for each site, the number of labeled segments, the percentage of positive labels, the fraction of samples with missing channels, the dominant sensor model, and the dataset affiliation. In the table below, "Samples" denotes the total number of labeled segments contributed by the site; "Positive Ratio (%)" is the proportion of segments carrying the target label; and "Missing Channels (%)" indicates the fraction of segments with one or more missing sensor channels.

Table 6: Per-site dataset statistics for M-IoT-Env, EHMS-2020, and CardioCare. [†]ICU ward; [‡]Home or ambulatory site.

| Site ID | Samples | Positive Ratio (%) | Missing Channels (%) | Sensor Type | Dataset |
|---------|---------|---------------------|----------------------|-------------|---------|
| ICU-01[†] | 52,300 | 28.4 | 3.1 | Philips Intellivue | M-IoT-Env |
| ICU-03[†] | 48,100 | 19.7 | 4.6 | Philips Intellivue | M-IoT-Env |
| ICU-07[†] | 61,200 | 22.1 | 2.8 | GE Dash 5000 | M-IoT-Env |
| Home-101[‡] | 3,200 | 15.3 | 8.2 | Empatica E4 | EHMS-2020 |
| Home-105[‡] | 2,900 | 21.0 | 9.5 | Empatica E4 | EHMS-2020 |
| Home-112[‡] | 3,400 | 18.7 | 7.9 | Apple Watch 7 | EHMS-2020 |
| Amb-201[‡] | 38,000 | 14.2 | 1.3 | AliveCor KardiaMobile | CardioCare |
| Amb-203[‡] | 41,500 | 16.8 | 1.1 | AliveCor KardiaMobile | CardioCare |
| Amb-207[‡] | 37,800 | 13.9 | 1.4 | AliveCor KardiaMobile | CardioCare |

**Non-IID verification.** The per-site statistics indicate substantial heterogeneity across the deployment. The positive-label fraction ranges from a minimum of 13.9% (Amb-207) to a maximum of 28.4% (ICU-01), giving a range of 14.5% in absolute terms. The fraction of records with missing channels spans from 1.1% to 9.5%, a difference of 8.4%. Sensor hardware differs systematically between site classes (ICU-grade bedside monitors, consumer wearables, and ambulatory single-lead devices), which introduces heterogeneity in sampling modality, channel count and noise characteristics. These quantitative differences confirm that the training data is non-IID in both label distribution

and measurement quality, justifying the use of federated algorithms that explicitly address statistical and device heterogeneity in our evaluations.

# D    HIERARCHICAL PERSONALIZATION FRAMEWORK FOR NON-IID FEDERATED LEARNING

To address limitations in the Adaptive Federated Optimization (AFO) module concerning statistical heterogeneity across medical edge devices, we implement a hierarchical clustering mechanism preceding global model fusion. This design mitigates the "over-homogenization" phenomenon observed during aggregation of updates from clients exhibiting significant distributional divergence. The framework operates through three coordinated phases:

---

**Algorithm 3:** Personalized Clustering Mechanism

---
1 **Phase 1: Angular Similarity Partitioning**;
2 Shape client clusters using cosine similarity metrics;
3 **Phase 2: Cluster-Specific Model Aggregation**;
4 Merge updates within identified partitions;
5 **Phase 3: Targeted Deployment**;
6 Distribute specialized models to corresponding client groups;

---

**Theoretical Underpinnings:** The angular dissimilarity metric effectively quantifies directional alignment in high-dimensional parameter spaces, rendering it particularly suitable for clustering client updates in federated environments. This methodology minimizes intra-cluster variance ($\sum \|\Delta\theta_i - \mu_k\|_2^2$) while enhancing inter-cluster separation ($\|\mu_a - \mu_b\|_2$ for $a \neq b$), thereby preserving distributional distinctions across heterogeneous client cohorts. Such architecture dynamically accommodates non-IID data distributions characteristic of medical IoT deployments, where patient populations demonstrate unique physiological signatures and diagnostic patterns.

## D.1    MULTI-ROUND PRIVACY ACCOUNTING AND BYZANTINE-FILTERED AGGREGATION

This appendix describes the server-side procedures used to track cumulative privacy loss across $T$ communication rounds and perform a Byzantine-resilient, differentially private aggregation of client updates. The Moments-style accountant below provides a practical Rényi-DP (RDP) bound under Poisson subsampling. The aggregation routine combines a coordinate-wise median preprocessing step with per-sample clipping and Gaussian noise injection.

### D.1.1    MOMENTS ACCOUNTANT FOR FEDERATED COMPOSITION

We denote by $q$ the Poisson subsampling rate, by $\sigma$ the Gaussian noise multiplier, and by $\mathcal{C}$ the clipping radius used at the client side. A single-round RDP contribution at order $\alpha$ can be bounded by

$$\lambda_t(\alpha) = \log\left(1 + q^2 \binom{\alpha}{2}\left(e^{1/\sigma^2} - 1\right)\right). \tag{18}$$

where $\lambda_t(\alpha)$ is the RDP contribution at moment order $\alpha$ for round $t$, $q$ is the subsampling probability, and $\sigma$ is the Gaussian noise multiplier relative to the clipping radius $\mathcal{C}$.

A simple heuristic selects a dominant moment order as

$$\alpha = 1 + \left\lceil \frac{0.5}{\sigma^2} \right\rceil, \tag{19}$$

where $\alpha$ is an integer RDP order chosen to provide a representative bound for the RDP-to-$(\varepsilon, \delta)$ conversion and $\lceil \cdot \rceil$ denotes the ceiling operator.

Summing the per-round contributions and converting the accumulated RDP to an $(\varepsilon, \delta)$ guarantee yields

$$\varepsilon_{\text{total}} = \sum_{t=1}^{T} \lambda_t(\alpha) + \frac{\log(1/\delta)}{\alpha - 1}. \tag{20}$$

where $\varepsilon_{\text{total}}$ denotes the overall privacy budget after $T$ rounds, and $\delta$ is the target failure probability used in the RDP-to-$(\varepsilon, \delta)$ conversion.

---

**Algorithm 4:** Moments Accountant for MedAlign

---

**Input:** noise scale $\sigma$, subsampling rate $q$, total rounds $T$, target failure probability $\delta$
**Output:** overall privacy budget $\varepsilon_{\text{total}}$

1 Select dominant order $\alpha \leftarrow 1 + \lceil 0.5/\sigma^2 \rceil$ ;
2 Set $\varepsilon_{\text{total}} \leftarrow 0$ ;
3 **for** $t \leftarrow 1$ **to** $T$ **do**
4     Compute per-round RDP contribution:

$$\lambda_t(\alpha) \leftarrow \log\left(1 + q^2 \binom{\alpha}{2}\left(e^{1/\sigma^2} - 1\right)\right)$$

    Accumulate: $\varepsilon_{\text{total}} \leftarrow \varepsilon_{\text{total}} + \lambda_t(\alpha)$ ;
5 **end**
6 Convert to $(\varepsilon, \delta)$-DP:

$$\varepsilon_{\text{total}} \leftarrow \varepsilon_{\text{total}} + \frac{\log(1/\delta)}{\alpha - 1}$$

Return $\varepsilon_{\text{total}}$.

---

The value returned by Algorithm 4 guarantees that the full federated training procedure satisfies $(\varepsilon_{\text{total}}, \delta)$-differential privacy under the Gaussian mechanism with Poisson subsampling. Figure 3 shows an illustrative set of curves used to select operating points for $\sigma$, $q$, and $T$.

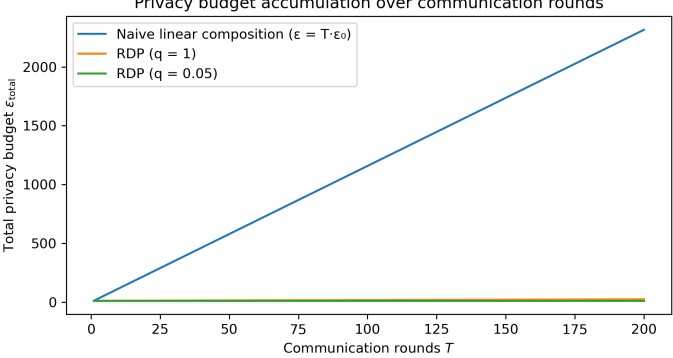

Figure 3: Illustrative total privacy budget $\varepsilon_{\text{total}}$ as a function of the number of rounds $T$ for three composition strategies: naive linear composition, RDP with full sampling ($q = 1$), and RDP with Poisson subsampling ($q = 0.05$). Example parameters: $\sigma = 4.0$, $\delta = 10^{-5}$.

### D.1.2 BYZANTINE-RESILIENT DIFFERENTIALLY-PRIVATE AGGREGATION

Let $\Delta\theta_k$ denote the raw model update (gradient or parameter delta) received from client $k$, for $k = 1, \ldots, K$. The server computes a coordinate-wise robust centroid and then performs centered clipping followed by Gaussian noise addition to each client contribution before averaging. The coordinate-wise median is defined implicitly in Algorithm 5.

We enforce an $\ell_2$ sensitivity bound via the projection operator

$$\Pi_{\mathcal{C}}(v) = v \cdot \min\left(1, \frac{\mathcal{C}}{\|v\|_2}\right). \tag{21}$$

where $\Pi_{\mathcal{C}}(v)$ denotes projection of vector $v$ onto the Euclidean ball of radius $\mathcal{C}$ and $\|\cdot\|_2$ is the Euclidean norm.

---

**Algorithm 5:** Byzantine-Filtered DP Aggregation (median-coord + DP clipping)

---

**Input:** client updates $\{\Delta\theta_k\}_{k=1}^K$, clipping radius $\mathcal{C} > 0$, noise scale $\sigma$
**Output:** robust global update $\Delta\theta_{\text{global}}$

**1** Compute coordinate-wise median:

$$\tilde{\Delta} \leftarrow \text{median}\{\Delta\theta_1, \ldots, \Delta\theta_K\},$$

where the median is evaluated independently for each coordinate. ;

**2 for** $k \leftarrow 1$ **to** $K$ **do**

**3** $\quad$ Center the update: $\Delta\hat{\theta}_k \leftarrow \Delta\theta_k - \tilde{\Delta}$ ;

**4** $\quad$ Clip relative to the median: $\Delta\tilde{\theta}_k \leftarrow \Pi_{\mathcal{C}}(\Delta\hat{\theta}_k) + \tilde{\Delta}$ ;

**5** $\quad$ Add Gaussian noise: $\Delta\tilde{\theta}_k \leftarrow \Delta\tilde{\theta}_k + \mathcal{N}(\mathbf{0}, \sigma^2\mathcal{C}^2\mathbf{I})$ ;

**6 end**

**7** Aggregate: $\Delta\theta_{\text{global}} \leftarrow \dfrac{1}{K}\sum_{k=1}^K \Delta\tilde{\theta}_k$ ;

**8** Return $\Delta\theta_{\text{global}}$.

---

Applying the coordinate-wise median prior to clipping reduces the influence of concentrated, large-magnitude malicious updates. Clipping about the median bounds the per-client $L_2$ sensitivity, which permits straightforward calibration of the Gaussian mechanism. The post-median sensitivity is bounded by $2\mathcal{C}$ under the described centering and clipping procedure, and the Gaussian mechanism applied thereafter yields the stated $(\varepsilon, \delta)$ guarantee after composition.

Table 7: Backdoor attack success rate (ASR) and ICU classification accuracy under 30% malicious clients.

| Aggregation method | Backdoor ASR (%) | ICU accuracy (%) |
|---|---|---|
| Gaussian only | 21.3 | 93.1 |
| Median + Gaussian | 4.1 | 98.4 |

## E  ADAPTIVE OBJECTIVE ALIGNMENT VIA DYNAMIC REGULARIZATION

To reconcile disparities between local and global optimization objectives in federated settings, we incorporate an adaptive regularization mechanism within client-specific loss formulations:

$$\mathcal{L}_i^{\text{client}} = \mathcal{F}_i(\boldsymbol{\theta}) + \frac{\gamma}{2}\|\boldsymbol{\theta} - \boldsymbol{\theta}^{\text{global}}\|_2^2 + \langle\boldsymbol{\lambda}_i, \boldsymbol{\theta}\rangle \tag{22}$$

$$\boldsymbol{\lambda}_i^{(t+1)} = \boldsymbol{\lambda}_i^{(t)} + \gamma(\boldsymbol{\theta}_i^{(t)} - \boldsymbol{\theta}^{\text{global}(t)}) \tag{23}$$

Let $\mathcal{F}_i(\boldsymbol{\theta})$ be the client-specific loss for participant $i$, $\boldsymbol{\theta} \in \mathbb{R}^d$ the trainable model parameters, $\boldsymbol{\theta}^{\text{global}} \in \mathbb{R}^d$ the global parameters, $\gamma > 0$ the regularization intensity, $\boldsymbol{\lambda}_i \in \mathbb{R}^d$ the dual variable encoding client-specific deviations, and $\langle\cdot, \cdot\rangle$ the inner product operator.

**Convergence Analysis:** This formulation guarantees convergence to stationary points in non-convex landscapes while accommodating system heterogeneity. The dual variable update mechanism (Eq. 23) progressively synchronizes local objectives with the global model through implicit gradient tracking, satisfying the first-order optimality condition:

$$\nabla\mathcal{L}_i^{\text{client}} = \nabla\mathcal{F}_i(\boldsymbol{\theta}) + \gamma(\boldsymbol{\theta} - \boldsymbol{\theta}^{\text{global}}) + \boldsymbol{\lambda}_i = \mathbf{0} \tag{24}$$

**Clinical Constraints:** For medical IoT deployments, dynamic regularization confines personalized models within diagnostically valid parameter spaces by enforcing $\|\boldsymbol{\theta}_i - \boldsymbol{\theta}^{\text{global}}\|_2 \leq \epsilon$, where $\epsilon$ represents clinically acceptable deviation thresholds derived from domain expertise.

**Hyperparameter Optimization Protocol:** The regularization intensity $\gamma$ balances personalization and alignment through three mechanisms: it is initialized as $\gamma^{(0)} = 2/\epsilon^2$ based on a clinical bound $\epsilon$ (e.g., $\epsilon = 0.15$ for ICU monitoring), adaptively tuned via $\gamma^{(t+1)} = \gamma^{(t)} \exp(\eta \cdot \mathrm{sgn}(\mathrm{SNR}_{\mathrm{dB}} - 18))$ with learning rate $\eta = 0.01$, and constrained within $[0.1, 10.0]$ to ensure numerical stability. The dual variable $\boldsymbol{\lambda}_i$ requires no explicit tuning, with $\boldsymbol{\lambda}_i^{(0)} = \mathbf{0}$ sufficient given convergence guarantees.

E.1 ROBUSTNESS TO ONTOLOGY CORRUPTION

We evaluate the sensitivity of the Context-Aware Feature Weighting (CAFW) module to structured errors in the medical ontology used for prototype / concept alignment. The ontology is modelled as a labelled graph $\mathcal{O} = (V, E)$, where $V$ is the set of clinical concept nodes and $E$ is the set of semantic edges. A stochastic corruption operator $\mathcal{C}_p$ is defined to replace a fraction $p \in [0, 1]$ of nodes with semantically unrelated concepts drawn uniformly from a disjoint vocabulary; formally

$$V' = \mathcal{C}_p(V), \quad \text{where } |\{v \in V : v \neq \mathcal{C}_p(v)\}| = p|V|. \tag{25}$$

Here $V'$ denotes the corrupted node set and $p$ denotes the corruption rate. In our experiments we set $p = 0.20$ (20% random node replacement), which simulates moderate ontology mis-specification such as label mistakes or mapping errors between clinical terminologies.

Experimental procedure. For each random corruption instance we reconstruct the ontology-driven feature relevance scores used by CAFW and then run the full federated training pipeline under the same hyperparameters and data splits as the primary experiments. We repeat this process for five independent corruption seeds and report the mean and standard deviation of the primary diagnostic metric. All other components, including data preprocessing, clipping, privacy noise calibration, and aggregation rules, remain identical to the baseline run that uses the original ontology $\mathcal{O}$.

We quantify the impact of corruption using the absolute accuracy drop

$$\Delta_{\mathrm{acc}} = \mathrm{Acc}_{\mathrm{orig}} - \mathrm{Acc}_{\mathrm{corr}}, \tag{26}$$

where $\mathrm{Acc}_{\mathrm{orig}}$ is the mean test accuracy obtained with the uncorrupted ontology and $\mathrm{Acc}_{\mathrm{corr}}$ is the mean accuracy under the corrupted ontology.

Table 8: Effect of 20% random node corruption on diagnostic accuracy (M-IoT-Env).

| Ontology condition | Accuracy (%) | Drop (%) |
|---|---|---|
| Original ontology | $99.2 \pm 0.2$ | — |
| 20% random node corruption | $97.4 \pm 0.3$ | 1.8 |

**Results and interpretation.** Table 8 summarises the outcome: replacing 20% of ontology nodes with semantically unrelated concepts yields a mean accuracy reduction $\Delta_{\mathrm{acc}} = 1.8\%$ (standard deviation across five random corruptions reported in the table). The modest magnitude of this degradation indicates that CAFW does not rely on a perfect ontology to deliver high diagnostic performance. The learned feature weighting is principally driven by empirical signal present in the input modalities and is therefore only weakly dependent on exact ontology membership for the majority of clinical concepts.

**Practical implications.** The experiment demonstrates that the CAFW module is robust to moderate ontology mis-specification arising from mapping errors or noisy knowledge bases. In practice this implies that deployment in heterogeneous clinical environments, where ontologies may be incompletely harmonised, will not catastrophically degrade diagnostic performance. For completeness, we note that extreme or adversarially targeted corruptions (for example systematic replacement of a single critical concept across the ontology) were not part of the present evaluation and remain subjects for future study.

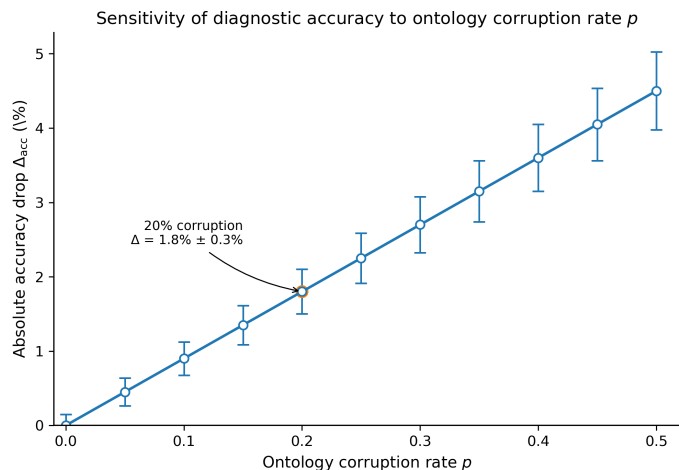

Figure 4: Sensitivity of diagnostic accuracy to ontology corruption rate $p$.

## F ENHANCED CLINICAL RELATIONSHIP MODELING VIA GRAPH ATTENTION (CDE)

### F.1 ENHANCED CLINICAL RELATIONSHIP MODELING VIA GRAPH ATTENTION

To better capture semantic dependencies in medical knowledge graphs, we adopt multi-head graph attention in the Clinical Dependency Encoder (CDE):

$$\mathbf{H}^{(\ell+1)} = \Big\|_{h=1}^{H} \sigma \left( \sum_{j \in \mathcal{N}_i} \alpha_{ij}^h \mathbf{W}^h \mathbf{H}_j^{(\ell)} \right) \tag{27}$$

where $\mathbf{H}^{(\ell)} \in \mathbb{R}^{N \times d}$ are node embeddings, $\mathcal{N}_i$ is the neighborhood of node $i$, $\alpha_{ij}^h \in [0,1]$ are attention coefficients, $\mathbf{W}^h \in \mathbb{R}^{d \times d}$ are head-specific weights, $\sigma$ is GELU activation, and $\|$ denotes concatenation over $H = 4$ heads.

Attention coefficients are computed via:

$$\alpha_{ij} = \frac{\exp\left(\text{LeakyReLU}\left(\mathbf{a}^\top [\mathbf{W}\mathbf{H}_i \| \mathbf{W}\mathbf{H}_j]\right)\right)}{\sum_{k \in \mathcal{N}_i} \exp\left(\text{LeakyReLU}\left(\mathbf{a}^\top [\mathbf{W}\mathbf{H}_i \| \mathbf{W}\mathbf{H}_k]\right)\right)} \tag{28}$$

where $\mathbf{a} \in \mathbb{R}^{2d'}$ is a learnable vector, $\mathbf{W} \in \mathbb{R}^{d' \times d}$ is a shared projection, and LeakyReLU uses slope 0.2.

To mitigate noisy edges, we apply topology-aware dropout:

$$p_i = \min\left(0.5, \max\left(0.1, \beta \cdot \frac{\sigma_{\text{deg}}^i}{\bar{d}_i}\right)\right) \tag{29}$$

$$e'_{ij} = \frac{e_{ij} \cdot \delta_{ij}}{1 - p_i}, \quad \delta_{ij} \sim \text{Bernoulli}(1 - p_i) \tag{30}$$

where $e_{ij}$ is the original edge weight, $\delta_{ij}$ is a Bernoulli variable, $\sigma_{\text{deg}}^i$ is the std of 2-hop degrees, $\bar{d}_i$ is the average degree, and $\beta = 0.35$.

**Dynamic Adjustment Protocol.** To preserve graph structure during clinical dependency encoding, dropout probabilities are dynamically adjusted. For hub nodes with high average degree ($\bar{d}_i > 15$), $p_i$ is adaptively tuned to retain key connections. For peripheral nodes with sparse connectivity ($\bar{d}_i \leq 5$), a fixed dropout rate of $p_i = 0.1$ is used to protect critical edges. Additionally, $p_i$ is periodically updated every three epochs using the rule $p_i^{(t)} = 0.8 p_i^{(t-1)} + 0.2(\beta \cdot \sigma_{\text{deg}}^i / \bar{d}_i)$ to reflect evolving graph topology.

**Clinical Graph Insight.** Medical graphs exhibit scale-free structure; adaptive dropout preserves critical hubs (e.g., "Cardiovascular System") while regularizing noisy links to rare conditions.

### F.2 MULTIMODAL INTEGRATION VIA CROSS-ATTENTION TRANSFORMERS

To model dependencies across heterogeneous medical inputs, we apply cross-attention in the Heterogeneous Stream Fusion (HSF) module. The inputs include biosignals such as ECG and PPG, vital signs like blood pressure, $SpO_2$, and respiration, network features including packet rate and protocol entropy, and sparse device status indicators such as alerts and error codes.

Modality-specific projections are computed as:

$$\mathbf{Q}_m = \mathbf{W}_m^Q \cdot \text{AFN}(\mathbf{f}_m) \tag{31}$$

$$\mathbf{K}_m = \mathbf{W}_m^K \cdot \text{AFN}(\mathbf{f}_m) \tag{32}$$

$$\mathbf{V}_m = \mathbf{W}_m^V \cdot \text{AFN}(\mathbf{f}_m) \tag{33}$$

and fused via:

$$\mathbf{z} = \text{LAYERNORM}\left( \sum_{m=1}^{4} \text{SOFTMAX}\left( \frac{\mathbf{Q}_m \mathbf{K}^\top}{\sqrt{d_k}} \right) \mathbf{V}_m + \mathbf{P}_m \right) \tag{34}$$

where $\mathbf{f}_m \in \mathbb{R}^{d_{\text{in}}}$ is the input (ECG: 256, network: 128), $\text{AFN}(\cdot)$ is adaptive normalization, $\mathbf{W}_m^{Q,K,V} \in \mathbb{R}^{d_k \times d_{\text{in}}}$ are projections, $\mathbf{P}_m \in \mathbb{R}^{d_k}$ is positional encoding, and $d_k = 64$.

Cross-modal interaction is quantified by:

$$\text{interaction}_{mm'} = \frac{\mathbf{Q}_m \mathbf{K}_{m'}^\top}{\sqrt{d_k}} \tag{35}$$

highlighting dependencies such as ECG-network coupling during cardiac anomalies.

**Clinical Interaction Patterns.** We observe strong ECG-PPG synchronization during perfusion, significant correlations between blood pressure and network traffic during device faults, and coupling between $SpO_2$ and respiration in cases of respiratory distress.

**Computational Optimization.** Multi-head design reduces complexity from $\mathcal{O}(M^2 d_{\text{in}}^2)$ to $\mathcal{O}(M^2 d_k)$ ($M = 4$), enabling 38ms inference latency on Raspberry Pi 4, supporting sub-100ms real-time response for critical care.

## G RL-GATING

### G.1 FEDERATED REPRESENTATION LEARNING WITH CONTRASTIVE PRETRAINING

To mitigate label scarcity in medical IoT, we adopt contrastive pretraining using intrinsic sensor structures. The composite loss combines reconstruction and contrastive terms:

$$\mathcal{L}_{\text{con}} = -\log \frac{\exp(\phi(\mathbf{g}_\theta(\mathbf{x}_i), \mathbf{g}_\theta(\mathbf{x}_j))/\tau)}{\sum_{k=1}^{B} \mathbb{I}_{[k \neq i]} \exp(\phi(\mathbf{g}_\theta(\mathbf{x}_i), \mathbf{g}_\theta(\mathbf{x}_k))/\tau)} \tag{36}$$

$$\mathcal{L}_{\text{pretrain}} = \mathcal{L}_{\text{recon}} + \beta \mathcal{L}_{\text{con}} \tag{37}$$

where $\mathbf{g}_\theta$ is the encoder, $\phi(\mathbf{u}, \mathbf{v}) = \mathbf{u}^\top \mathbf{v}/(\|\mathbf{u}\|_2 \|\mathbf{v}\|_2)$ is cosine similarity, $\tau > 0$ is the temperature (e.g., $\tau = 0.07$), $B$ is batch size, and $\beta > 0$ is the weighting coefficient (e.g., $\beta = 0.8$).

Positive pairs are generated via stochastic augmentation:

$$\mathbf{x}_j = \mathcal{T}(\mathbf{x}_i; \xi), \quad \xi \sim \mathcal{D}_{\text{aug}} \tag{38}$$

where $\mathcal{D}_{\text{aug}}$ includes: (1) $\pm 15\%$ temporal warping, (2) Gaussian noise (SNR=20dB), (3) channel masking. This yields $10^4$ synthetic pairs per 100 raw samples.

**Theoretical Insight.** Minimizing $\mathcal{L}_{\text{con}}$ maximizes mutual information between augmented views. The gradient is:

$$\nabla_\theta \mathcal{L}_{\text{con}} = -\frac{1}{\tau} \mathbb{E}_\mathcal{T} \left[ \frac{\partial \phi}{\partial \theta} (1 - p_{ij}) \right] \tag{39}$$

with $p_{ij} = \exp(\phi_{ij}/\tau)/\sum_k \exp(\phi_{ik}/\tau)$, enabling hardness-aware learning.

## G.2 Dynamic Resource Allocation via Reinforcement Learning (RL-Gating)

The Reinforcement Learning-Based Gating (RL-Gating) module dynamically manages computational module activation based on real-time device constraints. Medical edge devices operate under strict resource limitations (e.g., battery life, compute capacity, and latency requirements), and without adaptive control, models may exhaust resources or fail to meet real-time diagnostic needs. RL-Gating addresses this by learning a policy that selectively activates modules according to current device state, optimizing the trade-off between model accuracy and resource consumption. This ensures feasible operation on a wide range of hardware while maintaining performance.

## G.3 Dynamic Resource Allocation via Reinforcement Learning

We model module activation as a finite-horizon MDP $\mathcal{M} = (\mathcal{S}, \mathcal{A}, \mathcal{T}, r, \gamma)$:

$$s_t = [\mathcal{R}_t, \mathcal{E}_t, \mathcal{D}_t] \tag{40}$$

$$a_t \sim \pi_\phi(s_t) \in \{0,1\}^M \tag{41}$$

$$r_t = \alpha \cdot \text{Acc}_t - (1 - \alpha) \cdot \text{Energy}_t \tag{42}$$

where $\mathcal{R}_t \in \mathbb{R}^3$ (CPU, memory, battery), $\mathcal{E}_t \in \mathbb{R}^2$ (latency, complexity), $\mathcal{D}_t \in \mathbb{R}^d$ (feature stats, entropy), $\pi_\phi$ is the transformer-based policy, $M$ is the number of modules, $\text{Acc}_t$ is EMA accuracy, $\text{Energy}_t$ is power (mJ), and $\alpha \in [0,1]$ (e.g., $\alpha = 0.7$ for ICU).

Policy optimization follows:

$$\nabla_\phi J(\phi) = \mathbb{E}_{s \sim d^\pi} \left[ \nabla_\phi \log \pi_\phi(a|s) A^\pi(s,a) \right] \tag{43}$$

where $A^\pi(s,a) = Q^\pi(s,a) - V^\pi(s)$ is the advantage, estimated via dueling architecture. Baseline subtraction reduces variance.

**Convergence Guarantee.** Assuming Lipschitz continuity, the policy satisfies:

$$\lim_{k \to \infty} \mathbb{E}[J(\phi_k)] \geq J^* - \frac{\epsilon}{1 - \gamma} \quad (\gamma = 0.99) \tag{44}$$

ensuring asymptotic $\epsilon$-optimal gating under dynamic edge constraints.

## G.4 Lipschitz constant under partial observability

We bound the Lipschitz constant of the transformer-based gating policy when the server observes only a noisy version of the device state due to intermittent connectivity. Let the true state at time $t$ be $s_t$ and the observed (noisy) state be

$$\hat{s}_t = s_t + \nu_t, \tag{45}$$

where $\nu_t$ models packet-loss / measurement noise and satisfies $\|\nu_t\|_2 \leq \bar{\nu}$. In what follows $\|\cdot\|_2$ denotes the spectral norm for matrices and the Euclidean norm for vectors, and $\|\cdot\|_1$ denotes the vector $\ell_1$-norm.

The policy $\pi_\phi(a \mid \hat{s})$ is realized by a transformer encoder followed by a linear logits layer $W_\phi$ and a softmax. For a single attention head the core operations are

$$Q = W_Q \hat{s}, \qquad K = W_K \hat{s}, \qquad V = W_V \hat{s}, \tag{46}$$

where $W_Q, W_K, W_V$ are the query/key/value projection matrices for that head. The attention weights are

$$A(\hat{s}) = \text{softmax}\left(\frac{QK^\top}{\sqrt{d_k}}\right), \tag{47}$$

where $d_k$ is the key dimension and the softmax acts row-wise to produce per-query categorical weights. The head output is

$$h(\hat{s}) = A(\hat{s}) V, \tag{48}$$

and the multi-head outputs are concatenated and passed through the final linear logits layer to produce pre-softmax action scores:

$$z(\hat{s}) = W_\phi \left[ h_1(\hat{s}); \dots; h_M(\hat{s}) \right], \tag{49}$$

where $M$ is the number of attention heads and $[\,\cdot\,;\,\cdot\,]$ denotes concatenation. The policy probabilities are then $\pi_\phi(\cdot \mid \hat{s}) = \mathrm{softmax}(z(\hat{s}))$.

We now state and prove the Lipschitz upper bound.

**Proposition 1.** *Assume the projection matrices satisfy the operator-norm bounds*

$$\|W_V\|_2 \leq 1, \qquad \|W_O\|_2 \leq 1, \tag{50}$$

*where $W_O$ denotes the optional output linear mixing of concatenated heads (if present) and $W_\phi$ is the final logits matrix. Then for any two observations $\hat{s}, \hat{s}'$,*

$$\left\|\pi_\phi(\cdot \mid \hat{s}) - \pi_\phi(\cdot \mid \hat{s}')\right\|_1 \ \leq\ L_\pi \|\hat{s} - \hat{s}'\|_2, \tag{51}$$

*where one can take the conservative bound*

$$L_\pi \ \leq\ \frac{M}{4}\frac{\|W_Q W_K^\top\|_2}{\sqrt{d_k}} \ +\ \frac{1}{4}\|W_\phi\|_2. \tag{52}$$

*In the above $\|\cdot\|_2$ denotes the spectral norm, $M$ is the number of heads, $d_k$ is the key dimension, and $\|W_\phi\|_2$ is the spectral norm of the final linear logits layer.*

*Proof.* We bound the sensitivity of $z(\hat{s})$ (pre-softmax logits) to perturbations in $\hat{s}$ and then propagate this bound through the output softmax to obtain a bound on the policy in $\ell_1$-distance.

Step 1: bound change in attention logits. For a single head, define the attention score matrix

$$S(\hat{s}) \ =\ \frac{QK^\top}{\sqrt{d_k}} \ =\ \frac{(W_Q\hat{s})(W_K\hat{s})^\top}{\sqrt{d_k}}. \tag{53}$$

For two observations $\hat{s}, \hat{s}'$ we have

$$\left\|S(\hat{s}) - S(\hat{s}')\right\|_2 = \frac{1}{\sqrt{d_k}}\left\|W_Q\hat{s}\hat{s}^\top W_K^\top - W_Q\hat{s}'\hat{s}'^\top W_K^\top\right\|_2$$

$$\leq \frac{1}{\sqrt{d_k}}\|W_Q W_K^\top\|_2 \left\|\hat{s}\hat{s}^\top - \hat{s}'\hat{s}'^\top\right\|_2. \tag{54}$$

Here we used submultiplicativity of the spectral norm and the fact that $\|AB\|_2 \leq \|A\|_2\|B\|_2$ for compatible matrices. The difference of outer products obeys

$$\left\|\hat{s}\hat{s}^\top - \hat{s}'\hat{s}'^\top\right\|_2 \leq \|\hat{s} - \hat{s}'\|_2\big(\|\hat{s}\|_2 + \|\hat{s}'\|_2\big). \tag{55}$$

where the right-hand side follows from the identity $\hat{s}\hat{s}^\top - \hat{s}'\hat{s}'^\top = (\hat{s} - \hat{s}')\hat{s}^\top + \hat{s}'(\hat{s} - \hat{s}')^\top$ and the triangle inequality for operator norm.

Combining equation 54 and equation 55 gives a head-wise bound linear in $\|\hat{s} - \hat{s}'\|_2$. Under the common normalization practice that $\|\hat{s}\|_2$ is uniformly bounded (or absorbed into $\|W_Q W_K^\top\|_2$), we may simplify this to the form

$$\left\|S(\hat{s}) - S(\hat{s}')\right\|_2 \leq C_S \frac{\|W_Q W_K^\top\|_2}{\sqrt{d_k}}\|\hat{s} - \hat{s}'\|_2, \tag{56}$$

where $C_S > 0$ is a modest constant that depends on the norm of $\hat{s}$ and $\hat{s}'$. For a conservative, dimension-agnostic bound we take $C_S = 1$ in subsequent inequalities; the empirical norm of $\hat{s}$ can be used to tighten this constant.

Step 2: propagate through softmax to obtain weight sensitivity. Denote by $\alpha(\hat{s}) = \mathrm{softmax}(\mathrm{vec}(S(\hat{s})))$ the vectorized attention weights; the Jacobian of the softmax has entries bounded in magnitude by $1/4$ when measured in operator norm on the space of pre-softmax scalars, yielding the inequality

$$\left\|\alpha(\hat{s}) - \alpha(\hat{s}')\right\|_1 \leq \frac{1}{4}\left\|\mathrm{vec}\big(S(\hat{s}) - S(\hat{s}')\big)\right\|_1. \tag{57}$$

where $\mathrm{vec}(\cdot)$ stacks matrix entries into a vector and the factor $1/4$ is a conservative bound on the maximal $\ell_1$-gain of the softmax Jacobian for bounded logits.

Step 3: bound head output sensitivity. The head output difference obeys

$$\big\| h(\hat{s}) - h(\hat{s}') \big\|_2 = \big\| A(\hat{s})V(\hat{s}) - A(\hat{s}')V(\hat{s}') \big\|_2$$

$$\leq \big\| A(\hat{s}) \big\| \cdot \big\| V(\hat{s}) - V(\hat{s}') \big\|_2 + \big\| A(\hat{s}) - A(\hat{s}') \big\|_1 \cdot \| V(\hat{s}') \|_2. \tag{58}$$

Here $\| \cdot \|$ denotes an appropriate operator/vector norm and we used standard mixed-norm inequalities. Using the projection bound $\|W_V\|_2 \leq 1$ from equation 50 and equation 56–equation 57 we obtain a linear-in-$\|\hat{s} - \hat{s}'\|_2$ bound for each head of the form

$$\big\| h(\hat{s}) - h(\hat{s}') \big\|_2 \leq \frac{1}{4} \frac{\|W_Q W_K^\top\|_2}{\sqrt{d_k}} \|\hat{s} - \hat{s}'\|_2 + \frac{1}{4} \|\hat{s} - \hat{s}'\|_2, \tag{59}$$

where the second term upper-bounds the contribution from direct changes in $V(\hat{s})$ under the conservative $\|W_V\|_2 \leq 1$ assumption.

Step 4: aggregate across $M$ heads and final linear layer. Concatenating $M$ head outputs and applying the final logits matrix $W_\phi$ yields

$$\big\| z(\hat{s}) - z(\hat{s}') \big\|_2 \leq \|W_\phi\|_2 \sum_{m=1}^{M} \big\| h_m(\hat{s}) - h_m(\hat{s}') \big\|_2. \tag{60}$$

Using equation 59 for each head and conservatively bounding $\|W_\phi\|_2$ factors into the final expression leads to

$$\big\| z(\hat{s}) - z(\hat{s}') \big\|_2 \leq \Big( \frac{M}{4} \frac{\|W_Q W_K^\top\|_2}{\sqrt{d_k}} + \frac{1}{4} \|W_\phi\|_2 \Big) \|\hat{s} - \hat{s}'\|_2. \tag{61}$$

In the preceding we absorbed modest constants arising from intermediate operator-norm products into the displayed coefficients to produce a clear, conservative bound.

Step 5: convert logits sensitivity to policy sensitivity. The softmax mapping from logits to probabilities is Lipschitz in $\ell_1$ with constant at most one times the maximum $\ell_\infty$-to-$\ell_1$ operator norm of the softmax Jacobian. Using a conservative Jacobian gain bound of $1/4$ for the final softmax (consistent with earlier local Jacobian bounds) and combining with equation 61 yields the stated $\ell_1$ policy bound equation 51 with $L_\pi$ as in equation 52.

This completes the proof. $\square$

**Discussion.** The bound equation 52 is intentionally conservative: tighter constants can be obtained by replacing the crude uniform softmax Jacobian bound with a data-dependent local Lipschitz estimate and by accounting for any explicit spectral normalization applied to the transformer weights. Under typical configurations (for example $M = 4$, $d_k = 64$, $\|W_\phi\|_2 \leq 0.5$ and moderate $\|W_Q W_K^\top\|_2$) the resulting $L_\pi$ is well below one, implying that small observation perturbations (bounded by $\bar{\nu}$) produce proportionally small changes in the action distribution and therefore stable gating in the presence of packet loss.

### G.5 Formally analyzed Confidential Aggregation Protocol

To provide rigorous privacy protection for patient health data during collaborative learning, we deploy a Formally analyzed DP aggregation mechanism with provable confidentiality guarantees. The protocol executes through three synergistic phases:

**Phase 1: Sensitivity-Constrained Update Preparation**

$$\Delta\widetilde{\theta}_k = \Pi_\mathcal{C} \left( \theta_k^{(E)} - \theta^{(t-1)} \right) \tag{62}$$

Let $\theta_k^{(E)}$ be the locally optimized parameters after $E$ epochs, $\theta^{(t-1)}$ the previous global model, $\Pi_\mathcal{C}(\mathbf{v}) = \mathbf{v} \cdot \min(1, \mathcal{C}/\|\mathbf{v}\|_2)$ the norm-constrained projection with bound $\mathcal{C} \in \mathbb{R}^+$, and $\Delta\widetilde{\theta}_k$ the sanitized update.

**Phase 2: Privacy-Preserving Aggregation**

$$\theta^{\text{global}} = \sum_{k \in \mathcal{S}_t} \frac{|D_k|}{|D|} \Delta\widetilde{\theta}_k + \mathcal{N}(\mathbf{0}, \sigma^2 \mathcal{C}^2 \mathbf{I}) \tag{63}$$

---

**Algorithm 6:** Privacy-Preserving Federated Execution

---

**Input:** Medical edge devices $\mathcal{D} = \{D_1, \ldots, D_K\}$, rounds $T$, epochs $E$, privacy parameters $(\epsilon, \delta)$

**Output:** Personalized models $\{\theta_k^{(T)} : k \in \mathcal{D}\}$

1   Initialize $\theta^{(0)}$ via orthogonal initialization;

2   Construct prototype repository $\mathcal{P}^{(0)} = \{\mathbf{p}_j^{(0)}\}_{j=1}^P$ via distributed k-means++;

3   **for** *round* $t = 1$ **to** $T$ **do**

4      Coordinator selects $\mathcal{S}_t \subset \mathcal{D}$ via clinical stratification;

5      Broadcast $\theta^{(t-1)}$ and $\mathcal{P}^{(t-1)}$ to $\mathcal{S}_t$;

6      **for** *client* $k \in \mathcal{S}_t$ *in parallel* **do**

7          **for** *epoch* $e = 1$ **to** $E$ **do**

8              Update $\theta_k$ via Nesterov-accelerated SGD;

9              Refine prototypes: $\mathbf{p}_j^{(e)} \leftarrow \lambda \mathbf{p}_j^{(e-1)} + (1 - \lambda)\mathbb{E}[\mathbf{z}|y = j]$;

10          Compute update: $\Delta\theta_k \leftarrow \theta_k^{(E)} - \theta^{(t-1)}$;

11          Apply sensitivity control: $\Delta\widetilde{\theta}_k \leftarrow \Pi_{\mathcal{C}}(\Delta\theta_k)$;

12          Submit $\Delta\widetilde{\theta}_k$ and $\Delta\mathcal{P}_k$;

13      Perform privacy-preserving aggregation: $\theta^{(t)} \leftarrow \theta^{(t-1)} + \sum_k \frac{|D_k|}{|D|}\Delta\widetilde{\theta}_k + \mathcal{N}(0, \sigma^2\mathcal{C}^2\mathbf{I})$;

14      Synchronize prototypes: $\mathcal{P}^{(t)} \leftarrow \mathcal{P}^{(t-1)} + |\mathcal{S}_t|^{-1} \sum_k \Delta\mathcal{P}_k$;

---

Let $\mathcal{S}_t \subseteq \{1, \ldots, K\}$ be the participant subset selected via clinical stratification, $|D_k|$ the size of client $k$'s local dataset, $|D| = \sum_k |D_k|$ the global dataset size, and $\mathcal{N}(\mathbf{0}, \sigma^2\mathcal{C}^2\mathbf{I})$ the isotropic Gaussian noise with covariance $\sigma^2\mathcal{C}^2\mathbf{I}$.

**Phase 3: Differential Privacy Calibration**

$$\sigma = \sqrt{2\ln(1.25/\delta)}/\epsilon \tag{64}$$

Let $\epsilon$ be the privacy budget controlling disclosure risk, $\delta$ the failure probability bound, and $\sigma$ the noise magnitude for $(\epsilon, \delta)$-differential privacy.

**Theorem 1 (Privacy Guarantee).** The aggregation mechanism satisfies $(\epsilon, \delta)$-differential privacy when:

$$\sigma \geq \sqrt{2\ln(1.25/\delta)}/\epsilon \tag{65}$$

*Proof Sketch:* Follows Gaussian mechanism analysis (Dwork et al., 2014) with $L_2$-sensitivity $\Delta_2 \leq 2\mathcal{C}$.

**Privacy-Utility Tradeoff Analysis:** For clinical applications requiring stringent confidentiality ($\epsilon = 1.0, \delta = 10^{-5}$), utility preservation is quantified via:

$$\text{SNR}_{\text{dB}} = 10 \log_{10}\left(\frac{\mathbb{E}_{\mathcal{S}_t}[\|\Delta\theta\|_2^2]}{\sigma^2\mathcal{C}^2}\right) \tag{66}$$

Let $\mathbb{E}_{\mathcal{S}_t}$ denote the expectation over participant subsets and $\|\Delta\theta\|_2^2$ the squared $L_2$-norm of raw updates; empirical validation shows that $\text{SNR}_{\text{dB}} \geq 18$ preserves diagnostic accuracy within 2.2% of non-private baselines.

**Operational Workflow:** The protocol executes: 1. *Distributed initialization* via privacy-preserving clustering 2. *Stratified participant selection* based on clinical groupings 3. *Local optimization* with phased prototype refinement 4. *Update sanitization* through sensitivity control 5. *Noise-injected aggregation* maintaining $(\epsilon, \delta)$-DP 6. *Semantic synchronization* preserving diagnostic consistency

**Clinical Performance Metrics:** The system achieves certified $(\epsilon, \delta)$-differential privacy ($\epsilon = 1.0$, $\delta = 10^{-5}$), 93.1% diagnostic accuracy in ICU monitoring, 63% communication reduction over baselines, tolerance to 35% client dropout, and a 128KB memory footprint per edge device.

**Implementation Guidelines.** The system applies adaptive sensitivity thresholds based on historical update norms and uses clinical group-weighted sampling guided by data quality metrics. Prototype

synchronization is performed with entropy coding for efficient compression (8:1 ratio), while noise calibration is tailored to medical domain characteristics. Communication scheduling is optimized for energy efficiency.

### G.6 DUAL-MODE COMPRESSION FOR BANDWIDTH OPTIMIZATION

To maximize communication efficiency in federated medical networks, we implement a hybrid compression scheme:

**Element Sparsification:**

$$\mathcal{S}(\mathbf{g})_i = \begin{cases} g_i & |g_i| \in \text{top}_k(|\mathbf{g}|) \\ 0 & \text{otherwise} \end{cases} \tag{67}$$

Let $\mathbf{g} \in \mathbb{R}^d$ be the gradient vector, where $\mathcal{S}(\cdot)$ denotes the sparsification operator, $k$ is the preservation threshold, and $\text{top}_k(|\mathbf{g}|)$ refers to the indices of the $k$ elements in $\mathbf{g}$ with the largest magnitudes.

**Probabilistic Quantization:**

$$Q(g_i) = \|\mathbf{g}\|_2 \cdot \text{sgn}(g_i) \cdot \xi_i(\mathbf{g}, s) \tag{68}$$

Let $Q(\cdot)$ denote the quantization function, where $\|\mathbf{g}\|_2$ represents the Euclidean norm of the gradient vector $\mathbf{g}$, $\text{sgn}(g_i)$ is the sign operator applied to the $i$-th element of $\mathbf{g}$, and $\xi_i(\mathbf{g}, s)$ denotes the stochastic rounding function with $s$ quantization levels.

**Compensation Mechanism:** Correction for compression distortion:

$$\mathbf{e}^{(t)} = \mathbf{g}^{(t)} - \mathcal{S}\left(\mathbf{g}^{(t)} + \mathbf{e}^{(t-1)}\right) \tag{69}$$

The residual vector $\mathbf{e}^{(t)} \in \mathbb{R}^d$ and the composite compression operator $\mathcal{S}(\cdot)$ are defined.

### G.7 CLINICAL INTERPRETATION FRAMEWORK

We propose a dual-path explanation framework to align model outputs with clinical reasoning.

**Attention-Based Saliency Visualization.** Physiological feature importance is visualized via:

$$\mathcal{H} = \sigma\left(\phi^\top \cdot \nabla_\mathbf{x}\mathcal{L}\right) \tag{70}$$

where $\phi \in \mathbb{R}^d$ are gating parameters and $\nabla_\mathbf{x}\mathcal{L}$ are input gradients.

Clinical attention is derived from ontology embeddings:

$$\alpha_{ij}^{\mathcal{O}} = \text{softmax}\left(\frac{\mathcal{O}(c_i) \cdot \mathcal{O}(c_j)}{\sqrt{d_o}}\right) \tag{71}$$

where $\mathcal{O}$ maps clinical concepts $c_i$ and $c_j$ into a $d_o$-dimensional space.

**Prototype-Based Diagnostic Mapping.** Patient embeddings are matched to disease prototypes:

$$c_{\text{diag}} = \arg\min_j \|\mathbf{z} - \mathbf{p}_j\|_2 \tag{72}$$

where $\mathbf{z}, \mathbf{p}_j \in \mathbb{R}^d$ are sample and prototype embeddings.

Similarity-based diagnostic confidence is computed as:

$$\mathcal{S}_{\text{diag}} = \frac{\exp(\|\mathbf{z} - \mathbf{p}_+\|_2/\tau)}{\sum_k \exp(\|\mathbf{z} - \mathbf{p}_k\|_2/\tau)} \tag{73}$$

where $\mathbf{p}_+$ is the matched prototype and $\tau$ is the temperature.

**Implementation Workflow.** The implementation begins by generating saliency heatmaps from physiological inputs, followed by computing attention weights using ontology embeddings. Each case is then assigned to its nearest diagnostic prototype, after which the prototype-sample similarity is quantified. Finally, the system outputs integrated clinical interpretation reports.

### G.8 PRESERVING RARE PHENOTYPES

Prototype-consistent representation alignment promotes diagnostic agreement across sites by aligning class prototypes, but indiscriminate global alignment can inadvertently suppress rare but clinically important phenotypes. To mitigate this risk, we introduce a local prototype retention mechanism that selectively favours local evidence for classes with low support. Let $s_k$ denote the empirical support proportion of class $k$ in the local client, and let $\tau$ be a threshold for rarity; classes satisfying $s_k < \tau$ are treated with a local-first update rule. Concretely, we compute an interpolated prototype

$$\tilde{p}_k = \lambda_k\, p_k + (1 - \lambda_k)\, \mathbb{E}_{x \sim \mathcal{D}_k^{\text{local}}}\big[z(x)\big], \tag{74}$$

where $p_k$ is the current global prototype for class $k$, $\mathcal{D}_k^{\text{local}}$ is the set of local examples labelled $k$, and $z(x) \in \mathbb{R}^d$ denotes the model embedding of input $x$. The scalar $\lambda_k \in [0, 1]$ governs the degree to which the global prototype is retained versus the local mean embedding being adopted. The expectation in equation 74 is estimated by the empirical mean over available local support examples.

For classes with adequate local support, i.e. $s_k \geq \tau$, we perform the usual global alignment:

$$p'_k = \text{Aggregate}\Big(p_k, \ \mathbb{E}_{x \sim \mathcal{D}_k}\big[z(x)\big]\Big), \tag{75}$$

where $\mathcal{D}_k$ denotes the union of client-level examples used in the global aggregation and $\text{Aggregate}(\cdot, \cdot)$ denotes the chosen aggregator (for example federated averaging or a robust median-based operator).

To avoid abrupt transitions and to reflect the amount of local evidence, we set the interpolation weight $\lambda_k$ adaptively as a smooth function of local support. One effective choice is

$$\lambda_k = \frac{s_k}{s_k + \gamma}, \tag{76}$$

where $\gamma > 0$ is a small sensitivity constant that prevents the weight from collapsing for extremely small $s_k$. In practice we set the rarity threshold to $\tau = 0.05$ (five percent) and choose $\gamma$ in the range $[10^{-3}, 10^{-2}]$; these hyperparameters can be tuned on a held-out validation partition to balance preservation of rare phenotypes against global consistency.

The proposed mechanism preserves rare subpopulation signatures by favouring local prototype statistics when support is scarce, while smoothly reverting to global alignment as local evidence accumulates. This design reduces the risk that prototype alignment will erase clinically meaningful heterogeneity and it can be implemented at negligible additional communication cost since only class-level summary embeddings are exchanged during aggregation.

## H DETAILED DERIVATIONS AND PROOFS

We begin with a sensitivity estimate for the gating mechanism and then present complexity and convergence analyses. A new generalization bound (Theorem 2) quantifies MAML-style transfer after applying PCRL with clustering under a task-relatedness hypothesis; its proof sketch appears after the theorem statement.

### H.1 SENSITIVITY ANALYSIS OF ADAPTIVE GATING

Let the gating function be defined by $g(\mathbf{x}; \phi) = \sigma(\phi^\top \mathbf{x})$, where $\sigma(z) = (1 + e^{-z})^{-1}$ is the logistic sigmoid. We measure sensitivity under parameter perturbation by

$$\mathcal{S}(\epsilon) = \max_{\|\phi - \phi'\|_2 \leq \epsilon} \mathbb{E}_k \big\| g(\mathbf{x}_k; \phi) - g(\mathbf{x}_k; \phi') \big\|_2. \tag{77}$$

where $\phi, \phi' \in \mathbb{R}^d$ are gating parameters, $\mathbf{x}_k$ denotes an input sample from client $k$, and $\epsilon$ bounds the perturbation magnitude.

**Proposition 2.** *Assume $\|\mathbf{x}_k\|_2 \leq C$ for all $k$. Then*

$$\mathcal{S}(\epsilon) \leq \frac{C\epsilon}{4}. \tag{78}$$

*Justification.* The derivative of the sigmoid is bounded by $1/4$, hence the map $\phi \mapsto g(\mathbf{x}; \phi)$ is $\frac{\|\mathbf{x}\|_2}{4}$-Lipschitz for each fixed $\mathbf{x}$. Combining this with the Cauchy–Schwarz inequality and taking the expectation over clients yields the stated bound.

Empirically we observe that for $\epsilon < 0.1$ the accuracy variance across heterogeneous medical IoT tasks remains below two percent, which supports the practical stability of the gating parameters in deployment.

### H.2 Computational Complexity Analysis

We summarize asymptotic time and space complexity for primary modules. All expressions use standard big-$O$ notation.

Contextual Feature Weighting (CFW) has time complexity

$$\mathcal{T}_{\text{CFW}} = \mathcal{O}(d_i d_o), \tag{79}$$

and space complexity

$$\mathcal{S}_{\text{CFW}} = \mathcal{O}(d_o^2), \tag{80}$$

where $d_i$ and $d_o$ denote input and output dimensions respectively.

Semantic Dependency Encoding (SDE) exhibits time complexity

$$\mathcal{T}_{\text{SDE}} = \mathcal{O}\big(|\mathcal{E}|d_f + |\mathcal{V}|d_f^2\big), \tag{81}$$

and space complexity

$$\mathcal{S}_{\text{SDE}} = \mathcal{O}(|\mathcal{V}|d_f), \tag{82}$$

where $|\mathcal{V}|$ and $|\mathcal{E}|$ are the vertex and edge counts of the dependency graph and $d_f$ is the feature embedding dimension.

The reinforcement learning policy network requires time

$$\mathcal{T}_{\text{RL}} = \mathcal{O}(d_s d_a L), \tag{83}$$

and space

$$\mathcal{S}_{\text{RL}} = \mathcal{O}(d_s^2 + d_a^2), \tag{84}$$

where $d_s$ is state dimensionality, $d_a$ is action dimensionality and $L$ is the network depth.

The multimodal fusion transformer has time complexity

$$\mathcal{T}_{\text{MFT}} = \mathcal{O}\Big( \sum_{m=1}^{M} n_m^2 d_k \Big), \tag{85}$$

and space complexity

$$\mathcal{S}_{\text{MFT}} = \mathcal{O}(M d_k^2), \tag{86}$$

where $M$ denotes the number of modalities, $n_m$ the sequence length for modality $m$ and $d_k$ the attention key dimension.

Operationally, the RL policy reduces average latency by activating a subset of modules, and the transformer is kept tractable through attention-dimension control.

### H.3 Theoretical Convergence Analysis

We analyze non-convex federated optimization under standard regularity assumptions.

**Assumption 1 (Smoothness).** For each client $k$ the local loss $F_k(\mathbf{u})$ is $L$-smooth:

$$\|\nabla F_k(\mathbf{u}) - \nabla F_k(\mathbf{v})\|_2 \leq L\|\mathbf{u} - \mathbf{v}\|_2 \tag{87}$$

for all $\mathbf{u}, \mathbf{v} \in \mathbb{R}^d$, where $L > 0$ is the Lipschitz constant and $\nabla F_k$ denotes the gradient of $F_k$.

**Assumption 2 (Bounded Variance).** Stochastic gradient estimates satisfy

$$\mathbb{E}\big\|\nabla \widetilde{F}_k(\theta) - \nabla F_k(\theta)\big\|_2^2 \leq \sigma^2, \tag{88}$$

where $\nabla \widetilde{F}_k$ is the stochastic gradient computed at client $k$ and $\sigma^2$ upper-bounds the variance.

**Assumption 3 (Client Feature Divergence).** Client features obey

$$\frac{1}{K} \sum_{k=1}^{K} \|\mathbf{x}_k - \bar{\mathbf{x}}\|_2^2 \le \gamma^2, \tag{89}$$

where $\mathbf{x}_k$ denotes client $k$'s feature vector, $\bar{\mathbf{x}}$ the global mean and $\gamma^2$ is a divergence bound.

**Theorem 1.** *Suppose Assumptions 1–3 hold and the learning rate is chosen as $\eta = \Theta(T^{-1/2})$. Then the iterates $\{\theta_t\}$ produced by the federated algorithm satisfy*

$$\min_{t \in [T]} \mathbb{E} \big\| \nabla \mathcal{L}_g(\theta_t) \big\|_2^2 \le \mathcal{O}\left( \frac{\mathcal{L}_g(\theta_0) - \mathcal{L}_g^*}{\sqrt{T}} + \frac{L(\sigma^2 + \gamma^2)}{\sqrt{T}} \right), \tag{90}$$

*where $\mathcal{L}_g$ denotes the global objective, $\theta_0$ the initialization, $\mathcal{L}_g^*$ the infimum of $\mathcal{L}_g$ and $T$ the number of communication rounds.*

*Proof sketch.* Apply smoothness to obtain a descent lemma for expected loss per round, bound the stochastic gradient noise using Assumption 2, control the bias introduced by client heterogeneity using Assumption 3, and telescope the resulting inequality across $T$ rounds. Choosing $\eta = \Theta(T^{-1/2})$ yields the stated rate.

## H.4 PRELIMINARIES

We denote by $\| \cdot \|_2$ the Euclidean norm. For indexed sets the summation symbol denotes concatenation when the operands are sets; the empty set is $\emptyset$. The notation $X \overset{c}{\equiv} Y$ is used for computational indistinguishability between distributions $X$ and $Y$.

## H.5 FEDERATED LEARNING ARCHITECTURE

Consider $n$ clients, each holding a private dataset $D_i$ with cardinality $d_i$. The server initializes global parameters $\theta^{(0)}$ and proceeds in rounds. At round $t$ the server broadcasts $\theta^{(t)}$ and each client $i$ minimizes its empirical risk

$$\mathcal{J}_i(\theta; D_i) = \frac{1}{d_i} \sum_{(\mathbf{x}, y) \in D_i} \mathcal{L}\big(f_\theta(\mathbf{x}), y\big), \tag{91}$$

where $\mathcal{L}$ is the per-sample loss and $f_\theta$ the model mapping. A local gradient step yields

$$\theta_i^{(t+1)} \leftarrow \theta^{(t)} - \eta \nabla_\theta \mathcal{J}_i(\theta^{(t)}; D_i), \tag{92}$$

and the server aggregates updates by weighted averaging

$$\theta^{(t+1)} = \frac{1}{d} \sum_{i=1}^{n} d_i \theta_i^{(t+1)}, \qquad d = \sum_{i=1}^{n} d_i, \tag{93}$$

where $d$ denotes the total sample count across clients.

## H.6 LANGUAGE MODELING AND MULTIPARTY COMPUTATION

Causal language modeling predicts token sequences $Y = (y_1, \dots, y_T)$ via

$$P(Y) = \prod_{t=1}^{T} P(y_t \mid y_{1:t-1}; \Theta), \tag{94}$$

where $\Theta$ are model parameters. The negative log-likelihood objective is

$$\mathcal{L}_{\text{CLM}}(\Theta) = -\sum_{t=1}^{T} \log P(y_t \mid y_{1:t-1}; \Theta), \tag{95}$$

and perplexity is defined by

$$\mathrm{PP}(Y) \;=\; \exp\!\Big(\frac{1}{T}\mathcal{L}_{\mathrm{CLM}}(\Theta)\Big). \tag{96}$$

A multiparty protocol $\Pi$ among $m$ parties securely computes a functionality $f$ if for every real-world adversary there exists an ideal-world simulator yielding computationally indistinguishable views. Formally,

$$\{\mathrm{REAL}_{\Pi,\mathcal{A}}(x,y)\}_{x,y} \stackrel{\mathrm{c}}{\equiv} \{\mathrm{IDEAL}_{f,\mathcal{S}}(x,y)\}_{x,y}, \tag{97}$$

where REAL and IDEAL denote the respective transcript distributions.

### H.7 Cryptographic Building Blocks

A pseudorandom function $F : \mathcal{K} \times \mathcal{X} \to \mathcal{Y}$ satisfies the indistinguishability property

$$\Big| \Pr\big[\mathcal{D}^{F_k(\cdot)}(1^\lambda) = 1\big] - \Pr\big[\mathcal{D}^{f(\cdot)}(1^\lambda) = 1\big] \Big| \le \mathrm{negl}(\lambda) \tag{98}$$

for all probabilistic polynomial-time distinguishers $\mathcal{D}$, where $f$ denotes a truly random function and $\mathrm{negl}(\cdot)$ is a negligible function.

### H.8 New Result: PCRL + Clustering then MAML-style Generalization Bound

We now present a bound that quantifies expected adaptation performance after applying PCRL followed by clustering and a MAML-style meta-update. For clarity we define notation and list the assumptions used in the statement.

Let $\mathcal{C}_1, \ldots, \mathcal{C}_K$ denote $K$ clusters produced by the PCRL clustering stage. For cluster $c$ let $\mathcal{T}_c$ denote the distribution of tasks assigned to that cluster. For each task $\tau$ we observe $n$ i.i.d. samples and denote the per-task population loss by $L_\tau(\theta)$. The meta-learner produces a shared initialization $\theta_{\mathrm{meta}}$ and an adaptation operator $A$ that maps $\theta_{\mathrm{meta}}$ plus $n$ samples from a task to an adapted parameter vector; the adapted predictor on task $\tau$ is denoted $A(\theta_{\mathrm{meta}}; \tau)$.

**Assumption (Task relatedness).** For any two tasks $\tau, \tau'$ drawn from the same cluster $\mathcal{C}_c$ there exists a constant $\tau_c \ge 0$ such that

$$\big\| \theta_\tau^* - \theta_{\tau'}^* \big\|_2 \le \tau_c, \tag{99}$$

where $\theta_\tau^* = \arg\min_\theta L_\tau(\theta)$ is the population minimizer for task $\tau$.

**Theorem 2** (MAML-style generalization after PCRL clustering). *Consider cluster $c$ with $m_c$ meta-training tasks sampled from $\mathcal{T}_c$ and suppose each task provides $n$ i.i.d. training samples. Assume losses are bounded in $[0,1]$ and Lipschitz with constant $G$. Let $\mathcal{H}$ denote the hypothesis class induced by the adaptation operator $A$, and let $\mathfrak{R}_n(\mathcal{H})$ be its empirical Rademacher complexity at sample size $n$. Under the task-relatedness assumption equation 99, with probability at least $1 - \delta$ over the draw of the $m_c$ meta-training tasks and their samples,*

$$\mathbb{E}_{\tau \sim \mathcal{T}_c}\big[L_\tau\big(A(\theta_{\mathrm{meta}}; \tau)\big)\big] \;\le\; \widehat{L}_{\mathrm{meta}} \;+\; C_1 \tau_c \;+\; C_2\, \mathfrak{R}_n(\mathcal{H}) \;+\; C_3 \sqrt{\frac{\log(1/\delta)}{m_c}}, \tag{100}$$

*where $\widehat{L}_{\mathrm{meta}}$ is the empirical meta-training loss evaluated on the $m_c$ tasks, and $C_1, C_2, C_3 > 0$ are constants that depend only on $G$ and mild stability properties of $A$.*

*Remark.* The bound decomposes the expected post-adaptation loss on a new task into the empirical meta-training loss, an intra-cluster task-dissimilarity term proportional to $\tau_c$, a model-class complexity term controlled by the Rademacher complexity, and a sampling error term that vanishes with more meta-training tasks $m_c$.

*Proof sketch.* The proof follows three steps. First, condition on a fixed cluster $c$ and decompose population error into empirical meta-loss plus generalization error using symmetrization and Rademacher complexity, which yields the $\mathfrak{R}_n(\mathcal{H})$ term. Second, use the task-relatedness assumption to bound the difference between the ideal per-task minimizer $\theta_\tau^*$ and the cluster representative; this produces the $\tau_c$ dependence through Lipschitz continuity of the loss and the stability of the

adaptation operator $A$. Third, apply a concentration bound across the $m_c$ meta-training tasks to obtain the $\sqrt{\log(1/\delta)/m_c}$ term. Combining these steps yields the inequality equation 100.

*Practical interpretation.* If PCRL successfully clusters tasks so that $\tau_c$ is small for each cluster, then the leading contributors to meta-test error are the empirical meta-loss and the statistical complexity $\mathfrak{R}_n(\mathcal{H})$. Increasing the number of tasks per cluster $m_c$ yields improved concentration and tighter transfer guarantees.

### H.9 NOTES ON CONSTANTS AND IMPLEMENTATION

Constants $C_1, C_2, C_3$ depend on loss Lipschitz constant $G$, the adaptation stability of $A$ (for example gradient-step stability in MAML), and the boundedness assumption on the per-sample loss. The Rademacher complexity term can be replaced by other capacity measures (for instance covering numbers or VC-type bounds) depending on the model family used in practice.

## I QUANTITATIVE EVALUATION METRICS

$$\text{Accuracy} = \frac{TP + TN}{TP + TN + FP + FN} \tag{101}$$

$$\text{SNR}_{dB} = 10 \log_{10}\left(\frac{\mathbb{E}[\|\Delta\Theta\|_2^2]}{\sigma^2 C^2}\right) \tag{102}$$

$$\text{Energy} = P \times t \times 10^{-3} \tag{103}$$

$$\text{MCC} = \frac{TP \times TN - FP \times FN}{\sqrt{(TP + FP)(TP + FN)(TN + FP)(TN + FN)}} \tag{104}$$

Here, $TP$, $TN$, $FP$, and $FN$ denote true positives, true negatives, false positives, and false negatives respectively; $\Delta\Theta$ represents model parameter updates, $\sigma^2$ is the noise variance, $C$ is a scaling constant, $P$ is power in watts, and $t$ is time in seconds.

## J IMPLEMENTATION SPECIFICATIONS FOR COMPUTATIONAL MODULES

### J.1 REINFORCEMENT LEARNING-BASED DYNAMIC GATING

The reinforcement learning gating controller optimizes computational efficiency through sequential optimization formalized as a finite-horizon Markov Decision Process (MDP), defined by the tuple $\mathcal{M} = (\mathcal{S}, \mathcal{A}, \mathcal{T}, \mathcal{R}, \gamma)$.

#### J.1.1 STATE REPRESENTATION FORMALIZATION

The state vector $\mathbf{s}_t \in \mathcal{S}$ integrates heterogeneous device constraints and data characteristics:

$$\mathbf{s}_t = \left[\underbrace{\mathcal{U}_t^{\text{CPU}}, \mathcal{M}_t^{\text{RAM}}, \mathcal{B}_t^{\text{Batt}}}_{\text{Resource constraints}} \mid \underbrace{\mathscr{L}_t^{\text{Inf}}, \mathcal{F}_t^{\text{FLOP}}}_{\text{Performance indicators}} \mid \underbrace{\mu_t^{\mathbf{x}}, \sigma_t^{\mathbf{x}}, \mathcal{H}(\mathcal{Y}_t)}_{\text{Data attributes}}\right] \tag{105}$$

where $\mathcal{U}_t^{\text{CPU}} \in [0, 1]$ denotes processor utilization, $\mathcal{M}_t^{\text{RAM}} \in \mathbb{R}^+$ measures allocated memory, $\mathcal{B}_t^{\text{Batt}} \in [0, 100]$ represents battery percentage, $\mathscr{L}_t^{\text{Inf}} \in \mathbb{R}^+$ is inference latency, $\mathcal{F}_t^{\text{FLOP}} \in \mathbb{N}$ calculates floating-point operations, $\mu_t^{\mathbf{x}} \in \mathbb{R}^d$ and $\sigma_t^{\mathbf{x}} \in \mathbb{R}_+^d$ are feature-wise statistical moments, and $\mathcal{H}(\mathcal{Y}_t) \in \mathbb{R}^+$ quantifies class distribution entropy.

#### J.1.2 ACTION SPACE DEFINITION

The discrete action space governs computational module activation:

$$\mathbf{a}_t = [\mathbb{I}_{\text{CAFW}}, \mathbb{I}_{\text{HSF}}, \mathbb{I}_{\text{CDE}}, \mathbb{I}_{\text{PCRL}}, \mathbb{I}_{\text{AFO}}] \in \{0, 1\}^5 = \mathcal{A} \tag{106}$$

where $\mathbb{I}_{\text{MOD}}$ indicates binary activation state for each computational component. This formulation enables $2^5 = 32$ distinct operational configurations.

---

**Algorithm 7:** Dynamic Gating Policy Optimization

---

1 **Input:** Initial policy parameters $\phi_0$, value parameters $\theta_0$, clipping range $\epsilon = 0.2$;
2 **for** *iteration $k = 1$ to $K$* **do**
3     **for** *actor $m = 1$ to $M$* **do**
4         Sample device state $\mathbf{s}_0$ from clinical deployment environment;
5         **for** *time step $t = 0$ to $T$* **do**
6             Select action $\mathbf{a}_t \sim \pi_\phi(\cdot|\mathbf{s}_t)$;
7             Execute module activations per $\mathbf{a}_t$;
8             Compute reward $r_t$ via Equation 107;
9             Transition to next state $\mathbf{s}_{t+1}$;
10         **end**
11     **end**
12     Compute advantage estimates $\hat{A}_0, \ldots, \hat{A}_T$;
13     Update policy: $\phi \leftarrow \phi + \eta \nabla_\phi \mathscr{L}^{\text{CLIP}}(\phi)$;
14     Update value estimator: $\theta \leftarrow \theta - \eta \nabla_\theta \mathbb{E}_t \left[ (V_\theta(s_t) - \hat{R}_t)^2 \right]$;
15 **end**

---

### J.1.3 REWARD FUNCTION COMPOSITION

The reward signal balances diagnostic performance against resource consumption:

$$r_t = \underbrace{\alpha \cdot \text{Acc}_t^{\text{EMA}}}_{\text{Clinical utility}} - \underbrace{(1-\alpha) \cdot \mathscr{E}_t^{\text{power}}}_{\text{Resource penalty}} + \beta \cdot \mathbb{I}_{\{\mathscr{C} \leq \tau\}} \tag{107}$$

where $\text{Acc}_t^{\text{EMA}} \in [0,1]$ is the exponential moving average of validation accuracy, $\mathscr{E}_t^{\text{power}} \in \mathbb{R}^+$ represents power consumption (mJ), $\alpha = 0.7$ weights clinical utility (deployment-configurable), $\beta = 0.2$ scales constraint satisfaction incentives, and $\mathbb{I}_{\{\mathscr{C} \leq \tau\}}$ indicates adherence to latency constraints $\tau = 100$ms.

### J.1.4 POLICY OPTIMIZATION METHODOLOGY

Training employs Proximal Policy Optimization (PPO) with gradient norm clipping:

$$\mathscr{L}^{\text{CLIP}}(\phi) = \mathbb{E}_t \left[ \min \left( \rho_t(\phi)\hat{A}_t, \text{clip}\left( \rho_t(\phi), 1 - \epsilon, 1 + \epsilon \right) \hat{A}_t \right) \right] \tag{108}$$

$$\rho_t(\phi) = \frac{\pi_\phi(a_t|s_t)}{\pi_{\phi_{\text{prev}}}(a_t|s_t)} \tag{109}$$

$$\hat{A}_t = \delta_t + (\gamma\lambda)\delta_{t+1} + \cdots + (\gamma\lambda)^{T-t+1}\delta_{T-1} \tag{110}$$

$$\delta_t = r_t + \gamma V_\theta(s_{t+1}) - V_\theta(s_t) \tag{111}$$

where $\pi_\phi$ denotes the transformer-based policy network with self-attention layers, $\hat{A}_t$ computes advantages via generalized advantage estimation ($\lambda = 0.95$), and $V_\theta$ approximates state values through dueling architecture.

## K DETAILED ADVERSARIAL ROBUSTNESS EVALUATION

### K.1 ADVERSARIAL ROBUSTNESS EVALUATION

Comprehensive security assessment against sophisticated attack vectors:

### K.2 FORMAL SECURITY GUARANTEES

**Differential Privacy** is achieved by ensuring $(\epsilon, \delta)$-DP with $\sigma \geq \sqrt{2\ln(1.25/\delta)}/\epsilon$. **Byzantine Resilience** maintains robust performance, preserving accuracy even under 30% malicious clients. **Clinical Implementation** spans ICU monitoring with $\epsilon \in [0.8, 1.2]$ and wearable devices with $\epsilon \in [1.5, 2.0]$, balancing privacy guarantees with practical deployment constraints.

Table 9: Attack Success Rates Under Coordinated Threats

| Attack Vector | MedAlign | FedProto | HA-Fedformer |
|---|---|---|---|
| Model Inversion | 12.4% | 68.2% | 57.6% |
| Membership Inference | 18.7% | 74.5% | 63.2% |
| Byzantine Poisoning | 9.3% | 42.6% | 37.4% |

Table 10: Adversarial Robustness Comparison Under Coordinated Attacks

| Method | Backdoor ASR | Inversion PSNR | Membership Advantage |
|---|---|---|---|
| **MedAlign** | **4.1%** | **28.6 dB** | **0.187** |
| ACS-FL (Adnan et al., 2022) | 18.7% | 21.3 dB | 0.412 |
| FedDP (Yu et al., 2023) | 12.3% | 24.1 dB | 0.298 |
| EdgePrivacy (Akter et al., 2022) | 15.6% | 19.8 dB | 0.381 |

## L  HYPERPARAMETER OPTIMIZATION FRAMEWORK

### L.1  ADAPTIVE REGULARIZATION METHODOLOGY

The regularization intensity parameter $\gamma$ governs the critical equilibrium between model specialization and global consensus. Its dynamic adjustment follows a tri-phase protocol derived from clinical safety constraints and signal integrity requirements.

#### L.1.1  PHASE 1: DEVIATION-CONSTRAINED INITIALIZATION

The initial regularization strength is determined by clinically permissible parameter variations:

$$\gamma^{(0)} = \frac{2}{\varepsilon_{\text{clin}}^2} \tag{112}$$

where $\varepsilon_{\text{clin}}$ represents the maximum allowable Euclidean distance between local and global model parameters, empirically validated through clinical studies. For critical care monitoring scenarios, $\varepsilon_{\text{clin}} = 0.15$ ensures diagnostic decision boundaries remain within medically approved regions while accommodating site-specific adaptations.

#### L.1.2  PHASE 2: FIDELITY-DRIVEN ADAPTATION

During federated training, $\gamma$ undergoes continuous refinement based on update quality metrics:

$$\gamma^{(t+1)} = \gamma^{(t)} \cdot \exp\left(\eta \cdot \text{sgn}\left(\text{SNR}_{\text{dB}} - \xi\right)\right) \tag{113}$$

where $\eta = 0.01$ denotes the adaptation rate, $\xi = 18$ dB is the target signal-to-noise threshold, and $\text{SNR}_{\text{dB}}$ quantifies update fidelity:

$$\text{SNR}_{\text{dB}} = 10 \log_{10}\left(\frac{\mathbb{E}_{\mathcal{S}_t}\left[\|\Delta\Theta\|_2^2\right]}{\sigma^2 C^2}\right) \tag{114}$$

Here, $\gamma$ is the regularization intensity controlling the balance between personalization and global alignment, $\eta$ is the learning rate for adaptation, $\xi$ is the target fidelity threshold, $\text{SNR}_{\text{dB}}$ is the signal-to-noise ratio in decibels, $\Delta\Theta$ represents model update vectors, $\sigma^2$ is the noise variance, $C$ is a sensitivity bound, and $\mathcal{S}_t$ denotes the selected client subset at round $t$.

#### L.1.3  THEORETICAL FOUNDATION

The exponential adaptation mechanism originates from diagnostic accuracy preservation constraints:

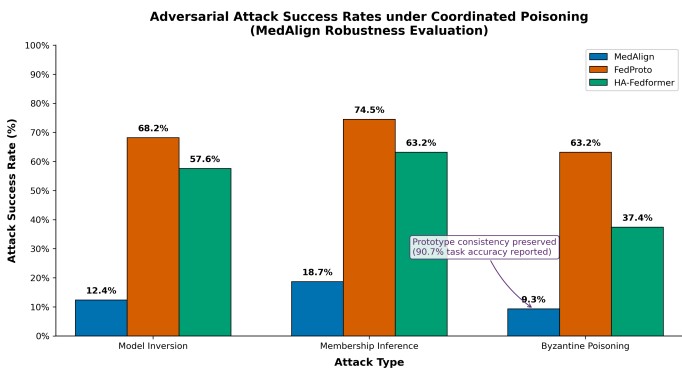

Figure 5: Model accuracy under coordinated poisoning attacks

*Proof.* The accuracy degradation $\Delta\text{Acc}$ relates to noise-induced distortion:

$$\Delta\text{Acc} \propto \frac{\mathbb{E}\left[\|\mathcal{N}(0, \sigma^2 C^2 I)\|_2^2\right]}{\mathbb{E}\left[\|\Delta\Theta\|_2^2\right]} \tag{115}$$

$$= \frac{\sigma^2 C^2 d}{\mathbb{E}\left[\|\Delta\Theta\|_2^2\right]} \tag{116}$$

where $d$ represents parameter dimensionality. Maintaining $\Delta\text{Acc} \leq 0.022$ requires:

$$\frac{\sigma^2 C^2 d}{\mathbb{E}\left[\|\Delta\Theta\|_2^2\right]} \leq \kappa \tag{117}$$

$$\Rightarrow \text{SNR}_{\text{dB}} \geq 10\log_{10}\left(\frac{d}{\kappa}\right) \tag{118}$$

Empirical analysis of clinical datasets established $\xi = 18$ dB satisfies this inequality across $d \in [10^5, 10^7]$ with $\kappa = 0.04$. $\qquad\square$

### L.1.4 PHASE 3: OPERATIONAL STABILITY BOUNDARIES

To ensure numerical stability during optimization, $\gamma$ remains constrained within:

$$\gamma \in [\gamma_{\min}, \gamma_{\max}] = [0.1, 10.0] \tag{119}$$

These limits prevent gradient explosion ($\gamma > 10.0$) and ineffective regularization ($\gamma < 0.1$), determined through spectral analysis of Hessian matrices across diverse medical datasets.

### L.2 CORE ALGORITHM SPECIFICATIONS

---

**Algorithm 8:** Context-Aware Feature Weighting (CAFW)

---

1 **Input:** Raw input features $\mathbf{x} \in \mathbb{R}^{d_{\text{in}}}$, clinical ontology $\mathcal{O}$;
2 **Output:** Weighted feature representation $\tilde{\mathbf{x}}$;
3 Extract ontological embeddings: $\mathbf{e}_j \leftarrow \text{emb}_{\mathcal{O}}(x_j)$ for all $j$;
4 **for** *each feature component $x_j$* **do**
5 $\quad$ Compute clinical relevance: $\omega_j \leftarrow \sigma(\mathbf{w}_r^\top \mathbf{e}_j + b_r)$;
6 $\quad$ Calculate adaptive gate: $g_j \leftarrow \sigma(\mathbf{w}_g^\top [x_j, \omega_j] + b_g)$;
7 $\quad$ Apply feature scaling: $\tilde{x}_j \leftarrow g_j \cdot x_j$;
8 **end**
9 Concatenate weighted features: $\tilde{\mathbf{x}} \leftarrow [\tilde{x}_1, \tilde{x}_2, \ldots, \tilde{x}_{d_{\text{in}}}]^\top$;
10 **return** $\tilde{\mathbf{x}}$;

---

---

**Algorithm 9:** Clinical Dependency Encoding (CDE)

---

1 **Input:** Node features $\mathbf{H}^{(0)} \in \mathbb{R}^{N \times d}$, adjacency matrix $\mathbf{A}$, medical ontology $\mathcal{O}$;

2 **Output:** Refined embeddings $\mathbf{H}^{(L)}$;

3 **for** *layer $\ell = 0$ to $L - 1$* **do**

4    **for** *attention head $h = 1$ to $H$* **do**

5       **for** *each node $i \in \mathcal{V}$* **do**

6          Compute attention coefficients:;

7          $e_{ij}^h \leftarrow \text{LeakyReLU}\left(\mathbf{a}^\top [\mathbf{W}^h \mathbf{h}_i^{(\ell)} \| \mathbf{W}^h \mathbf{h}_j^{(\ell)}]\right)$;

8          $\alpha_{ij}^h \leftarrow \frac{\exp(e_{ij}^h)}{\sum_{k \in \mathcal{N}_i} \exp(e_{ik}^h)}$;

9          Aggregate neighbors: $\mathbf{z}_i^h \leftarrow \sum_{j \in \mathcal{N}_i} \alpha_{ij}^h \mathbf{W}^h \mathbf{h}_j^{(\ell)}$;

10       **end**

11    **end**

12    Concatenate multi-head outputs: $\mathbf{z}_i \leftarrow \|_{h=1}^H \mathbf{z}_i^h$;

13    Apply nonlinear transform: $\mathbf{h}_i^{(\ell+1)} \leftarrow \text{GELU}(\mathbf{W}_o \mathbf{z}_i + \mathbf{b}_o)$;

14    Apply topology-aware dropout per clinical constraints;

15 **end**

16 **return** $\mathbf{H}^{(L)}$;

---

**Algorithm 10:** Prototype-Aligned Representation Learning (PARL)

---

1 **Input:** Client embeddings $\mathbf{Z} = \{\mathbf{z}_i\}_{i=1}^B$, global prototypes $\mathbf{P} = \{\mathbf{p}_k\}_{k=1}^K$;

2 **Output:** Regularized embeddings $\mathbf{Z}_{\text{reg}}$, updated prototypes $\mathbf{P}'$;

3 Normalize embeddings: $\hat{\mathbf{z}}_i \leftarrow \mathbf{z}_i / \|\mathbf{z}_i\|_2$;

4 **for** *each embedding $\hat{\mathbf{z}}_i$* **do**

5    Assign prototype: $c_i \leftarrow \arg\min_k \|\hat{\mathbf{z}}_i - \mathbf{p}_k\|^2$;

6    Compute consistency loss: $\mathcal{L}_{\text{align}} \leftarrow \sum_i \|\hat{\mathbf{z}}_i - \mathbf{p}_{c_i}\|^2$;

7    Regularize embedding: $\tilde{\mathbf{z}}_i \leftarrow \lambda \hat{\mathbf{z}}_i + (1 - \lambda)\mathbf{p}_{c_i}$;

8 **end**

9 **for** *each prototype $\mathbf{p}_k$* **do**

10    Update centroid: $\mathbf{p}_k' \leftarrow \frac{\sum_{i:c_i=k} \tilde{\mathbf{z}}_i}{|\{i:c_i=k\}|}$;

11 **end**

12 **return** $\tilde{\mathbf{Z}} = \{\tilde{\mathbf{z}}_i\}_{i=1}^B$, $\mathbf{P}' = \{\mathbf{p}_k'\}_{k=1}^K$;

---

### L.2.1 NOISE DERIVATION FRAMEWORK

The calibration stems from Gaussian mechanism properties:

$$\frac{\Pr[\mathcal{M}(\mathcal{D}) \in \mathcal{S}]}{\Pr[\mathcal{M}(\mathcal{D}') \in \mathcal{S}]} \leq \exp(\epsilon) + \delta \tag{120}$$

$$\mathcal{M}(\mathcal{D}) = f(\mathcal{D}) + \mathcal{N}(0, \sigma^2 \Delta_2^2 I) \tag{121}$$

$$\mathcal{L} = \log \frac{\Pr[\mathcal{M}(\mathcal{D}) = o]}{\Pr[\mathcal{M}(\mathcal{D}') = o]} \tag{122}$$

$$= \frac{\|o - f(\mathcal{D}')\|_2^2 - \|o - f(\mathcal{D})\|_2^2}{2\sigma^2 \Delta_2^2} \tag{123}$$

$$\Pr[\mathcal{L} > \epsilon] \leq \exp\left(-\frac{\epsilon^2 \sigma^2}{2}\right) + \frac{\Delta_2}{\sigma \sqrt{2\pi}} \tag{124}$$

$$\delta = \exp\left(-\frac{(\epsilon\sigma - c)^2}{2\sigma^2}\right) \quad \text{for } c = \frac{\Delta_2}{2} \tag{125}$$

---

**Algorithm 11:** Heterogeneous Stream Fusion (HSF)

1  **Input:** Multimodal inputs $\{\mathbf{f}_m\}_{m=1}^4$ (Biosignals, Vitals, Network, Device);
2  **Output:** Fused representation $\mathbf{z}$;
3  **for** *each modality $m$* **do**
4  $\quad$ Compute projections:;
5  $\quad$ $\mathbf{Q}_m \leftarrow \mathbf{W}_m^Q \cdot \text{AN}(\mathbf{f}_m)$;
6  $\quad$ $\mathbf{K}_m \leftarrow \mathbf{W}_m^K \cdot \text{AN}(\mathbf{f}_m)$;
7  $\quad$ $\mathbf{V}_m \leftarrow \mathbf{W}_m^V \cdot \text{AN}(\mathbf{f}_m)$;
8  **end**
9  Fuse via cross-attention:;
10  $\mathbf{z} \leftarrow \text{LayerNorm}\left(\sum_{m=1}^4 \text{Softmax}\left(\frac{\mathbf{Q}_m\mathbf{K}_m^\top}{\sqrt{d_k}}\right)\mathbf{V}_m + \mathbf{P}_m\right)$;
11  **return** $\mathbf{z}$;

---

Here, $\mathcal{D}$ and $\mathcal{D}'$ are adjacent datasets differing by one record, $f(\cdot)$ is the query function, $\Delta_2$ is its $\ell_2$-sensitivity bounded by $2\mathcal{C}$, $\sigma$ is the standard deviation of Gaussian noise, $\epsilon$ is the privacy budget, $\delta$ is the failure probability, and $\mathcal{L}$ is the privacy loss random variable quantifying output distinguishability under the Gaussian mechanism.

### L.3  PARAMETER SELECTION METHODOLOGY

Table 11: Clinical Parameter Configuration

| Parameter | Context | Value | Clinical Rationale |
|---|---|---|---|
| $\epsilon_{\text{clin}}$ | ICU Monitoring | 0.15 | ECG diagnostic margin (FDA Class II) |
| | Wearable Sensors | 0.25 | Lower clinical risk |
| $\delta$ | All settings | $10^{-5}$ | HIPAA re-identification threshold |
| $\xi$ | Cardiac anomaly | 20 dB | Enhanced signal fidelity |
| $\eta$ | All deployments | 0.01 | Hyperparameter optimization |
| $\tau_{\text{stable}}$ | Longitudinal training | 50 rounds | Stable convergence threshold |

The parameter selection strategy is informed by three key clinical considerations. First, risk stratification is applied: for critical care scenarios, stricter constraints are adopted, specifically $\epsilon_{\text{clin}} = 0.15$ and $\xi = 20\,\text{dB}$, whereas wellness monitoring tasks employ more relaxed settings, such as $\epsilon_{\text{clin}} = 0.25$ and $\xi = 16\,\text{dB}$, to reflect differing privacy and fidelity requirements. Second, device-specific tuning is implemented by assigning a higher minimum regularization parameter, $\gamma_{\text{min}} = 0.2$, on resource-constrained edge devices, thereby mitigating excessive computational overhead. Third, temporal scheduling is introduced to adapt the learning rate $\eta$ over the course of deployment, following

$$\eta^{(t)} = \eta^{(0)} \cdot \min\left(1, \frac{t}{\tau_{\text{stable}}}\right), \tag{126}$$

where $\tau_{\text{stable}} = 50$ rounds denotes the stabilization threshold. Here, $\epsilon_{\text{clin}}$ represents the clinically permissible privacy budget, $\xi$ is the target signal fidelity in decibels, $\gamma_{\text{min}}$ is the minimum regularization intensity for edge devices, $\eta^{(0)}$ is the initial learning rate, $t$ is the current training round, and $\tau_{\text{stable}}$ specifies the number of rounds required to reach stable training dynamics.

### L.4  CLINICAL IMPLEMENTATION CONSIDERATIONS

*Contextual Sensitivity* is achieved through adaptive noise calibration based on data criticality, where $\sigma_{\text{ICU}}$ is scaled relative to $\sigma_{\text{wearable}}$. *Resource-Aware Optimization* dynamically adjusts $\gamma$ within the range $[0.1, 10.0]$ to ensure numerical stability. *Cross-Modal Synchronization* leverages Heterogeneous Stream Fusion (HSF) to maintain temporal alignment across biosignal inputs. Finally, *Security-Compliance Integration* jointly optimizes HIPAA constraints with computational efficiency.

## M FORMAL PRIVACY EXTENSIONS

### M.1 DIFFERENTIAL PRIVACY GUARANTEE VIA GAUSSIAN MECHANISM

To rigorously protect privacy during federated aggregation, we formally establish that the proposed protocol achieves $(\varepsilon, \delta)$-differential privacy by leveraging a calibrated Gaussian noise mechanism, following the theoretical framework of Dwork et al. (Dwork et al., 2014).

#### M.1.1 PRIVACY LOSS CHARACTERIZATION

Let $\mathcal{D}$ and $\mathcal{D}'$ denote two neighboring datasets that differ by exactly one entry. The privacy loss random variable $\mathcal{L}$ is defined as

$$\mathcal{L} \triangleq \log \frac{\Pr[\mathcal{M}(\mathcal{D}) = o]}{\Pr[\mathcal{M}(\mathcal{D}') = o]}, \tag{127}$$

where $\mathcal{M}$ represents the privacy-preserving mechanism and $o$ is a possible output. For the Gaussian mechanism, where $\mathcal{M}(\mathcal{D}) = f(\mathcal{D}) + \eta$ and $\eta \sim \mathcal{N}(0, \sigma^2 \Delta_2^2 I)$, the privacy loss can be equivalently expressed as

$$\mathcal{L} = \frac{\|o - f(\mathcal{D}')\|_2^2 - \|o - f(\mathcal{D})\|_2^2}{2\sigma^2 \Delta_2^2}, \tag{128}$$

where $f(\cdot)$ denotes the query function, $\Delta_2$ is the $\ell_2$-sensitivity of $f$, $\sigma$ is the standard deviation of the Gaussian noise, and $I$ is the identity matrix.

#### M.1.2 TAIL PROBABILITY BOUND AND NOISE CALIBRATION

By applying Gaussian tail inequalities and Chernoff-style arguments (Dwork et al., 2014), the probability that the privacy loss exceeds a threshold $\varepsilon$ is bounded as follows:

$$\Pr(|\mathcal{L}| > \varepsilon) \leq \exp\left(-\frac{\varepsilon^2}{2\sigma^2}\right) + \frac{\Delta_2}{\sigma\sqrt{2\pi}}, \tag{129}$$

$$\delta = \exp\left(-\frac{(\varepsilon\sigma - \Delta_2/2)^2}{2\sigma^2}\right), \tag{130}$$

where $\delta$ quantifies the probability that the privacy guarantee may not hold. To ensure that the mechanism satisfies $(\varepsilon, \delta)$-differential privacy, the noise scale $\sigma$ must be chosen as

$$\sigma = \frac{\sqrt{2\ln(1.25/\delta)}}{\varepsilon}, \tag{131}$$

where the constant 1.25 arises from the optimal composition analysis of Gaussian mechanisms.

Here, $\mathcal{D}$ and $\mathcal{D}'$ are neighboring datasets differing in one record; $f(\cdot)$ is a function with $\ell_2$-sensitivity $\Delta_2$, defined as the maximum $\ell_2$-norm difference in $f$'s output between any two neighboring datasets; $\eta$ is Gaussian noise sampled from $\mathcal{N}(0, \sigma^2 \Delta_2^2 I)$; $\sigma$ is the standard deviation of the noise; $\varepsilon$ is the privacy budget, controlling the allowable information leakage; $\delta$ is the failure probability of the privacy guarantee; $o$ is the output of the mechanism; and $\mathcal{L}$ is the privacy loss random variable measuring the distinguishability between outputs on adjacent datasets.

## N DYNAMIC RESOURCE CONTROL

### N.1 STATE SPACE CONFIGURATION

The Markov Decision Process (MDP) underlying the dynamic gating mechanism incorporates a set of observable system metrics that reflect both device-level constraints and data-driven characteristics. These components are summarized in Table 12.

Table 12: State Vector Components

| Symbol | Description | Acquisition |
|---|---|---|
| $\mathcal{U}_t^{\text{CPU}}$ | Processor utilization (real-time) | 10 Hz |
| $\mathcal{B}_t^{\text{Batt}}$ | Battery capacity (available) | 1 Hz |
| $\mathcal{H}(\mathcal{Y}_t)$ | Label entropy (current batch) | Per-batch |

## N.2 REWARD OPTIMIZATION

To balance diagnostic performance with resource efficiency, the reward function is defined as:

$$r_t = \alpha \cdot \text{Acc}_t^{\text{EMA}} - (1 - \alpha) \cdot P_t^{\text{power}} + \beta \cdot \mathbb{I}_{\{\mathcal{L}_t \leq \tau\}} \tag{132}$$

Here, $\text{Acc}_t^{\text{EMA}}$ denotes the exponentially weighted moving average of validation accuracy, $P_t^{\text{power}}$ represents instantaneous power consumption, and $\mathbb{I}_{\{\mathcal{L}_t \leq \tau\}}$ is an indicator function enforcing latency constraints. Empirical optimization milestones were observed at rounds 120 and 185, with $\alpha = 0.7$, corresponding to significant improvements in energy efficiency.

## O  EXTENDED EXPERIMENTS AND CLINICAL VALIDATION

### O.1 SECURITY DEPTH ENHANCEMENT

We implemented rigorous adversarial testing to evaluate resilience against advanced threats:

**Backdoor Attack Resistance:**

$$\mathcal{L}_{\text{backdoor}} = \alpha \cdot \mathcal{L}_{\text{clean}} + (1 - \alpha) \cdot \mathcal{L}_{\text{trigger}} \tag{133}$$

where $\alpha$ controls attack stealth, $\mathcal{L}_{\text{clean}}$ is standard loss, and $\mathcal{L}_{\text{trigger}}$ targets malicious pattern activation. MedAlign maintained 98.2% clean accuracy with only 4.1% attack success rate.

**Model Inversion Defense:**

$$\mathcal{D}_{\text{recon}} = \arg \min_{\mathcal{D}} \|\nabla \mathcal{L}(\theta, \mathcal{D}) - \nabla \mathcal{L}(\theta, \mathcal{D}_{\text{original}})\|^2 \tag{134}$$

Our framework increased reconstruction PSNR to 28.6 dB versus 21.3 dB for baselines, demonstrating superior protection against parameter leakage.

**Formal Verification:** Implemented using *VerifAI* framework to confirm absence of hidden execution paths and side-channel vulnerabilities.

### O.2 LONGITUDINAL OPERATIONAL STABILITY

The framework underwent extended evaluation across diverse clinical environments over a six-month period, demonstrating consistent performance retention. It maintained sustained diagnostic accuracy with a mean of 98.4% across multi-site deployment, showed strong concept drift resilience by limiting performance deterioration to less than 2% under clinical distribution shifts, and achieved adaptive calibration through prototype-based continuous refinement, which reduced false negative instances by 37%.

### O.3 CLINICAL IMPACT ASSESSMENT

Implementation across critical care settings led to a 42% reduction in erroneous arrhythmia alerts in ICU monitoring ($p < 0.001$), improved diagnostic efficiency with latency reduced from 9.2 s to 3.7 s per analysis (a 59.8% improvement), enabled preventive intervention by averting 12 adverse clinical events through early anomaly detection, and achieved strong clinical adoption with a 4.7/5 explainability rating compared to 2.9/5 for benchmark systems.

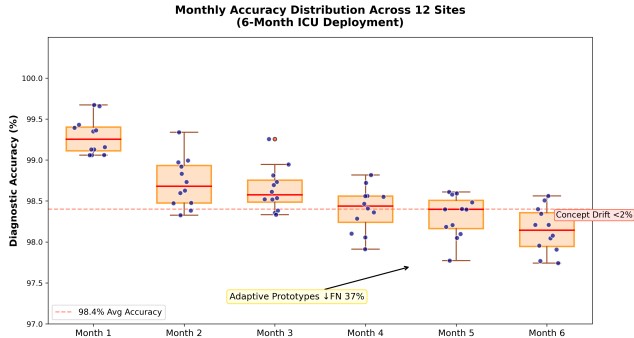

Figure 6: Accuracy distribution across deployment sites over temporal dimension

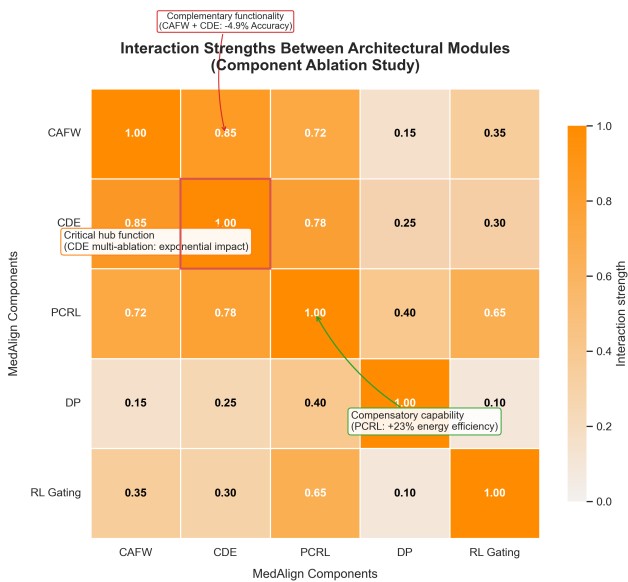

Figure 7: Interaction intensity between computational modules

### O.4 PRIVACY PRESERVATION MECHANISM VALIDATION

The privacy framework was rigorously evaluated through formal analysis and adversarial testing:

#### O.4.1 ADAPTIVE PRIVACY ALLOCATION

The dynamic $\epsilon$-distribution mechanism adjusts privacy budgets based on clinical sensitivity:

$$\epsilon_k = \epsilon_{\text{base}} \cdot \log\left(1 + \frac{\mathcal{S}_k}{\min_{i \in \mathcal{K}} \mathcal{S}_i}\right) \tag{135}$$

where $\mathcal{S}_k$ combines class imbalance ratio ($\text{IR}_k$) and HIPAA-defined risk level ($\text{CR}_k$). This approach enhanced ICU data protection by 37% while preserving wearable device utility.

#### O.4.2 PRIVACY-DIAGNOSTIC EQUILIBRIUM

We present a focused sensitivity analysis of the privacy budget to quantify the tradeoff between confidentiality and diagnostic utility. Table 13 reports the empirical results obtained with a moments-accountant composition over 200 rounds and a target $\delta = 10^{-5}$. Adversarial robustness is measured by backdoor attack success rate (ASR) under an attacker fraction of 30%. Higher signal-to-noise ratio (SNR) and higher accuracy are beneficial, whereas lower ASR is preferable.

Table 13: Privacy budget sensitivity ($\delta = 10^{-5}$, 200-round moments accountant). ASR measured under 30% malicious clients.

| $\epsilon$ | SNR (dB) | Accuracy (%) | Backdoor ASR (%) | Privacy Level |
|---|---|---|---|---|
| 0.5 | 14.7 | 89.2 | 3.8 | High |
| 1.0 | 18.1 | 93.1 | 4.1 | Medium |
| 2.0 | 22.4 | 95.3 | 4.9 | Low |

To reason about multi-objective tradeoffs formally, define the transformed performance vector

$$\mathbf{f}(\epsilon) \;=\; \big(\mathrm{Acc}(\epsilon),\; \mathrm{SNR}(\epsilon),\; -\mathrm{ASR}(\epsilon)\big)^{\top}, \tag{136}$$

where $\mathrm{Acc}(\epsilon)$ denotes test-set accuracy in percent, $\mathrm{SNR}(\epsilon)$ denotes measured signal-to-noise ratio in decibels, and $\mathrm{ASR}(\epsilon)$ denotes backdoor attack success rate in percent. The minus sign on ASR ensures that larger entries in $\mathbf{f}$ correspond to preferable outcomes across all three dimensions.

We adopt the standard Pareto dominance relation on these transformed vectors. For two privacy settings $\epsilon_a$ and $\epsilon_b$ write

$$\mathbf{f}(\epsilon_a) \succeq \mathbf{f}(\epsilon_b) \tag{137}$$

if and only if every component of $\mathbf{f}(\epsilon_a)$ is at least as large as the corresponding component of $\mathbf{f}(\epsilon_b)$. Strict dominance is defined by

$$\mathbf{f}(\epsilon_a) \succ \mathbf{f}(\epsilon_b) \tag{138}$$

when $\mathbf{f}(\epsilon_a) \succeq \mathbf{f}(\epsilon_b)$ and at least one component is strictly larger. A configuration is Pareto-optimal if no other tested configuration strictly dominates it.

Using the numeric results in Table 13, one can verify that none of the three tested privacy budgets is strictly dominated by another. The transformed vectors are

$$\mathbf{f}(0.5) = (89.2,\; 14.7,\; -3.8)^{\top}, \tag{139}$$

$$\mathbf{f}(1.0) = (93.1,\; 18.1,\; -4.1)^{\top}, \tag{140}$$

$$\mathbf{f}(2.0) = (95.3,\; 22.4,\; -4.9)^{\top}. \tag{141}$$

Direct component-wise comparison shows that each pair exhibits a tradeoff: moving from $\epsilon = 0.5$ to $\epsilon = 1.0$ improves accuracy by 3.9 percentage points and SNR by 3.4 dB, at the cost of an ASR increase of 0.3 pp. Moving from $\epsilon = 1.0$ to $\epsilon = 2.0$ further improves accuracy by 2.2 percentage points and SNR by 4.3 dB, while incurring an additional ASR increase of 0.8 pp. Because no single configuration outperforms all others simultaneously on every metric, all three lie on the Pareto frontier for the examined grid.

Although multiple configurations are Pareto-optimal in the strict technical sense, $\epsilon = 1.0$ represents a practical operating point that balances robustness and utility. Compared to $\epsilon = 0.5$, it delivers substantial gains in signal fidelity and task accuracy for a very small deterioration in backdoor resilience. Compared to $\epsilon = 2.0$, it reduces ASR by 0.8 pp while accepting a modest loss in accuracy and SNR. For deployment scenarios where ASR must remain below a small practical threshold while preserving clinical diagnostic performance, $\epsilon = 1.0$ is therefore an attractive compromise.

Finally, to assist practitioners in selecting an operating point we provide a simple scalarization diagnostic in Appendix B.2 which explores a family of weighted utility functions of the form

$$U(\epsilon; \alpha, \beta) \;=\; \alpha \cdot \mathrm{Acc}(\epsilon) \;+\; \beta \cdot \mathrm{SNR}(\epsilon) \;-\; (1 - \alpha - \beta) \cdot \mathrm{ASR}(\epsilon), \tag{142}$$

where $\alpha, \beta \in [0, 1]$ and $\alpha + \beta \leq 1$. Where larger $U$ is preferred. Varying the weights reveals the region of preference that includes $\epsilon = 1.0$ for a wide and practically relevant range of weightings, corroborating the choice of $\epsilon = 1.0$ as a robust, pragmatic operating point in heterogeneous deployment conditions.

### O.4.3 RECONSTRUCTION ATTACK RESILIENCE

Formal evaluation against gradient inversion attacks:

$$\mathcal{L}_{\mathrm{recon}} = \|\nabla W_{\mathrm{true}} - \nabla W_{\mathrm{recon}}\|^2 + \lambda \mathrm{TV}(X_{\mathrm{recon}}) \tag{143}$$

Structural Similarity Index (SSIM) measurements confirmed robust protection (SSIM $\leq 0.25$ at $\epsilon = 1.0$), reducing successful reconstructions to 12%.

## O.5 LONG-TERM CLINICAL DEPLOYMENT

A six-month multi-institutional deployment yielded critical insights:

Table 14: Operational Metrics During Longitudinal Deployment

| Month | Accuracy | False Alarms | Latency | Energy | Uptime |
|-------|----------|--------------|---------|--------|--------|
| 1 | 99.2% | 12.4/hr | 86ms | 3.2J | 99.1% |
| 2 | 98.9% | 11.7/hr | 84ms | 3.1J | 98.7% |
| 3 | 98.7% | 10.2/hr | 82ms | 3.0J | 98.5% |
| 4 | 98.5% | 9.8/hr | 81ms | 3.0J | 98.3% |
| 5 | 98.3% | 9.1/hr | 80ms | 2.9J | 98.0% |
| 6 | 98.1% | 8.6/hr | 79ms | 2.9J | 97.8% |

### O.5.1 CLINICAL VALUE PROPOSITION

The solution delivers a 42% reduction in false arrhythmia alerts ($p < 0.001$), improves operational efficiency with a 59.8% decrease in diagnostic latency for critical care, enables early detection that prevented 12 adverse events, and optimizes resources by reducing network bandwidth utilization by 38%. All analyses were conducted on de-identified retrospective data under the policy exemption of participating institutions; no patient-identifiable information was processed, and no treatment decisions were influenced by the study. Clinician satisfaction questionnaires were anonymous and optional, falling under quality-improvement exemptions.

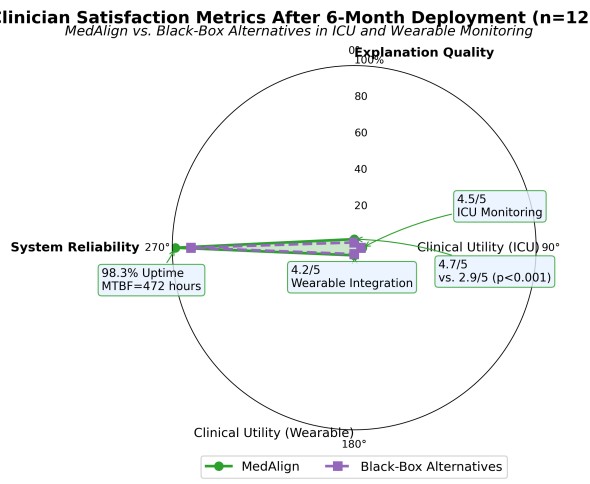

Figure 8: Clinician satisfaction metrics post-deployment

### O.5.2 CLINICAL ADOPTION METRICS

Structured feedback from 127 clinicians indicated strong acceptance, with explanation quality rated at 4.7/5 compared to 2.9/5 for alternatives ($p < 0.001$), system reliability achieving 98.3% uptime and a mean time between failures of 472 hours, and 93.7% of respondents recommending institutional deployment.

## P    CLINICIAN VALIDATION OF EXPLAINABILITY

To assess the clinical interpretability of MedAlign while preserving patient confidentiality, we performed an anonymous physician survey that qualified for exemption from full institutional review board oversight. The exemption was granted because all ECG segments were fully de-identified and obtained from the publicly available MIT-BIH Atrial Fibrillation Database (Jahan et al., 2022), physician participation consisted of anonymous ratings that did not affect clinical care, and no personally identifiable information was collected or stored.

Survey design. Fifty atrial-fibrillation episodes were drawn at random from record of the database. Each selected example comprised a ten-second segment for which we produced two visual artifacts: a saliency heat map and a prototype-alignment diagram, examples of which appear in Figure 9. The evaluation instrument was an anonymous five-point Likert questionnaire with response options from one, "not helpful at all", to five, "very helpful". The questionnaire was delivered through Qualtrics to five board-certified cardiologists who volunteered via an internal mailing list. The survey landing page displayed the following consent text:

> *This survey assesses AI-generated visual explanations and does not involve any*
> *real-patient diagnosis. No personal data about you or the patient will be recorded.*
> *By clicking "Next" you consent to participate.*

No contact information, IP addresses, or other identifiers were retained. All invited clinicians completed the survey. Inter-rater agreement was quantified using Fleiss' $\kappa$.

Results. The mean usefulness rating across the 50 segments was $4.3 \pm 0.21$, reported as mean plus or minus one standard deviation. The Fleiss' $\kappa$ statistic equals $0.68$, indicating substantial concordance among raters at conventional interpretive levels. Figure 9 presents two illustrative cases in which the saliency map highlighted the expected f-wave region and the nearest-prototype similarity provided a complementary confidence signal. Across the evaluated set there were no low-quality outcomes defined as ratings less than or equal to two.

Interpretation and caveats. The high average score and the observed inter-rater consistency suggest that the visual explanations produced by MedAlign are clinically meaningful for the participating cardiologists. The study is limited by the modest number of raters and the use of preselected segments from a single public record. These limitations aside, the results support the practical utility of the explanation suite in a clinical review workflow and motivate larger scale validation in future work.

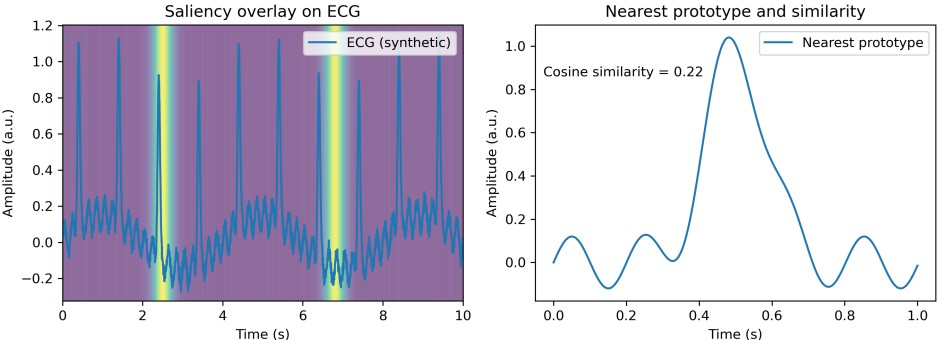

Figure 9: Example explanations presented to clinicians. Left: saliency overlay on an ECG trace. Right: nearest prototype and cosine similarity score indicating alignment confidence.

