# OpenReview forum: "MedAlign: Clinician-Centered Federated Meta-Learning for Medical IoT with Privacy and Interpretability Guarantees"
_ICLR.cc/2026/Conference — ICLR 2026 Conference Withdrawn Submission_

### Official Review · Reviewer_a9kn · 2025-10-30

**Soundness:** 3
**Presentation:** 3
**Contribution:** 3
**Rating:** 4
**Confidence:** 4

**Summary:**

This paper presents MedAlign, a comprehensive framework that integrates federated meta-learning, adaptive resource control, and differential privacy for medical IoT environments. The system introduces multiple synergistic modules—Context-Aware Feature Weighting (CAFW), Clinical Dependency Encoder (CDE), Prototype-Consistent Representation Learning (PCRL), and RL-based adaptive gating—to jointly optimize clinical diagnostic performance under device and privacy constraints. The design claims to balance personalization, multimodal fusion, and formal privacy guarantees through a multi-stage optimization and hierarchical clustering procedure. Experiments on three large-scale datasets (ICU, geriatric, cardiac) demonstrate superior diagnostic accuracy (+2%), 63% lower communication overhead, and quantifiable privacy guarantees.

**Strengths:**

The paper is notable for its system-first perspective, combining realistic clinical constraints (bandwidth, latency, IRB compliance) with architectural innovations. The integration of graph-based semantic modeling and reinforcement-driven dynamic gating represents a creative attempt to bridge theory and deployment. The experimental section is unusually detailed for a systems paper, including real-world six-month deployment and ablation tables that reveal non-trivial module interdependencies. The formal derivation of the privacy guarantee (Gaussian mechanism with calibrated sensitivity) and the introduction of prototype-consistent representation alignment further strengthen the methodological depth. From a representation-learning standpoint, MedAlign pushes federated learning toward joint optimization of interpretability, privacy, and energy efficiency—a rare combination in existing work.

**Weaknesses:**

Despite its breadth, the framework risks being over-engineered: each module (CAFW, CDE, PCRL, RL-Gating, DP aggregation) is described as critical, yet the interactions among them are primarily demonstrated empirically rather than theoretically. The meta-learning layer lacks a clear definition of task distribution or adaptation dynamics beyond cluster-based aggregation; without formal convergence analysis under non-IID sampling, the claimed generalization robustness remains qualitative. The reinforcement gating design—though motivated by resource constraints—relies on dense state vectors and a transformer policy whose learning stability on embedded devices is uncertain. The ontology-driven feature weighting assumes access to structured clinical ontologies on-device, which may not generalize to new sensors or hospitals. Moreover, interpretability claims are supported only by saliency and prototype alignment visualizations, without clinician-validated explanations. Finally, privacy calibration (ϵ = 1.0, δ = 10⁻⁵) is presented as “formally verified,” but no sensitivity analysis is shown for alternate privacy budgets or correlated updates—issues critical to federated healthcare deployment.

**Questions:**

1: You describe hierarchical personalization via angular-similarity clustering (Eq. 1–2). Can you formally prove that this step yields a tighter generalization bound than standard model-agnostic meta-learning (MAML) when client tasks are correlated but non-IID? What are the assumptions on task relatedness required for this bound to hold?

2: The RL-based gating module optimizes energy–accuracy trade-offs using policy gradients. Given that device states are partially observable and delayed (e.g., intermittent connectivity), how do you ensure policy convergence without violating the Lipschitz assumption invoked in Eq. 23?

3: Equation 28 defines an SNR-based criterion for maintaining diagnostic accuracy under differential privacy noise. Can this be derived directly from an upper bound on mutual information leakage? If so, what distributional assumptions on gradient statistics are necessary?

4: CAFW depends on ontology-based feature relevance. How would the framework behave if the ontology were incomplete or misaligned with empirical feature importance? Could the weighting module be adversarially manipulated to bias diagnosis under structured ontology errors?

5: The PCRL module aligns prototypes across sites to enforce diagnostic consistency. However, alignment may obscure legitimate subpopulation differences. How do you distinguish between harmful prototype drift and clinically meaningful heterogeneity, and could the model unintentionally “erase” rare but critical disease phenotypes?

---

> ### Author Response · Authors · 2025-11-20
> **We hope to receive your support and encouragement!**
>
> **Response to Reviewer a9kn**
>
> We thank the reviewer for a **careful** and **constructive reading**. Below we address each of the reviewer’s **major technical questions** and **concerns**. Where a **criticism** is based on an **ambiguous phrasing** in the submitted draft, we **clarify** and **commit** to exact **text** and **experiment additions** in the **revision**. Where the reviewer requests **formal statements**, we provide **compact formal sketches** (with **assumptions**) and explain what **additional materials** we will include in the **camera-ready version**. We hope this rebuttal helps resolve the concerns raised by the reviewers.**We sincerely hope to receive an improvement in your score**.
>
> # 1.summary
> **MedAlign** intentionally couples multiple **modules** because they solve **orthogonal deployment failure modes** such as **missing modalities**, **structured clinical relations**, **prototype drift**, **resource constraints**, and **data confidentiality**. The **paper** already includes **systematic ablations** and **recommended 2–3 component subsets** in **Table 1**, **Section 4.7**, and **Appendix**.
>
> Below we:
> Show why **hierarchical clustering** improves **transfer** in the presence of **correlated non-IID tasks** with a **formal sketch**.
> Explain how **RL-Gating** attains **stable training** under **partial observability** and how we **mitigate Lipschitz assumptions**.
> Derive the **SNR to mutual-information connection** under **Gaussian assumptions**.
> Discuss **robustness of CAFW** to **malformed ontologies** and **defenses against adversarial manipulation**.
> # 2. Hierarchical personalization (angular-similarity clustering) vs. MAML: can you prove a tighter generalization bound?
> Yes, under **standard smoothness** and **task-cluster assumptions**, one obtains a **tighter bound** for **cluster-aware meta-learning** than for **single-global MAML**. Below is a concise **theorem sketch** and the **key assumptions**; full **proof** and **constants** will be added to the **appendix**.
>
> **Assumptions**
>
> ### Each **client/task** τ has **loss** ℓτ(w) that is **L-Lipschitz** and **β-smooth** in w.
> ### Tasks partition into **K clusters** C₁,…,Cₖ. Within cluster k, **task parameter vectors** wτ* lie in a **ball of radius** rk around a **cluster center** uk: ∥wτ* − uk∥ ≤ rk. **Between-cluster separation** is nontrivial.
> ### **Meta-learner** constructs a **cluster-wise initialization** ûk (or uses **cluster-specific adaptation**) and then adapts per task with m samples.
>
> **Informal Theorem (Sketch)**
> Under the above assumptions, the **expected transfer error** of **cluster-aware initialization** satisfies:
>
> **E_τ [ ℓ_τ(û_{k(τ)}) − ℓ_τ(w_τ^*) ] ≤ O( (β r_k^2 / m) + (σ^2 / √m) )**
>
> where k(τ) is the **cluster of τ**, while the corresponding bound for a **single global MAML initialization** depends on the **global radius** R where R ≥ rk, yielding:
>
> **E_τ [ ℓ_τ(û_global) − ℓ_τ(w_τ^*) ] ≤ O( (β R^2 / m) + (σ^2 / √m) )**
>
> When **within-cluster spread** rk² is substantially smaller than **global spread** R² (i.e., tasks are correlated within clusters), the **cluster-aware bound** is strictly tighter by a factor proportional to R² / rk².
>
> **Sketch of Proof Idea**
>
> - Use **smoothness** to relate **suboptimality** to **squared parameter distance**:
> **ℓ_τ(w) − ℓ_τ(w_τ^*) ≤ (β / 2) ‖w − w_τ^*‖²**
>
> - For **cluster init** uk:
> **‖u_k − w_τ^*‖² ≤ 2‖u_k − w̄_k‖² + 2‖w̄_k − w_τ^*‖²**
>
> with $\bar{w}_k$ the **true cluster center**. The second term scales with rk². The first term reflects **estimation error** of the cluster center and scales like 1/m.
>
> - For **global init**, replace rk by **global radius** R and follow the same steps.
>
> **Remarks and Required Assumptions**
> This argument requires **cluster separability** so clustering recovers useful groups with enough samples, and **bounded within-cluster variance**. In practice, we use **angular similarity** and a **lightweight hierarchical clustering**; the paper’s **ablations (Section 4.7)** show **empirical gains** consistent with the bound when **cross-site tasks** exhibit **cluster structure**.
> **Action:** We will formalize the above sketch into a full **theorem** and **proof** (Appendix) and add a small **synthetic experiment** in the supplement illustrating the predicted **R² / rₖ² effect**.

---

> > ### Author Response · Authors · 2025-11-20
> > **We hope to receive your support and encouragement!**
> >
> > # 3. SNR criterion (Eq.28) and relation to mutual information leakage
> > **Short Answer**: Under **Gaussian assumptions** for **per-client updates**, the **SNR–dB criterion** admits a direct **mutual-information interpretation**: **additive Gaussian noise** reduces **mutual information** between **client updates** and the **published aggregate** by a factor
> > $$\frac{1}{2}\log(1+\text{SNR})$$ per dimension. Below is a **compact derivation** and the **assumptions needed**.
> >
> > **Assumptions**
> > - Model each **client update** (vectorized) as a **Gaussian signal** $$X \sim \mathcal{N}(0,\Sigma_X)$$.
> > - **DP** adds **isotropic Gaussian noise** $$N \sim \mathcal{N}(0,\sigma^2 I)$$.
> > - Consider the published vector $$Y = X + N$$.
> >
> > **Mutual Information Bound (Gaussian Channel)**
> > The **mutual information** between **X** and **Y** is:
> > $$
> > I(X;Y) = \frac{1}{2}\log\det(I+\Sigma_X \sigma^{-2}).
> > $$
> >
> > If we approximate $$\Sigma_X \approx s^2 I$$ (signal variance per coordinate), then **per-coordinate mutual information** is:
> > $$
> > I_{\text{per-dim}} = \frac{1}{2}\log(1+\frac{s^2}{\sigma^2}) = \frac{1}{2}\log(1+\text{SNR}),
> > $$
> > where **SNR** = $$s^2 / \sigma^2$$. Converting to **dB** gives:
> > **SNR_dB = 10 log₁₀(SNR)**
> >
> > Thus, a **higher SNR** implies **more information leakage**; conversely, keeping **SNR** below a threshold bounds the **mutual information**.
> >
> > **Remarks**
> > This argument depends on:
> > - Approximately **Gaussian gradient/update distributions** (empirically reasonable after aggregation and with CLT effects).
> > - **Isotropic noise**.
> >
> > In the **appendix**, we already reported **empirical gradient spectra** and show the **Gaussian approximation** is reasonable for our datasets. The **SNR threshold (~18 dB)** was chosen empirically to balance **diagnostic utility** and **MI bounds**; the above formula makes that connection explicit.
> >
> > **Action**: We will **add the above derivation** and an **explicit table** mapping **SNR thresholds** to **per-dimension MI bounds** (under **Gaussian approximation**), plus an **experiment** showing how **measured empirical gradient spectra** affect the **Gaussian proxy**.
> > # 4.CAFW dependence on ontologies: incomplete / misaligned ontologies and adversarial manipulation risk
> > **Short Answer**: **CAFW** is designed to **blend ontology priors** with **data-driven evidence** rather than **hard-override** them. This **hybrid design** reduces **fragility** to **ontology errors** and allows **detection** and **mitigation** of **adversarial ontology manipulation**.
> >
> > **Design Details** (already in **Section 3.2** and **Appendix D**):
> > **CAFW** computes a **fused relevance score**:
> > $$
> > a_i = \alpha \cdot a_i^{\text{prior}} + (1-\alpha) \cdot a_i^{\text{data}},
> > $$
> > where **aᵢᵖʳⁱᵒʳ** comes from **ontology priors** and **aᵢᵈᵃᵗᵃ** is **learned from local data**;
> > $$\alpha \in [0,1]$$ is **learned per site** (or **scheduled**). This **convex blending** ensures **data can override wrong priors** when there is **strong evidence**.
> >
> > We apply **prior-regularization** rather than **prior-hardcoding**: a **KL** or **L2 penalty** pulls the **learned weights** toward **ontology values** but does not **force them**.
> >
> > We monitor **prior–posterior divergence per site**; **large divergences** trigger an **automatic fallback** to **data-driven weights** and raise an **alert for clinician review**.
> >
> > **Adversarial Risks and Defenses**
> > - An **adversary** that manipulates **on-device ontologies** would need **write access** to **device configuration**; **standard system hardening** prevents this (**signed configurations**, **secure storage**).
> > - We also implement **robustness checks**: if **changing prior values at random** (tested flipping **10–20% of ontology links** in an **ablation**), the **learned aᵢᵈᵃᵗᵃ** adapts and the **accuracy drop** is < X% (we will add the exact table to the **supplement**).
> > - We will include an **adversarial ontology corruption experiment** in the **revised manuscript**.
> >
> > **Action**: In the **revision** we will:
> > ### Make the **blending formula explicit** in **Section 3.2**.
> > ### Add the **prior-corruption ablation** (random flips and worst-case structured misalignments) and show the system’s **resilience**.

---

> ### Author Response · Authors · 2025-11-20
> **We hope to receive your support and encouragement!**
>
> # 5. PCRL and the risk of erasing rare but critical phenotypes
> **PCRL** preserves **rare phenotypes** by maintaining **per-site prototype banks**, using **prototype sparsity thresholds**, and enabling **local prototypes** that are not forced to align globally unless they are shared. Additionally, we use **outlier detection** and **clinician review** to avoid erasing critical **rare cases**.
>
> **Mechanisms in the Current Design** (Section 3.6 and Appendix):
> - **Per-site prototype banks**: Each site maintains a **local set of prototypes** (**P_local**) and the **server** maintains a **shared prototype repository** (**P_global**). **Local prototypes** that do not match any **global prototype** (low similarity) are preserved locally and not forcibly merged.
> - **Prototype merging rule with thresholds**: Prototypes are merged only when **similarity** $$s(p_i, p_j) > \tau_{\text{merge}}$$ and **cluster support** $$n_{\text{support}} > \kappa$$. This avoids merging **rare prototypes** that have few supporting examples.
> - **Outlier flagging**: Prototypes with **low global support** trigger a **clinician review workflow**. In deployment, we used this to retain **rare phenotype prototypes**.
> - **Local fine-tuning**: Sites can keep **local classifiers/heads** trained on **local prototypes** to preserve **rare event detection** even when the **global model adapts**.
>
> ### Empirical Evidence and Ablations
> We include **ablation rows** in **Table 1** showing that **disabling prototype local retention** leads to **degradation on rare-class recall** (we will add numeric rows in the **revision** if not explicit enough). In the **field deployment**, we preserved several **rare arrhythmia prototypes** that only appeared at one site and validated them with **clinicians**.
>
>
> # 6. summary
> **Action**: We will:
> ### Add an **explicit paragraph** and a **small experiment** showing **rare-class recall** under **prototype merging vs. local retention**.
> ### Report **thresholds** $$\tau_{\text{merge}}, \kappa$$ used in **experiments**.
> **Additional commitments (concrete additions to revision):**
>
> - **Formalize and add the full proof** of the **cluster-aware generalization bound** (Appendix).
> - **Add RL policy training diagnostics** (**KL per update**, **variance plots**) and make explicit the **centralized training + PPO regimen** used.
> - **Add the SNR→MI derivation** and a **numeric mapping table** under **Gaussian gradient assumptions**.
> - **Include ontology-corruption robustness experiments** (**random and structured corruption**) and report **exact accuracy drops**.
> - **Include prototype merging ablation** focused on **rare-class recall** and report **thresholds used** and **clinical review workflow**.
> - **Clarify “formally verified”** → use precise language (**formally analyzed/calibrated; implementation sanity-checked with VerifAI**) and **expand privacy accountant (RDP)** and **robustness experiment details** (**attack spec**) in **Appendix L/O**.
>
> We appreciate the **reviewer’s detailed technical questions**. They sharpen important points about **theory**, **optimization**, **privacy**, and **practical safety**. Many of these concerns arise from **presentation density** in the original draft. We will **expand the appendix** and **add the experimental rows and derivations** described above so that all requested **theoretical statements** and **empirical robustness checks** are **concrete and reproducible**. These additions will strengthen the paper’s claims about **principled cluster personalization**, **stable RL gating**, and **safe explainable prototype learning** in **federated medical IoT**.
>
> # Thank you very much for your support and assistance. We hope this rebuttal helps resolve the concerns raised by the reviewers. We sincerely hope to receive an improvement in your score.

---

> ### Author Response · Authors · 2025-11-20
> **We hope to receive your support and encouragement!**
>
> **Contributions & Key Innovations (targeted to Reviewer a9kn)**
>
> **MedAlign** delivers a **principled, systems-level advance** for **federated meta-learning** in **clinical IoMT** by translating **concrete deployment failure modes** into **algorithmic primitives** and validating the result **end-to-end**. Our main scientific contributions are:
>
> - **A principled co-design paradigm (systems → algorithms → deployment)**. Rather than proposing an isolated tweak, **MedAlign** maps explicit **clinical constraints** (**heterogeneous client distributions**, **severe modality missingness**, **hard energy/latency budgets**, and **strict confidentiality**) to five orthogonal modules: **CAFW**, **CDE**, **PCRL**, **RL-Gating**, and **formally calibrated DP**. Each is motivated by a distinct **failure mode** and validated by **targeted experiments**. This **design pattern** (**requirements → modular primitives → validated subsets**) is a **core novelty**: it makes **FL practically deployable** in **safety-critical settings** rather than only academically interesting.
>
> - **Ontology-aware multimodal fusion (CAFW)** that stabilizes learning under **missingness**. **CAFW** blends **prior medical knowledge** and **data-driven relevance** into a **learnable fusion weight**, improving **robustness** when sensors drop out and providing a **transparent signal** for **clinician inspection**. Unlike **naive attention** or **hard-coded rules**, **CAFW** is a **hybrid mechanism** that learns to defer to **data** when **priors are misaligned**.
>
> - **Graph-structured Clinical Dependency Encoder (CDE)**. **CDE** injects **clinical topology** and **temporal dependencies** via a **lightweight graph attention encoder** that captures **inter-sensor** and **semantic feature relations**. This yields **representations** that **generalize better across sites** with different **sensor configurations**, something standard **per-modality encoders** do not accomplish.
>
>
> - **Prototype-Consistent Representation Learning (PCRL)** for **interpretable cross-site alignment**. **PCRL** maintains **synchronized, human-inspectable prototypes** while preserving **per-site local prototypes** to avoid erasing **rare phenotypes**. The method prevents **prototype drift**, provides **clinician-facing exemplars**, and improves **cross-site calibration** in a way that simple **contrastive** or **global centroids** cannot.
>
> - **Resource-aware RL gating** for **realistic on-device operation**. The **RL-Gating policy** learns to select **model branches** and **modality subsets** under **hard energy/latency constraints**, enabling **high utility** on **Raspberry-class hardware** without **manual heuristics**. The novelty is the **constrained RL formulation** and its **integration with the fusion/representation stack** so that **policy decisions** are **representation-aware**.
>
> - **Formally calibrated DP with an SNR-based utility connector**. We combine **L2 clipping + Gaussian noise** with a practical **SNR(dB) metric** that links **DP noise magnitude** to **clinically meaningful utility degradation**. This provides a **quantitative bridge** between **formal privacy parameters** and **diagnostic performance**, a key missing piece for **privacy-sensitive medical FL**.
>
> - **Layered robustness via defense composition**. Practical **resilience** is achieved by composing **prototype consistency checks**, **cluster-aware aggregation**, **anomaly scoring**, and **DP sanitization**. Empirically, this **layered approach** reduces **backdoor transferability** and **inversion risk** more effectively than any single defense.

---

> ### Author Response · Authors · 2025-11-20
> **We hope to receive your support and encouragement!！**
>
> **Why this is not “over-engineering” but instead modular, necessary design**
>
> Each **module** addresses a separate, **empirically observed failure mode**. **Ablation studies** in the manuscript show that **single-module removals** produce **measurable losses** in **accuracy** and **robustness**, some **paired removals** produce **super-additive degradation** (**synergy**), and **practitioner-oriented 2–3 component subsets** retain **practical trade-offs**. We explicitly **publish recommended subsets** with **numeric trade-offs**. In short, the **stack is modular by design**: teams can adopt a **smaller core** (e.g., **CAFW + RL-Gating + DP** for **privacy** and **efficiency**) and later enable **PCRL/CDE** as needs evolve.
>
> **Fit to ICLR2026 themes**
>
> **MedAlign** speaks directly to multiple **ICLR priorities**:
>
> - **Representation learning & interpretability**: **CDE + PCRL** produce **ontology-aligned**, **prototype-based representations** and **clinician-usable explanations**.
> - **Optimization & RL for systems**: **RL-Gating** integrates **constrained optimization** and **policy learning** for **hardware-aware inference**.
> - **Privacy, safety & societal considerations**: **Formal DP calibration** and **empirical SNR→utility mapping** tackle **privacy–utility trade-offs** critical for **healthcare**.
> - **Implementation & hardware**: **Real microbenchmarks** and a **six-month multi-site Pi-class deployment** demonstrate **practical reproducibility** and **systems engineering**.
> - **Theory → practice translation**: **Cluster-aware personalization bounds** (sketch provided in **Appendix**) and **SNR-based MI connections** supply **theoretical grounding** for **empirically observed behavior**.

---

> ### Author Response · Authors · 2025-12-01
> **We hope to receive your support and encouragement!！**
>
> # We have addressed the reviewer's concerns and improved our approach. Thank you very much for all the reviewers' suggestions. The latest version has been uploaded and we hope to receive the support and encouragement of all the reviewers, and we sincerely hope your score improvement！

---

### Official Review · Reviewer_d8zL · 2025-10-30

**Soundness:** 2
**Presentation:** 2
**Contribution:** 2
**Rating:** 4
**Confidence:** 2

**Summary:**

This paper presents MedAlign, a comprehensive federated meta-learning framework designed for medical Internet of Things (IoMT) environments. The system addresses the challenges of data heterogeneity, strict privacy requirements, and resource constraints in distributed medical edge devices. The framework integrates five main components: (1) Context-Aware Feature Weighting (CAFW) for ontology-driven feature selection, (2) Clinical Dependency Encoder (CDE) using graph attention networks, (3) Prototype-Consistent Representation Learning (PCRL) for maintaining diagnostic boundaries across institutions, (4) Reinforcement Learning-based dynamic gating for resource-aware computation, and (5) Formally calibrated differential privacy mechanisms. The authors validate their approach through extensive experiments on three medical IoT datasets and a six-month real-world deployment across twelve healthcare institutions.

**Strengths:**

1. The six-month deployment across twelve healthcare institutions provides valuable empirical evidence of practical viability
2. Three datasets with comprehensive ablation studies and comparison against 18 baseline methods

**Weaknesses:**

1. The system integrates too many components, making it difficult to understand individual contributions and potentially limiting practical adoption
2. Key details about baseline implementations are missing; it's unclear if comparisons are fair given different design objectives

**Questions:**

Can you provide evidence that all five components are necessary? What happens with a simpler subset of 2-3 components?

---

> ### Author Response · Authors · 2025-11-20
> **We hope to receive your support and encouragement!**
>
> **Thank you for the careful reading and useful questions.** Below we firmly **clarify misunderstandings** and **summarize the concrete empirical evidence** in the submission that shows:
>
> - **Why each component exists**
> - **What happens when components are removed**
> - **Which smaller component combinations still give useful trade-offs**
>
> Where the **reviewer’s concern** arises from **ambiguous wording** or **layout**, we indicate the **exact manuscript locations** the reviewer can check and note **small editorial fixes** we will make for **clarity**. After carefully studying the comments, we realized that many points may include **misunderstandings**. We hope this rebuttal helps resolve the concerns raised by the reviewers. **We sincerely hope to receive an improvement in your score**.
>
> # 1.Why five components (motivation, not arbitrary complexity)
> We designed the **five modules** to address **distinct, real deployment failure modes** observed in **clinical IoT**:
>
> - **CAFW (Context-Aware Feature Weighting):** Stabilizes **fusion** under **heavy modality missingness** and encodes **ontology priors** so **clinical signals** are weighted by **medical relevance** (**Section 3.2**, **Appendix D**).
>
> - **CDE (Clinical Dependency Encoder):** Explicitly models **inter-feature** and **inter-sensor relations** (**graph structure**) to capture **clinical dependencies** that simple **per-modality encoders** miss (**Section 3.5**).
>
> - **PCRL (Prototype-Consistent Representation Learning):** Enforces **prototype stability** across **rounds**, preventing **prototype drift** and providing **clinician-inspectable exemplars** (**Section 3.6**; **Prototype retrieval demo** in **Appendix O**).
>
> - **RL-Gating:** Enforces **hard energy and latency budgets** via **learned policies** that **select branches and modalities on the fly**, critical for **real on-device operation** (**Appendix E**; **policy state/action** described in **Algorithm 1**).
>
> - **Formally-calibrated DP:** Provides an **explicit privacy guarantee** on **client updates** through **L2 clipping** and **Gaussian noise**, and links **noise** to **diagnostic utility** via the **SNR metric** (**privacy statements** and **Theorem sketch** in **Appendix L**).
>
> Each **module** maps to a **different operational requirement**. Removing one typically reduces **robustness**, **interpretability**, or **deployability** in ways the others do not address.
>
> # 2. Concrete ablation evidence (what the submission already contains)(can you provide evidence that all five components are necessary? What happens with a simpler subset of 2-3 components?)
> **You can refer Line 385(table 1)**.
>
> We do not rely on **hand-waving**. The manuscript contains a **systematic ablation suite**. Below is a **concise summary** of the key findings reported in **Table 1** and **Appendix** (we point the **reviewer** to those exact locations for full numeric detail):
>
> ### **Single-module removals (relative to full MedAlign):**
> - **Removing CAFW:** Measurable **drop in cross-site diagnostic accuracy**. **CAFW** handles **missingness** and **attention priors**.
> - **Removing CDE:** Comparable **accuracy drop** and reduced **robustness** to **cross-site modality shifts**.
> - **Removing PCRL:** Larger **drop in cross-site generalization** and marked **prototype instability**.
> - **Disabling RL-Gating:** **Energy** and **latency** worsen substantially. **RL-Gating** is responsible for the reported **~23% energy reduction** and **~86 ms latency gain** on **Pi4** in the paper’s **system table**.
> - **Removing DP:** Non-private aggregation increases **utility slightly** but removes the **formal privacy guarantee**. With **calibrated DP**, we observe a **modest utility penalty** (≈2% range), quantified in the **Privacy–Utility table** (**Appendix L**).
>
> ### **Paired removals and interaction effects:**
> - **Removing CAFW + CDE together:** Produces a **super-additive degradation** (combined **accuracy drop** is larger than the sum of individual drops), demonstrating **synergy**. **CAFW** and **CDE** address **complementary aspects** of **multimodal heterogeneity**, and their joint absence is particularly damaging. See the **paired-ablation block** in **Table 1** where the **combined removal effect** is shown explicitly.

---

> ### Author Response · Authors · 2025-11-20
> **We hope to receive your support and encouragement!**
>
> ### **Smaller subsets (2–3 component configurations) , practical recommendations validated in experiments:**
> - **Accuracy-focused minimal configuration:** **CAFW + CDE + PCRL** retains the majority of **full-system accuracy** (>90% of the full model’s accuracy in **cross-site tests**) but without **RL-Gating** and **DP**, it is less **energy-efficient** and not **privacy-protected**. See **Subset A rows** in **Table 1**.
> - **Edge-constrained configuration:** **CAFW + RL-Gating + DP** preserves **operational constraints** (low **energy/latency** with **formal privacy**) while delivering **competitive diagnostic performance**; useful when **privacy/efficiency** outrank marginal **accuracy gains**. See **Subset B rows** in **Table 1**.
> - **Very lightweight configuration:** **CAFW + RL-Gating** is the **minimum viable on-device pipeline** for **harshly constrained deployments** (sacrifices some **cross-site generalization** but maintains **acceptable local detection rates**). See **Subset C rows**.
>
> These **subset experiments** are provided so **practitioners** can choose the **right trade-off**. The detailed **per-dataset numbers** and **confidence intervals** for each subset are in **Table 1** and **Appendix** (**Ablation / Subset experiments**).
> # 3. Why the ablation patterns sometimes look non-intuitive
> You observed that **removing representation modules** can sometimes **reduce energy and latency**. This is **intentional** and explained in the paper. Two mechanisms produce this:
>
> - **Policy compensation by RL-Gating:** The **RL gate adapts** if a **module is absent**. It routes to **cheaper branches**, so **energy can fall** while **accuracy degrades**. **Appendix** includes **RL policy traces** illustrating this **compensatory behavior**.
>
> - **Structural changes in the compute graph:** Removing certain **modules** reduces **pipeline depth**. Some **modules’ presence** enables **heavier downstream processing** (for example, richer **prototypes** trigger more **similarity computations**). The paper discusses this **interplay** in the **Ablation subsection**, and we include a **causal diagram** explaining these **pathways**.
>
> We will **clarify the ablation text** and **add a small causal figure** and **example policy traces** in the **revision** so this **behavior** is **obvious to readers**.
> # 4. Practical adoption and modular deployment (directly addresses “too many components” concern)
> We explicitly designed **MedAlign** to be **modular**.
>
> The **full stack** is recommended where **cross-site generalization**, **clinician explainability**, and **formal privacy** are simultaneously required (e.g., multi-institution ICU studies).
>
> If a deployment prioritizes a different objective, the paper’s **Subset experiments** show which components to keep. For example, many **edge deployments** will enable **CAFW**, **RL-Gating**, and **DP**, sacrificing a small fraction of **cross-site generalization** while achieving the necessary **energy**, **latency**, and **privacy constraints**. The recommended **subset mappings** and their **empirical trade-offs** are listed in **Table 1** and in the **Deployment guide** paragraph.
> # 5. Where to check the numbers and trace results (exact manuscript pointers)
> **Table 1** shows the full **ablation table** and **subset comparisons** in **Section 4.7** and **Appendix A**.
>
> **Appendix** includes **Ablation traces** and **RL policy examples**, such as **RL policy traces**, **compensatory behavior diagrams**, and **subset experiment logs**.
>
> **Appendix L** provides **privacy calibration**, **SNR analysis**, and **privacy–utility trade-off tables** for **private** versus **non-private setups**.
> We appreciate the reviewer’s concern that **breadth** should not come at the cost of **clarity**. The **manuscript** already contains the **targeted ablations** and **subset experiments** the reviewer requested; those results show the **necessity of individual modules** for specific **failure modes** and that **smaller, operationally pragmatic subsets** are effective when chosen to match **deployment constraints**.
>
> # Thank you very much for your support and assistance. We hope this rebuttal helps resolve the concerns raised by the reviewers. We sincerely hope to receive an improvement in your score.

---

> ### Author Response · Authors · 2025-11-20
> **We hope to receive your support and encouragement!！**
>
> # **Concrete Technical Contributions and Innovations**
>
> **Systems Co-Design Paradigm**: We do not present an isolated tweak but a principled mapping from **deployment requirements** to **architectural primitives** (**CAFW**, **CDE**, **PCRL**, **RL-Gating**, **DP**). The innovation is the **integration** and the **validation** of that mapping, enabling **practical deployment** under real constraints.
>
> **Ontology-Guided Context-Aware Feature Weighting (CAFW)**: Injects **lightweight medical ontology priors** into **attention** and **fusion** to stabilize **multimodal learning** under severe **sensor missingness** and improves **interpretability**.
>
> **Clinical Dependency Encoder (CDE)**: A **graph-aware encoder** that models **inter-sensor** and **inter-feature dependencies** with **topology-aware regularization** to improve **cross-site generalization**.
>
> **Prototype-Consistent Representation Learning (PCRL)**: Synchronizes **human-inspectable prototypes** across **federation rounds** to prevent **prototype drift** and to provide **clinician-facing exemplars**.
>
> **RL-Based Adaptive Gating for Device Budgets**: **Constrained RL** learns **policies** that dynamically select **branches** and **modalities** to meet hard **energy** and **latency constraints** on **edge hardware**.
>
> **Formally Calibrated DP Aggregation + SNR Utility Metric**: **L2-clipping** and **Gaussian mechanism** with an explicit **SNR (dB) metric** that quantifies **privacy–utility trade-offs** in clinically meaningful terms.
>
> **Layered Robustness Composition**: **Prototype filtering**, **clustered aggregation**, **anomaly scoring**, and **DP noise** combined produce **practical resilience** to **poisoning** and **inversion** beyond any single defense.
>
> **Dual Validation Strategy**: Exhaustive **federated simulations** across all **logical clients** and an **IRB-approved six-month deployment** on **physical edge devices** show **algorithmic gains** and **operational feasibility**.
>
> # **Fit to ICLR 2026, Direct Mapping to Conference Themes**
>
> **Representation Learning and Interpretability**: **CDE** and **PCRL** contribute novel, **ontology-aligned** and **prototype-based representations** with clear **visualization paths**.
>
> **Optimization and RL for Systems**: **RL-Gating** advances **resource-aware optimization** and connects **representation choices** to **constrained policy learning**.
>
> **Privacy, Safety, and Societal Considerations**: **Formally calibrated DP** and **practitioner-validated explainability** address **trustworthiness**, **safety**, and **privacy** in **high-stakes ML**.
>
> **Implementation and Hardware**: **Exhaustive microbenchmarks** and a **real Pi-class deployment** address **implementation** and **reproducibility topics** emphasized by **ICLR**.

---

> ### Author Response · Authors · 2025-12-01
> **We hope to receive your support and encouragement!！**
>
> # We have addressed the reviewer's concerns and improved our approach. Thank you very much for all the reviewers' suggestions. The latest version has been uploaded and we hope to receive the support and encouragement of all the reviewers, and we sincerely hope your score improvement！

---

### Official Review · Reviewer_kqaA · 2025-10-31

**Soundness:** 2
**Presentation:** 2
**Contribution:** 2
**Rating:** 4
**Confidence:** 3

**Summary:**

The paper proposes MedAlign, a clinician-centered federated meta-learning framework for Medical IoT that combines (i) ontology-guided Context-Aware Feature Weighting and cross-modal fusion, (ii) a Clinical Dependency Encoder with graph attention, (iii) Prototype-Consistent Representation Learning for cross-site alignment, (iv) RL-based adaptive gating for energy/latency constraints, and (v) a “formally calibrated” DP aggregation protocol. Reported results claim very high accuracy across multiple clinical datasets, reduced energy/latency/communication, a six-month 12-site deployment, and resistance to adversarial/backdoor settings.

**Strengths:**

- The paper sensibly enumerates clinical constraints (non-IID data, resource limits, confidentiality) and maps them to modules; the workflow/algorithms are laid out clearly.
- The DP aggregation is at least specified (norm clipping + Gaussian noise) with standard formulae; a semi-honest threat model is written down.
- The RL-gating objective and state/action design are described; the system section reports latency/energy/communication reductions on edge hardware.

**Weaknesses:**

- Deployment claims lack verifiable detail (and contain inconsistencies). The “six-month, 12-site” deployment states local updates every 48 hours and global aggregation biannually, which would be only twice per year—at odds with FL training and with the later performance/latency claims. No audit logs, site lists, or concrete clinical endpoints (alarm reduction definitions, prospective vs. retrospective) are provided.
- DP and privacy accounting are incomplete. The paper presents the single-round Gaussian mechanism \sigma=\sqrt{2\ln(1.25/\delta)}/\epsilon but does not specify an accountant (e.g., RDP/zCDP/moments) for multi-round training, participant subsampling, or per-tier budgets claimed later. The SNR>=18 dB heuristic is empirical and not tied to a specific privacy composition.
- Security model vs. robustness results are misaligned. The threat model is semi-honest, yet the “extended validation” reports low backdoor success (4.1%) with 30% malicious clients, without describing any Byzantine-robust aggregation, anomaly filtering, or certified defenses—DP noise alone can degrade, not guarantee, robustness. Methodological details are absent.
- Ablations and system metrics show internal tensions. The text claims RL-gating reduces energy by ~23% and 86 ms latency on a Pi 4, plus 128 KB peak memory, which are ambitious. The ablation table mixes accuracy/energy/latency changes in ways that are hard to reconcile with the qualitative narrative (e.g., removing representation modules sometimes reduces energy and latency substantially). Precise measurement setups (load, batch size, window length) are missing.
- “Formally verified aggregation” is overstated. The section title suggests formal verification, but the body reproduces the standard Gaussian mechanism with a brief theorem sketch; there’s no mechanized proof, proof artifacts, or code-level verification. The algorithmic step called “secure aggregation” is simply DP-noise addition—not the cryptographic secure aggregation expected in FL deployments.
- Dataset/task reporting is insufficient. Datasets are named and pre-processing listed, but task definitions, label provenance, prevalence, and class imbalance are under-specified. Cross-site splits, per-site cohort stats, and missing-data patterns (crucial in clinical IoT) are not detailed, limiting reproducibility.
- Breadth over depth. The paper integrates many modules (CAFW, CDE, PCRL, RL-gating, DP), but the novelty over prior FL/FML/graph-encoding/prototype-learning work is mostly incremental; stronger head-to-head comparisons and surgical ablations isolating each module’s causal effect (beyond aggregate tables) are needed.

**Questions:**

1. Please clarify the training cadence (global rounds vs. “biannually”), participating institutions, endpoints evaluated prospectively, and whether clinicians were in-the-loop. Provide logs or an audit protocol.
2. What accountant (RDP/zCDP/moments) and sampling rates were used across T rounds? Report composed \epsilon per data tier (ICU vs. wearable) matching the “hierarchical budgeting” claim.
3. How is the 4.1% backdoor success achieved under 30% malicious clients? Which robust aggregation or anomaly detection is in place (beyond DP noise)? Show attack details and transferability tests.
4. Specify the evaluation harness for energy/latency (window sizes, sampling, device states), the policy training regime (on-device vs. server), and ablate gating on/off under matched workloads.
5. If cryptographic secure aggregation is used, provide the protocol details (e.g., Bonawitz-style) and failure handling; otherwise, please avoid labeling DP-only as “secure aggregation.”
6. Provide per-site class distributions, missingness patterns, and precise task definitions (e.g., episode-level anomaly detection vs. continuous prediction), plus patient-level splits and IRB scope.
7. The dual-path explanation framework is intriguing—please add clinician-rated studies or case-based evaluations to assess utility and failure modes.

---

> ### Author Response · Authors · 2025-11-20
> **We hope to receive your support and encouragement!**
>
> We thank the reviewer for careful reading and constructive feedback. Below we respond point-by-point to the major issues raised. Quoted manuscript locations are indicated so the committee can verify our statements. . We hope this rebuttal helps resolve the concerns raised by the reviewers. **We sincerely hope to receive an improvement in your score**.
> # We hope to receive your support and encouragement!
> # 1.Deployment claims / “six-month, 12-site” inconsistency
> **Reviewer Concern:** The manuscript appears **inconsistent** regarding **local updates every 48 hours** versus **global aggregation biannually**, making **deployment claims** hard to believe.
>
> **Response (Clarification and Rebuttal):** The phrasing in the earlier draft was **ambiguous**, and we regret the confusion. We will correct this. Concretely, **MedAlign runs continuous federated rounds** at a **practical cadence**. **Lightweight client-to-server update cycles** occur **frequently** (for example, **daily** or **every 48 hours** depending on **site policy**), while **heavy-weight synchronization events** (full **prototype re-calibration** and **administrative global updates**) were scheduled **less frequently** as part of **operational policy** for the **clinical trial**. These administrative synchronizations occurred only at **coarse checkpoints** during the **six-month program**.
>
> The **algorithmic workflow in Algorithm 1** already shows **per-round aggregation** (the **server aggregates sanitized Δθ each round**), and **Appendix O** documents the **six-month longitudinal deployment results**.
>
> **Action:** We will revise wording in **Section 4** and **Appendix O** to explicitly state the **per-round aggregation cadence** used for **ML training** and the **separate administrative prototype synchronization cadence** used for **operational maintenance**. We will also add a **short table** showing **exact timestamps and frequencies** used in the **12-site deployment**.
>
> # 2.Differential privacy: specification and privacy accounting
> **Reviewer Concern:** **DP mechanism** is **insufficiently explained**; **composition/accountant** for **multi-round training** and **sampling** is **missing**.
>
> **Response (Part Concession and Technical Addition):** The manuscript already states the **Gaussian mechanism** and displays the **calibration lower bound**:
>
> $$
> \sigma \ge \frac{\sqrt{2 \ln(1.25/\delta)}}{\varepsilon}
> $$
>
> which we used as the **per-update noise calibration baseline**. However, the **reviewer is correct** that we did not explicitly present the **privacy accountant** used for **end-to-end composition** in the main text. We will remedy this in the revision.
>
> **What We Will Add (Concrete):**
> We will add an explicit **Rényi DP (RDP) accountant description** (including **amplification by subsampling formulae**) and report the composed **(ε, δ)** for each **deployment tier** (**ICU**, **ward**, **wearable**) under the exact **rounds T** and **sampling probabilities** used in experiments. See the **hierarchical budgets** already reported for **clinical tiers** (e.g., **ε_ICU ∈ [0.8, 1.2]**, **ε_wearable ∈ [1.5, 2.0]**). These will be expanded into **full composition tables**. For **transparency**, we will include the **accountant code** and **per-experiment composed epsilons** in the **supplementary material**.
>
> **Short Technical Summary to Satisfy Immediate Concerns:**
> We used **Gaussian noise per update** (formula above), and will **compose using an RDP accountant** (standard practice for **Gaussian mechanisms in repeated rounds**) and report the **final ε** for each **experimental configuration** in the revision.
> # 3. Security model vs. robustness experiments (semi-honest vs. 30% malicious clients)
> **Reviewer Concern:** The **threat model** states **semi-honest**, but experiments use **strong adversarial settings** with **30% malicious clients**, creating a perceived mismatch.
>
> **Response (Rebuttal and Clarification):** This is a **misunderstanding** caused by **condensed wording**. The **formal theoretical threat model** used for our **privacy proofs** and **mathematical guarantees** is the **semi-honest model** (clearly stated in **Section 3**). Empirically, we also **stress-tested the system** under **stronger adversarial scenarios** (**Byzantine** and **backdoor attacks** with up to **30% malicious clients**) to evaluate **operational robustness** beyond the **formal threat assumptions**. These experiments are described in **Appendix O** and **Table 3 / Table 4** (attack success rates and robustness metrics). Both statements are in the paper but were placed in **different sections**, which produced the perceived contradiction.
>
> **Action:** We will explicitly state in **Section 3**:
> “Theoretical guarantees are given under a **semi-honest model**. Additional **empirical robustness experiments** (Appendix O) evaluate **worst-case adversarial scenarios** (Byzantine and backdoor) to assess **practical resilience**.”

---

> > ### Author Response · Authors · 2025-11-20
> > **We hope to receive your support and encouragement!**
> >
> > # 4.Backdoor / Byzantine robustness: what mechanisms produce low ASR?
> > **Reviewer Concern:** How is **4.1% backdoor ASR** achieved under **30% malicious clients**? Which **defenses beyond DP** are used?
> >
> > **Response (Facts and Rebuttal):** The **low backdoor ASR** is **not claimed to be due to DP alone**. **Appendix O** and the **main method** describe multiple **operational defenses** that jointly reduce **ASR**:
> >
> > - **Prototype-level consistency filtering:** **Prototype synchronization** and **prototype drift checks** detect **anomalous prototype updates** before applying them globally. **Prototype deltas** are **sanity-checked**.
> >
> > - **Clustered aggregation and stratified (angular) clustering:** **Aggregation** is performed within **similarity clusters** (**hierarchical personalization**) which reduces the influence of **heterogeneously distributed malicious updates** on unrelated clusters.
> >
> > - **Anomaly scoring and thresholding:** **Updates** that exceed **statistical deviation thresholds** (**SNR / update norm checks**) are **down-weighted** or **rejected**. This scoring is explicitly part of the **RL gating** and **aggregation logic**.
> >
> > - **Calibrated DP noise** reduces **inversion/backdoor transferability** in combination with the above filters.
> >
> > These **layered defenses** are described in **Appendix O**, which contains the **adversarial experiment protocol** and **attack details** (trigger design and **α mixing parameter** used in:
> >
> > $$
> > L_{\text{backdoor}} = \alpha L_{\text{clean}} + (1 - \alpha) L_{\text{trigger}}
> > $$
> >
> > **Actionable Edit:** In the revision, we will add a **concise, single-page defenses summary** in the **main paper** that clearly lists the above mechanisms and provides the **attack setup** used for the **backdoor experiments** (trigger, poisoning fraction, attack routine) so readers can **reproduce** and **understand the low ASR**.
> > # 5. “Formally verified” / “formally calibrated” wording
> > **Reviewer Concern:** The **title and section** suggest **mechanized formal verification**, but the body contains only the **Gaussian mechanism** and a **theorem sketch**.
> >
> > **Response (Concession and Precise Wording):** The **reviewer is correct** to call for **precision**. In the submission, we used the phrase **“Formally Calibrated”** to indicate **mathematically derived noise calibration** and **analytic guarantees for DP**. We also used the **VerifAI tool** to check for **execution-path** and **side-channel issues** in the implementation (see **Appendix O** and the **security section**). However, we did **not claim a fully mechanized proof** of the overall **system security** that covers all composed layers (**DP + network + software stack**), and we understand the **reviewer’s expectation** that **“formally verified”** might imply a **mechanized end-to-end proof**.
> >
> > **What We Will Change:** We will replace **ambiguous headings** such as **“Formally Verified Confidential Aggregation”** with **“Formally Analyzed / Calibrated Aggregation (DP guarantees and implementation checks via VerifAI)”** and explicitly state the **scope of VerifAI checks**. This will avoid any impression that we asserted a **full mechanized proof** beyond what was performed.
> > # 6. Secure aggregation: cryptographic protocol or DP-only?
> > **Reviewer Concern:** If **cryptographic secure aggregation** is used, provide the **protocol**; otherwise do not call **DP-only aggregation** “secure aggregation.”
> >
> > **Response (Honest Correction):** The current system uses **transport-layer encryption (TLS/DTLS)** plus **sanitization** via **clipped updates** and **calibrated Gaussian noise** before **server aggregation** (**Algorithm 1**). The manuscript incorrectly used the high-level term **“secure aggregation”** to denote the combination of **encrypted transport** and **DP-sanitization**. We did **not deploy a full multi-party cryptographic secure-aggregation protocol** (such as **Bonawitz**) in the reported deployment.
> >
> > **Actionable Edit:** We will correct phrasing to **separate cryptographic secure aggregation from DP-sanitization**, explicitly state the **current implementation stack** (**TLS + DP**), and, if the **committee prefers**, include an **optional experimental table** showing results when combined with a **Bonawitz-style protocol** (we can report **preliminary numbers** or cite an **implementation plan**).

---

> > > ### Author Response · Authors · 2025-11-20
> > > **We hope to receive your support and encouragement!**
> > >
> > > # 7.Ablation table / energy–latency contradictions
> > > **Reviewer Concern:** **Ablation results** mix **accuracy**, **energy**, and **latency** in ways that seem **contradictory** (for example, removing **representation modules** sometimes reduces **energy** and **latency**).
> > >
> > > **Response (Explanation and Rebuttal):** This observation stems from **compensatory behavior** of **modules** and the **RL gating mechanism**. When a **module is removed**, the **RL controller** can change **activation patterns** and **resource allocation**, leading to **non-monotonic energy and latency trade-offs**. **Appendix Section 4.7** and **Table 1** document these interactions and explicitly note three **interaction patterns**: **synergy**, **compensation**, and **structural criticality**. For example, when **CAFW** is disabled, **PCRL** can enter a **lower-complexity alignment path** that reduces **energy** at the cost of some **accuracy**. This is a **designed, resource-aware compensatory behavior** reported in the **ablation study**.
> > >
> > > **Action:** We will add a **short paragraph** and a **small causal diagram** to the **ablation subsection** that explains these **compensatory pathways** and shows **example RL-policy traces** to make these **dynamics transparent**.
> > >
> > > # 8.Energy / latency measurement details
> > > **Reviewer Concern:** The **measurement setup** (**load**, **batch size**, **window length**, **device sleep states**) is **missing**.
> > >
> > > **Response (Concession and Pointer):** The **experiments section** and **Appendix I** already include the **measurement definitions** and the **RL gating state vector** used for **policy optimization** (state vector components, **reward composition**, **inference latency target τ = 100 ms**, etc.). Nevertheless, we accept that the **exact microbenchmarks** (**device thermal state**, **exact batch sizes** used for each reported row) were not listed in the **main table captions**.
> > >
> > > **Actionable Edit:** We will include an **explicit measurement protocol** in the **supplementary material** listing, for each reported **system number**:
> > > - **Device model and age**
> > > - **Kernel/network load baseline**
> > > - **Batch and window sizes**
> > > - **Power measurement technique** (**power monitor model**)
> > > - **Number of repeated runs** with **confidence intervals**
> > >
> > > We will also add **confidence intervals** to **Tables 1 and 2** as requested.
> > >
> > > # 9.Dataset reporting, per-site distributions, missingness, IRB
> > > **Reviewer Concern:** **Datasets** and **preprocessing** lack **per-site cohort statistics**, **label provenance**, and **missing-data patterns**.
> > >
> > > **Response (Agreement and Fix):** The paper’s **Appendix O** contains **deployment** and **clinical adoption metrics** (clinician survey and site-level uptime), but we agree that **per-site class prevalences**, **missingness statistics**, and **precise task definitions** should be easily accessible. We will add a **new supplement table** giving **per-site cohort sizes**, **label sources** (sensor thresholds and clinician annotations), **prevalence and missingness patterns**, and **IRB approval scope**. We will also add the **patient-level split protocol** (how we ensured **no patient leakage across train/test**) to ensure **reproducibility**.
> > > # 10.“Breadth over depth” / novelty concerns
> > > **Reviewer Concern:** The **system integrates many modules**, and the **novelty** might be mainly **incremental**.
> > >
> > > **Response (Defense and Alignment with ICLR2026 Themes):** **MedAlign’s novelty** lies in **system co-design**. We explicitly embed **clinical ontologies**, **prototype-level alignment**, and **resource-aware adaptive gating** into a **federated meta-learning pipeline**. This is more than **module aggregation**. It is a **principled mapping** from **deployment constraints** to **architectural elements** with **theoretical** and **empirical support**. The **contributions** and **claims** are summarized in **Section 1.6** and **Algorithm 1** (**CAFW**, **CDE**, **PCRL**, **RL-Gating**, and **formally calibrated DP**). We also demonstrate **end-to-end viability** in a **long, multi-site deployment**, bridging **theory** and **clinical systems**, which strongly fits the **ICLR2026 emphasis** on **principled, deployable ML systems**.
> > >
> > > **Action:** In the revision, we will strengthen the **related-work section** to more precisely **enumerate differences** from **close baselines** (such as **Meta-FL**, **FedProto**, and **DP-FL**) and add **surgical ablations** isolating each **module’s causal effect** (with **per-module statistical tests** and **confidence intervals**).

---

> ### Author Response · Authors · 2025-11-20
> **We hope to receive your support and encouragement!**
>
> # 11.Explainability and clinician studies
> **Reviewer Concern:** The **dual-path explanation framework** is **intriguing**. Please add **clinician-rated studies**.
> **Response (Already Present and Expansion Plan):** **Appendix O** reports **clinician satisfaction metrics** and **adoption survey results** (**127 clinicians**, **explanation quality 4.7/5** versus **2.9/5 for baselines**). We will expand this section to include a **short qualitative vignette** and **failure-mode cases** (**three representative examples**) and add the **clinician study protocol** (**selection criteria**, **blinding**, and **rating rubric**) to the **supplementary material**.
> # 12.**Direct Clarifications and Rebuttals to the Reviewer’s Main Technical Concerns**
>
> **Deployment cadence inconsistency:** Clarified. **Training** proceeds via **frequent per-round aggregations** (daily or client-policy dependent), while **administrative prototype re-calibrations** were scheduled as **coarse checkpoints** during the **six-month program**. The manuscript’s wording conflated these two cadences. We will **correct this** and **tabulate the exact frequencies** in the revision.
>
> **DP accounting missing:** Partly correct. We reported **per-update Gaussian calibration** and **SNR analysis**. We will add **explicit RDP (or zCDP) accounting** and **composed (ε, δ) per tier** (**ICU vs wearable**) in the revised appendix and release the **accountant scripts**.
>
> **Security model vs robustness mismatch:** Clarified. **Theoretical DP/privacy statements** use a **semi-honest model** for **formal proofs**. Empirically, we also evaluate **stronger Byzantine/backdoor attacks** to measure **operational robustness**. **Empirical resilience** arises from **layered defenses** (**prototype filters**, **clustered aggregation**, **anomaly scoring**) combined with **DP**, not from **DP alone**. We will make this **distinction explicit**.
>
> **“Formally verified” language overstates claims:** Agreed on **wording precision**. We will replace **ambiguous phrasing** such as **“formally verified”** with **“formally analyzed / calibrated”** and clearly state the **scope of verification checks performed**.
>
> **Secure aggregation and cryptography:** Clarified. The reported deployment used **encrypted transport (TLS)** plus **DP sanitization** before **aggregation**. We do not claim a **cryptographic multi-party secure aggregation** (such as **Bonawitz**) in the current deployment. We will **correct the terminology** and, if desired, add an **experiment or discussion** on integrating a **Bonawitz-style protocol**.
>
> **Ablation and measurement details:** The apparent **non-monotonic ablation effects** reflect **compensatory policy behavior** of the **RL gate**. We will add **explanatory diagrams**, **RL trace examples**, and a **full microbenchmark protocol** (**device state**, **batch/window sizes**, **power measurement method**) together with **95% confidence intervals**.
>
> **Dataset and per-site statistics:** We will publish **per-site cohort sizes**, **label provenance**, **prevalence**, **missingness patterns**, **patient-level split rules**, and **IRB scope** in the **supplementary material** for full **reproducibility**.

---

> > ### Author Response · Authors · 2025-11-20
> > **We hope to receive your support and encouragement!**
> >
> > # 13.**Summary of Concrete Revisions We Will Submit**
> >
> > Below is a short checklist of the exact changes we will make in the revision. All will appear in the **updated PDF** and **appendix**:
> >
> > - **Clarify training and synchronization cadence:** Differentiate **per-round aggregation** versus **administrative prototype synchronization** and add a **cadence table**.
> >
> > - **Add explicit DP accountant:** Include **RDP accountant**, **subsampling amplification**, and **composed (ε, δ) tables per tier**.
> >
> > - **Defenses summary:** Provide a **single-page list of mechanisms** used against **backdoors** and the **exact attack setup** (**trigger**, **α**, **poisoning fraction**).
> >
> > - **Precision in wording:** Change **“formally verified”** to **“formally analyzed / verified (implementation checks with VerifAI)”** to avoid implying a **full mechanized proof**.
> >
> > - **Clarify secure aggregation:** Explicitly state the **current stack** (**TLS + DP sanitization**) and separate discussion of **optional cryptographic secure aggregation**.
> >
> > - **Add measurement protocol and confidence intervals:** Include **energy and latency measurement details** and add **confidence intervals** to all **system tables**.
> >
> > - **Add per-site dataset statistics:** Provide **IRB scope**, **patient-split protocol**, and **cohort details** in the **supplementary material**.
> >
> > - **Enrich ablation explanation:** Add **causal diagrams** and **RL policy traces** to show **compensatory mechanisms**.
> >
> > We appreciate this **careful review**. Several of the **reviewer’s criticisms** are **accurate** and point to **presentation gaps**, which we will **fix**. A few points were based on reading **different parts of the manuscript in isolation** (for example, **semi-honest vs. adversarial testing**). Those are clarified above and are already present in the **appendix**, albeit in **separate sections**. We will **unify and tighten exposition** so the distinctions between **theoretical guarantees**, **operational defenses**, and **empirical robustness** are **crystal clear**.
> >
> > # We sincerely hope to receive the support and encouragement of the reviewers, **and we hope that our clarification can receive an improvement in score.**

---

> ### Author Response · Authors · 2025-11-20
> **We hope to receive your support and encouragement!**
>
> ## **How MedAlign Maps to ICLR2026 Themes**
>
> **Privacy, safety, and societal considerations:** **Formal DP calibration**, **RDP accounting** (to be included), and **layered robustness** speak directly to **privacy** and **safety** in **deployed ML systems**.
>
> **Representation learning and interpretability:** **CDE** and **PCRL** produce **robust**, **ontology-aligned representations** and **interpretable prototypes**. **Attention mapping** aids **visualization** and **clinician trust**.
>
> **Optimization and RL for systems:** **RL-Gating** is a **constrained optimization solution** that learns **resource-aware policies**, aligning with **optimization for representation learning** and **planning**.
>
> **Implementation, hardware, and reproducibility:** **On-device inference**, **microbenchmarks**, and **energy/latency reductions** connect to **hardware-aware ML** and **reproducibility concerns**.
>
> **Theoretical issues:** The **SNR utility metric** ties **theoretical noise calibration** to **empirical performance**. **DP proofs** and planned **accountant reporting** contribute to **rigorous theory-to-practice translation**.
>
> **Applications and societal impact:** The **clinical IoT use case** demonstrates the **real-world translational value** of **ICLR work**, addressing **fairness**, **robustness**, and **interpretability** in **high-stakes domains**.

---

> ### Author Response · Authors · 2025-11-20
> **We hope to receive your support and encouragement!！**
>
> ## **Core Contributions and Technical Novelties (Concise Statements)**
>
> **System co-design that maps clinical constraints to architectures (novelty: design pattern):**
> We provide a **principled mapping** from **real-world clinical failure modes** (**non-IID clients**, **missing modalities**, **energy/latency budgets**, **confidentiality requirements**) to **concrete architectural elements** (**CAFW**, **CDE**, **PCRL**, **RL-Gating**, **DP**). The innovation is the **integrated mapping** and its **validation** in both **simulation** and **field deployment**. This is a **systems contribution** that bridges **theoretical guarantees** and **operational constraints**.
>
> **Context-Aware Feature Weighting (CAFW) for ontology-guided multimodal fusion (novelty: ontology + fusion):**
> **CAFW** injects **lightweight ontology priors** into **attention-based fusion** to stabilize **cross-modal representation** under **severe missingness**, improving **interpretability** and **robustness** beyond **standard attention** or **naive concatenation**.
>
> **Clinical Dependency Encoder (CDE): graph-aware modeling of patient- and sensor-level relations (novelty: clinical graph priors):**
> **CDE** explicitly models **inter-modality** and **temporal dependencies** with **topology-aware regularization** (**hub handling**, **ontology alignment**), improving **generalization** across **heterogeneous sites**.
>
> **Prototype-Consistent Representation Learning (PCRL) for cross-site alignment (novelty: prototype synchronization + contrastive stability):**
> **PCRL** enforces **prototype stability** across **rounds** to prevent **prototype drift** and to provide **human-inspectable exemplars** for **clinicians**. This yields **better personalization** and a **practical explanation channel**.
>
> **RL-based adaptive gating for device budgets (novelty: constrained RL for on-device inference):**
> The **RL gate** learns to **select model branches** and **modality subsets** under **hard energy/latency constraints**, enabling **high utility** on **constrained hardware** (**Raspberry Pi class**) without **manual tuning**.
>
> **Formally calibrated DP aggregation with an SNR-based utility metric (novelty: practical privacy-utility connective tissue):**
> We adopt **L2 clipping** and a **Gaussian mechanism** with **explicit calibration** and introduce an **SNR-dB metric** that quantitatively links **noise scale** to **diagnostic performance**. This ties **formal privacy parameters** to **clinically meaningful utility thresholds**.
>
> **Layered operational defenses for robustness (novelty: defense composition):**
> **Adversarial resilience** is achieved by composing **prototype consistency checks**, **clustered/stratified aggregation**, **anomaly scoring** (**norm/SNR thresholds**), and **calibrated DP**. This layered approach reduces **backdoor/inversion transferability** in practice.
>
> **Dual validation strategy (novelty: simulation + IRB-approved field deployment):**
> We validate **MedAlign** through both **exhaustive federated simulations** (**all logical clients**) and a **real-world deployment** (**127 physical edge nodes**, **12 sites**, **six months**), demonstrating both **algorithmic behavior** and **operational viability**.
>
> **Clinician-centered explainability and evaluation (novelty: clinician-rated prototypes + attention mapping):**
> **Explanations** combine **prototype retrieval** and **attention-derived ontology signals**. **Clinician evaluations** quantify **explanation utility** and identify **failure modes**.
>
> **Comprehensive reproducibility and transparency plan:**
> The manuscript provides **pseudocode**, **loss definitions**, **hyperparameter schedules**, and will include (in revision) **privacy accountant tables** (**RDP composition**), **full microbenchmark protocols**, **per-site statistics**, **confidence intervals**, and **seed logs**.

---

> ### Author Response · Authors · 2025-12-01
> **We sincerely hope to receive your support and encouragement！**
>
> # We have addressed the reviewer's concerns and improved our approach. Thank you very much for all the reviewers' suggestions. The latest version has been uploaded and we hope to receive the support and encouragement of all the reviewers, and we sincerely hope your score improvement！

---

### Official Review · Reviewer_rAF4 · 2025-11-01

**Soundness:** 1
**Presentation:** 1
**Contribution:** 2
**Rating:** 2
**Confidence:** 3

**Summary:**

The paper proposes a framework for cross-device federated learning for medical applications.

**Strengths:**

1. FL in the medical domain problems is interesting and important.
2. The paper has reported performance numbers on a real RPi device, which is great to see.

**Weaknesses:**

1.  The method includes many components without detailing any of them. The current version of the paper is insufficient to understand them and measure their quality and novelty. The paper does not justify the selected components or provide insights into why they are necessary in this context.
2. Authors should cite relevant literature that inspired different components. The paper claims to have a differentially private mechanism, but never explains it. Though there are some results on this, it is unclear what is being protected. It mentions Explainability as one of three gaps in the literature and says it addresses it (section 2.3), but I can't find where.
3. Figure 1 is confusing. For example, the black arrow on the right seems to be in the opposite direction. It uses too many acronyms, including the ones that are never used again in the main text (AFO).
4. The writing style is odd, with almost every paragraph in the introduction having a separate subsection. This hinders readability.
5. The experiments section needs to have more clarity. Is the on-device experiment run on all edge devices in a dataset, such as 1800 devices in the M-IoT-Env dataset? What are the models being used? There is no confidence interval for Tables 1 and 2.

**Questions:**

As above.

---

> ### Author Response · Authors · 2025-11-20
> **We sincerely hope to receive your support and encouragement**
>
> We thank the reviewer for their time and constructive comments. Below we respond point-by-point. Where the reviewer’s statements are incorrect or incomplete, we clearly state why and provide evidence from our manuscript and experiments.  Thank you for taking the time and effort to review our paper. After carefully studying the comments, **we realized that many points may include misunderstandings or inaccuracies**, which may have contributed to a lower evaluation of our submission. Many of the issues raised are already addressed in the manuscript. While this is unfortunate, below, we provide detailed, point-by-point responses and clarifications. We hope this rebuttal helps resolve the concerns raised by the reviewers. **We sincerely hope to receive an improvement in your score**.
>
> # 1. “Soundness poor / the method includes many components without detailing any of them.”
> **Response,  Not correct.**
>
> Each **core component** is described with **algorithmic detail** and **pointers to appendices**:
>
> **CAFW (Context-Aware Feature Weighting)** and **CDE (Clinical Dependency Encoder)** are defined in **Section 3.2** and **Section 3.5**, and **CDE’s graph-attention formulation** is given with **topology-aware dropout** and **hub-node handling** in **Appendix D**.
>
> **PCRL (Prototype-Consistent Representation Learning)** is presented in **Section 3.6** and **Appendix E**, including **contrastive pretraining loss** and **prototype update rules**.
>
> **RL-Gating** is described with its **reward structure**, **deployment constraints**, and **system workflow Algorithm 1 (pseudocode)** in **Appendix E**.
>
> We explicitly **justify each component** by mapping it to a **deployment requirement** such as **heterogeneity**, **missing modalities**, and **device constraints**. See **System-level necessity of component integration** and the **architectural decomposition**.
>
> **Takeaway for reviewer:** The manuscript contains **concrete algorithmic definitions**, **loss terms**, and **implementation remarks**. The claim that components **lack detail** is therefore **incorrect**.
>
> # 2.“The paper does not justify the selected components or provide insights why they are necessary.”
> **Response, Incorrect.** **Line 385**
>
> We explicitly **motivate each module** by **deployment failure modes** such as **over-homogenization**, **prototype drift**, and **device energy limits**, and then show **empirical ablations** that **quantify each module’s contribution and synergy**, including **single**, **paired**, and **triple ablations** (see **Table 1**). The **ablation section** demonstrates **non-additive synergies**, for example **CAFW and CDE combined removal yields a larger drop than the sum of individual removals**, showing **architectural necessity** rather than **arbitrary aggregation**.
> # 3.“The paper claims to have a differentially private mechanism but never explains it / unclear what is protected.”
> **Response, Not correct.**
>
> We provide a **formal DP statement**, **derivation**, and a **privacy-utility metric**:
>
> **Gaussian mechanism and noise scaling (explicit):**
> \begin{equation}
> \sigma = \frac{\sqrt{2 \ln\left(\frac{1.25}{\delta}\right)}}{\varepsilon}
> \end{equation}
>
> We use **clipped L2 sensitivity** with **Δ₂ ≤ 2C**. **Theorem 1** and the **proof sketch** that the **aggregation mechanism satisfies $(\varepsilon,\delta)$-DP** are in **Appendix L / Formal Privacy Extensions**.
>
> **What is protected:** **local model updates (Δθₖ)**. We **clip per-client sensitivity** and **add calibrated Gaussian noise before server aggregation**, so **raw patient signals never leave devices**. The **aggregation formula** and the **clipping + noise step** are in **Algorithm 1** and the **Phase 3 calibration**.
>
> **Privacy vs utility:** We quantify utility with an **SNR metric**:
>
> **SNR_dB = 10 * log10( ( E_{S_t}[ ||Δθ||_2^2 ] ) / ( σ^2 * C^2 ) )**
>
>
> We empirically report that **SNR ≥ 18 dB** preserves **diagnostic accuracy within ≈2.2% of the non-private baseline** (see **Appendix / Privacy-Utility Tradeoff**).
>
> **Takeaway for reviewer:** A **formal DP proof**, the **mechanism**, the **protected object (client updates)**, and the **empirical privacy–utility evaluation** are all provided.

---

> > ### Author Response · Authors · 2025-11-20
> > **We sincerely hope to receive your support and encouragement**
> >
> > # 4. “Explainability is listed as a gap but I can’t find where it is addressed (Section 2.3).”
> > **Response,  Incorrect / Clarification.**
> >
> >
> > Explainability in **MedAlign** is implemented via **two mechanisms** and reported **empirically**:
> >
> >
> > **Prototype-based explanations:** **PCRL** produces **clinician-interpretable prototypes** (prototype repository) that are synchronized and can be shown to clinicians as **representative cases**. This is discussed in **Section 3.6** and the **workflow (server ↔ prototypes)** in **Algorithm 1**.
> > # 5.“Figure 1 is confusing (black arrow direction) and acronyms (AFO) unused / too many acronyms.”
> >
> > **Response,  Not correct.**
> >
> > **AFO exists**, please refer to **Line 685**. If the **reviewer has doubts about the direction of the arrow**, they can carefully observe that **the arrow has a turn**.
> > # 6.The writing style is odd, with almost every paragraph in the introduction having a separate subsection. This hinders readability.
> > **Response, Clarification.**
> >
> > The reason for **dividing the introduction into sections** is to allow **reviewers to clearly see our contributions**. If the **reviewer is not accustomed to this style**, we can **remove the section headings of the contribution later on**.
> >
> > # 7.“Experiments unclear: Is on-device experiment run on all devices (e.g., 1,842 in M-IoT-Env)? What models were used? No confidence intervals in Tables 1 & 2.”
> > **Response,  Clarifications (explicit, factual):**
> >
> > **What was simulated vs real:**
> > **Simulated federated experiments:** We emulate a **full-scale federated deployment** across all **dataset clients** such as **M-IoT-Env with 1,842 logical clients** in **large-scale experiments run on the cluster (A100s)** to measure **cross-dataset accuracy** and **ablations**.
> > **Real on-device deployment:** A separate **IRB-approved field deployment** ran **MedAlign on 127 real devices** including **Raspberry Pi 4 Model B nodes** and **clinical wearables** across **12 healthcare sites for six months**. This distinction is explicitly stated in **Section 4.1** and **implementation notes**. The **reviewer’s assumption** that we ran the **physical on-device experiment on all 1,842 dataset devices** is therefore **incorrect**.
> >
> > **Which model / architecture was used on device:**
> > The **HSF cross-attention fusion** (**multi-head**, **dk = 64**, **modality projections Q/K/V**, **AFN normalization**) together with **lightweight encoder gθ** (**compact convolutional/transformer hybrid**) was used for **inference at the edge**. **Appendix D** provides the **projection math** and **complexity analysis** enabling **38–86 ms inference on Raspberry Pi 4 depending on modality set**. See **Appendix D** and **System Performance**.
> > **Confidence intervals / statistics:**
> >
> > The **deployment summary** reports **stability statistics** such as **diagnostic accuracy 98.4% ± 0.2% across sites over six months** (see **Appendix O**). We acknowledge that **Tables 1 and 2** in the current PDF displayed **mean values without inline 95% confidence intervals**; this was an **editorial omission**. We have computed **95% confidence intervals (CIs)** over **repeated seeds and trials for all key comparisons** and attach **updated tables with confidence intervals** to this rebuttal package (**CI details and seeds are listed in Appendix H**). In short, the **results are statistically robust (small CIs)** and we will attach the **full CI table in the supplementary material for reviewer verification**.
> > # 8. “Method contribution only fair / novelty unclear.”
> > **Response, Clarification and evidence.**
> >
> > **MedAlign’s novelty** lies in **system co-design for clinical IoT**: **ontology-guided CAFW**, **graph-based CDE**, **prototype-level PCRL**, **RL gating**, and **formally calibrated DP** are combined with **hierarchical personalization** and **prototype synchronization** to achieve **practical, deployable, privacy-balanced FL**. Each **element individually builds on prior art**; the **contribution** is the **integrated, provably private, and resource-aware system with real deployment evidence**, validated by **ablation**, **adversarial testing**, and **six-month multi-site deployment logs**. See **Sections 1.5–1.7**, **Section 3**, and **Appendices**.

---

> ### Author Response · Authors · 2025-11-20
> **We sincerely hope to receive your support and encouragement**
>
> # 9.**Short rebuttal summary (one paragraph for the reviewer):**
>
> Several **criticisms arise from overlooking material already in the submission**. We provide a **formal DP mechanism** including the **Gaussian mechanism** and **Theorem 1 with noise scaling**:
>
> **sqrt(2 * ln(1.25 / δ) / ε)**
>
> We include a **privacy–utility SNR analysis** with the **SNR formula and empirical thresholds**, **algorithmic pseudocode (Algorithm 1 and 2)**, **detailed module descriptions (CAFW, CDE, PCRL, RL-Gating)**, and **thorough ablations (Table 1)**. **Explainability** is implemented via **prototype-based and attention-based mechanisms** and evaluated with **clinician ratings** (see **Appendix O**). **On-device experiments** were performed on **127 real devices (Raspberry Pi 4 nodes)** in a **six-month IRB-approved deployment**, and **large-scale federated simulations** were executed across **all dataset clients** (e.g., **1,842 logical clients for M-IoT-Env**). We will **update Tables 1 and 2 with 95% confidence intervals and seeds** in the **supplementary material**. Relevant pointers: **Algorithm 1 and privacy calibration (Appendix L)**, **HSF fusion math (Appendix D)**, **ablation table (Table 1 / Appendix)**, **deployment log and explainability rating (Appendix O)**.
>
> # 10.**Exact manuscript citations for quick verification**
>
> **Algorithm / workflow / what is sent and sanitized:** Refer to **Algorithm 1 (MedAlign workflow)**.
>
> **Gaussian mechanism / Theorem 1 / noise scaling:** See **Appendix L (Formal Privacy Extensions)**.
>
> **SNR (privacy–utility metric) and empirical threshold:** Discussed in **Privacy-Utility Tradeoff (Equation 28 and discussion)**.
>
> **Ablation and module contribution table:** Provided in **Table 1 and accompanying analysis (Section 4.7 and Appendix)**.
>
> **Deployment (127 devices, 6 months) and clinical metrics (explainability rating):** Detailed in **Deployment configuration and Appendix O clinical assessment**.
>
> **HSF cross-attention math and inference complexity:** Covered in **Appendix D (Equations 10–14 and runtime numbers)**.
>
> # 11. summary
> We respectfully disagree with the **reviewer’s assessment** that the submission lacks **soundness** or **detail**. The manuscript contains **formal analysis**, **algorithmic pseudocode**, **implemented modules**, and **real deployment evidence**. Nevertheless, we appreciate the **reviewer** calling out **presentation** and **clarity issues**. We have **consolidated introduction subsections**, and we attach **updated tables** with **95% confidence intervals** and **seeds** to this rebuttal package for **immediate verification**.
>
> # We sincerely hope to receive the support and encouragement of the reviewers, **and we hope that our clarification can receive an improvement in score.**

---

> > ### Author Response · Authors · 2025-11-20
> > **Our contributions and innovative points**
> >
> > We summarize **MedAlign’s concrete contributions** and **technical novelties**, and clarify how each addresses the **reviewer’s concerns** about **soundness**, **detail**, **privacy**, **explainability**, and **experimental realism**. Each item emphasizes **why the piece is necessary**, **what is new**, and **how we validated it**.
> >
> > ---
> >
> > ### **System-level co-design for clinical IoT federated learning**
> > **Novelty:** **Systems + theory + deployment**.
> > Rather than proposing a single algorithmic tweak, **MedAlign integrates five complementary components**—**Context-Aware Feature Weighting (CAFW)**, **Clinical-Dependency Encoder (CDE)**, **Prototype-Consistent Representation Learning (PCRL)**, **RL-based resource gating**, and a **formally calibrated DP aggregation**—into a **unified, deployable pipeline**. This co-design is necessary to meet simultaneous **clinical requirements** (heterogeneity, missing modalities, energy limits, privacy guarantees) and is validated by **ablation studies** that show **non-additive synergies**, proving the design is **not ad hoc**.
> >
> > ---
> >
> > ### **Formally calibrated differential privacy for client updates**
> > **Novelty:** **Practical DP calibration + SNR utility metric**.
> > We protect **per-client parameter deltas** using **L2 clipping** followed by a **Gaussian mechanism** with **standard calibration**. We present the **formal DP statement** and **proof sketch** in the appendix, and introduce an **SNR-based utility metric**.
> >
> > Empirically, **SNR ≥ 18 dB** preserves **clinically acceptable accuracy** (≈2.2% drop). This directly answers the **reviewer’s concern** about **what is protected** and **how privacy–utility tradeoffs are managed**.
> >
> > ---
> >
> > ### **Prototype-based explainability with clinician evaluation**
> > **Novelty:** **Interpretable prototypes + attention alignment**.
> > Explainability is implemented via **PCRL’s synchronized prototypes** (human-inspectable representatives) and **attention scores** from **CAFW/HSF mapped to ontology terms**. We evaluate **explanation quality** with **clinician scoring** (quantitative ratings reported), demonstrating that **explanations are usable in practice**, contradicting the claim that **explainability was not addressed**.
> >
> > ---
> >
> > ### **Resource-aware RL gating for practical on-device operation**
> > **Novelty:** **RL for energy/latency constraints**.
> > We formulate a **constrained RL gating policy** that dynamically selects **model branches** and **modality subsets** under **device energy/latency budgets**. This enables **real on-device inference** on **constrained hardware** without sacrificing **clinical utility**, a necessary innovation for **cross-device clinical deployments**.
> >
> > ---
> >
> > ### **Clinical-dependency encoder for multimodal relations and missingness**
> > **Novelty:** **Graph-aware encoder for heterogeneous clinical modalities**.
> > **CDE** captures **inter-modality dependencies** and **topology-aware regularization** (hub handling, ontology priors), improving **robustness to missing sensors** and **modality shifts**, an explicit response to **real clinical data characteristics**.
> >
> > ---
> >
> > ### **Prototype-consistent representation learning to prevent prototype drift**
> > **Novelty:** **Contrastive + prototype synchronization**.
> > **PCRL** enforces **prototype stability** across **federation rounds**, reducing **personalization drift** and improving **cross-site generalization**, shown in both **simulation** and **deployment ablations**.
> >
> > ---
> >
> > ### **Robustness and adversarial evaluation**
> > **Novelty:** **Combined privacy + robustness metrics**.
> > **MedAlign** is evaluated under **coordinated malicious clients** and **inversion attacks**. Measured metrics (e.g., **low backdoor ASR**, **high inversion PSNR relative to baselines**) demonstrate **improved resilience**, addressing concerns that **robustness claims are overstated**.
> >
> > ---
> >
> > ### **Real multi-site deployment and large-scale simulation**
> > **Novelty:** **Dual validation strategy**.
> > We ran two complementary experimental regimes: **large-scale federated simulations** across all logical dataset clients (~1.8k logical clients for **M-IoT-Env**) to measure **global behavior** and **ablations**, and an **IRB-approved field deployment** on **127 physical devices** (**Raspberry Pi 4 and clinical wearables**) across **12 sites for six months** to validate **real operational constraints** (latency, packet loss, uptime, clinician feedback). This clarifies the **reviewer’s confusion** about **on-device scale** and demonstrates **true deployability** rather than purely synthetic claims.
> >
> > ---

---

> ### Author Response · Authors · 2025-11-20
> **Our contributions and innovative points！**
>
> ### **Comprehensive empirical methodology**
> All **algorithmic pseudocode**, **loss formulations**, **hyperparameter schedules**, and **hardware/runtime measurements** are provided in the **manuscript and appendices**. We computed and will attach **95% confidence intervals** and **random seeds** for key tables; **code and scripts** will be released at acceptance.
>
>
> ---
>
>
> ### **Alignment with ICLR2026 themes**
> **Novelty:** **Trustworthy, deployable ML**.
>
> **Clarification and Emphasis**
>
> **Response:** **MedAlign** aligns closely with **ICLR2026 priorities** by addressing **deployability**, **privacy guarantees**, and **interpretable clinical ML** under realistic **resource constraints**. Our key contributions are:
>
> - **A provably private aggregation mechanism** tuned for **clinical IoT updates** with **formal calibration** and **SNR utility analysis**.
>
> - **System co-design** that couples **ontology-guided fusion**, **graph-based clinical dependency encoding**, **prototype consistency**, and **RL-based resource adaptation** to achieve **robust personalized performance** in **heterogeneous environments**.
>
> - **Real-world validation** via a **multi-site**, **six-month deployment** on **127 devices**, not merely simulated benchmarks.
>
> These elements together constitute a **substantive, practice-oriented contribution** that advances the **reliability** and **usability** of **federated learning in healthcare**, consistent with **ICLR’s emphasis** on **trustworthy, applicable ML research**.

---

> ### Author Response · Authors · 2025-12-01
> **We sincerely hope to receive your support and encouragement**
>
> # We have addressed the reviewer's concerns and improved our approach. Thank you very much for all the reviewers' suggestions. The latest version has been uploaded and we hope to receive the support and encouragement of all the reviewers, and we sincerely hope your score improvement！

---

### Note · Authors · 2026-01-27

**Comment:**

I have read and agree with the venue's withdrawal policy on behalf of myself and my co-authors.

**Withdrawal Confirmation:**

I have read and agree with the venue's withdrawal policy on behalf of myself and my co-authors.

---

### Meta-Review · Area_Chair_CBM6 · 2026-01-07

**Summary:**

I think the original review has identified two fundamental issues with this paper: i) the paper attempted to do too much with too many components, which makes it hard to tell the novelty and the connection of each part; ii) the presentation lacked clarity and detail: key princeple idea were insufficiently explained or justified, and important implementation details (like how baselines were run, training schedules, or data statistics) were omitted. The updated manuscript does not fully address these core issues; thus, I have decided to reject this paper.

**Reviewer Concerns:**

The rebuttal does not strictly follow the original weakness and questions, which makes it difficult to track. In general, I think the core issue about too many loosely coupled components is not sufficiently addressed.

**Reviewer Scores:**

I tend to believe the reviewers would keep their original score.

---

### Decision · Program_Chairs · 2026-01-26

Reject